# A time-resolved, multi-symbol molecular recorder via sequential genome editing

Junhong Choi[1,2 ✉], Wei Chen[1,3], Anna Minkina[1], Florence M. Chardon[1], Chase C. Suiter[1,4], Samuel G. Regalado[1], Silvia Domcke[1], Nobuhiko Hamazaki[1,2], Choli Lee[1], Beth Martin[1], Riza M. Daza[1] & Jay Shendure[1,2,5,6 ✉]

DNA is naturally well suited to serve as a digital medium for in vivo molecular recording. However, contemporary DNA-based memory devices are constrained in terms of the number of distinct 'symbols' that can be concurrently recorded and/or by a failure to capture the order in which events occur[1]. Here we describe DNA Typewriter, a general system for in vivo molecular recording that overcomes these and other limitations. For DNA Typewriter, the blank recording medium ('DNA Tape') consists of a tandem array of partial CRISPR–Cas9 target sites, with all but the first site truncated at their 5′ ends and therefore inactive. Short insertional edits serve as symbols that record the identity of the prime editing guide RNA[2] mediating the edit while also shifting the position of the 'type guide' by one unit along the DNA Tape, that is, sequential genome editing. In this proof of concept of DNA Typewriter, we demonstrate recording and decoding of thousands of symbols, complex event histories and short text messages; evaluate the performance of dozens of orthogonal tapes; and construct 'long tape' potentially capable of recording as many as 20 serial events. Finally, we leverage DNA Typewriter in conjunction with single-cell RNA-seq to reconstruct a monophyletic lineage of 3,257 cells and find that the Poisson-like accumulation of sequential edits to multicopy DNA tape can be maintained across at least 20 generations and 25 days of in vitro clonal expansion.

How do we learn the order of molecular events in living systems? A first approach is direct observation, for example, live-cell fluorescence microscopy to order interactions in real time. A second approach is time series experiments, for example, destructively sampling and transcriptionally profiling a system at different time points, followed by pseudotemporal ordering. A third approach is epistatic analysis, for example, ordering the actions of genes by comparing the phenotypes of single and double mutants. Although these and other approaches have important strengths, they are also limited in key ways. For example, live imaging is largely restricted to in vitro models. For time series experiments, resolution and accuracy are constrained by the frequency of sampling and the reproducibility of the biological process under investigation. Epistatic analysis is confounded by pleiotropy, particularly in multicellular organisms.

Another approach, theoretically promising but methodologically underdeveloped relative to the aforementioned alternatives, is a DNA memory device[3], which we define here as an engineered system for digitally recording molecular events through permanent changes to a cell's genome that can be read out post factum. Thus far, several proof-of-concept DNA memory devices have been described that leverage diverse approaches for the 'write' operation, including site-specific recombinases (SSRs)[4,5], CRISPR–Cas9 genome editing[6–9], CRISPR integrases[10,11], terminal deoxynucleotidyl transferases (TdTs)[12], base-pair misincorporation[13], base editing[14] and others[1].

The nature of the write operation in such DNA memory devices shapes their performance in terms of channel capacity for encoding and decoding signals, temporal resolution, interpretability and portability[1]. For example, SSRs record molecular signals with high efficiency, but the number of distinct signals that can be concurrently recorded is limited by the number of available SSRs. DNA memory devices relying on CRISPR–Cas9 can potentially overcome this limitation, for example, if each signal of interest were coupled to the expression of a different guide RNA (gRNA), but in that case each signal would also require its own target(s). Furthermore, the CRISPR–Cas9 molecular recorders described thus far rely on double-stranded breaks (DSBs) and non-homologous end joining (NHEJ) to 'scar' target sites[1]. In addition to being toxic, frequent DSBs often excise or corrupt consecutively located target sites, the molecular equivalent of accidental data deletion.

A further handicap of nearly all DNA memory devices described thus far is that, while recordings might stochastically accumulate at unordered target sites, the order in which they occurred is not explicitly captured. CRISPR spacer acquisition systems, which rely on signal-induced, unidirectional incorporation of DNA spacers or transcript-derived tags to an expanding CRISPR array, overcome this limitation[10,11,15–17]. However, at least thus far, their reliance on accessory integration host factors has restricted such recorders to prokaryotic

[1]Department of Genome Sciences, University of Washington, Seattle, WA, USA. [2]Howard Hughes Medical Institute, Seattle, WA, USA. [3]Molecular Engineering and Sciences Institute, University of Washington, Seattle, WA, USA. [4]Molecular and Cellular Biology Program, University of Washington, Seattle, WA, USA. [5]Brotman Baty Institute for Precision Medicine, Seattle, WA, USA. [6]Allen Discovery Center for Cell Lineage Tracing, Seattle, WA, USA. ✉e-mail: junhongc@uw.edu; shendure@uw.edu

systems. Another approach, CHYRON, enables directional writing of information to DNA by combining self-targeting CRISPR gRNAs with the expression of TdT, whose presence shifts the most likely outcome of NHEJ from short deletions to short insertions[18]. While this approach unidirectionally inserts nucleotides in a signal-responsive manner, it continues to rely on NHEJ-mediated repair of DSBs. Furthermore, because each gRNA/target yields a homogenous signal (TdT-mediated insertions of variable length), it is not clear how this approach could be used to explicitly record the precise order of more than a handful of distinct signals. Finally, at least two groups have independently developed 'logic-circuit architectures' that use sequential base editing to record the order and identity of biological signals in both bacterial and mammalian cells (DOMINO[19] and CAMERA[14]). However, because base editors are currently limited to writing single-base substitutions to predefined targets, the order of signals can only be recorded via preprogrammed circuits, rendering multiplex recording challenging.

Here we describe a DNA memory device that is (1) highly multiplexable, that is, compatible with the concurrent recording of at least thousands of distinct symbols or event types; (2) sequential and unidirectional in recording events to DNA and therefore able to explicitly capture the precise order of recorded events; and (3) active in mammalian cells. This system, which we call DNA Typewriter, begins with a tandem array of partial CRISPR–Cas9 target sites (DNA Tape), all but the first of which are truncated at their 5′ ends and are therefore inactive (Fig. 1a–c). Each of many prime editing gRNAs (pegRNAs), together with the prime editing enzyme[2], is designed to mediate the insertion of a *k*-mer within the sole active site of the tandem array, which is initially its 5′-most target site. In the simplest implementation, all pegRNAs target the same 20-bp spacer but each encodes a unique 'symbol' in the form of a *k*-mer insertion. Specifically, the 5′ portion of the *k*-mer insertion is the variable and encodes the identity of the pegRNA, while its 3′ portion is constant and activates the subsequent target site in the tandem array by restoring its 5′ end. Thus, each successive edit records the identity of the pegRNA mediating the edit while also shifting the position of the active target site by one unit along the array. At any moment, an intact spacer and protospacer adjacent motif (PAM) are present at only one location along the array, analogous to the 'write-head' of a disk drive or the 'type guide' of a typewriter.

## Proof of concept of DNA Typewriter

To test this idea, we designed a DNA Tape (TAPE-1) by modifying a spacer sequence previously shown to be highly amenable to prime editing by the PE2 enzyme (HEK293 target 3, or HEK3)[2]. In TAPE-1, a 3-bp key (GGA) is followed by a tandem array of a 14-bp monomer (TGATGGTGAGCACG) that includes the PAM sequence (TGG) at positions 4–6. At the 5′-most end of the TAPE-1 array, the key sequence, the first 14-bp monomer and the first 6 bases of the subsequent 14-bp monomer collectively make up an intact 20-bp spacer and PAM (Fig. 1a). We further designed a set of 16 pegRNAs to target TAPE-1, with each pegRNA programming a distinct 5-bp insertion (Fig. 1b). The first 2 bp of the insertion is unique to each of the 16 pegRNAs. The remaining 3 bp of the insertion corresponds to the key (GGA). We reasoned that, when a pegRNA/PE2-mediated insertion occurred at the active TAPE-1 site, it would (1) record the identity of the pegRNA via the 2-bp portion of the insertion; (2) inactivate the current active site by disrupting its sequence; and (3) activate the next monomer along the array, as the newly inserted GGA key, together with the subsequent 20 bp, creates an intact 20-bp spacer and PAM. In the next iteration of genome editing, a pegRNA-mediated insertion to the second monomer would be recorded while also moving the type guide to the third monomer and then to the fourth, the fifth and so on (Fig. 1c).

We synthesized and cloned TAPE-1 arrays with varying numbers of monomer units (2×TAPE-1, 3×TAPE-1, 5×TAPE-1) and stably integrated these arrays into the genome of HEK293T cells via the piggyBac system. We then transiently transfected the resulting cells with a pool

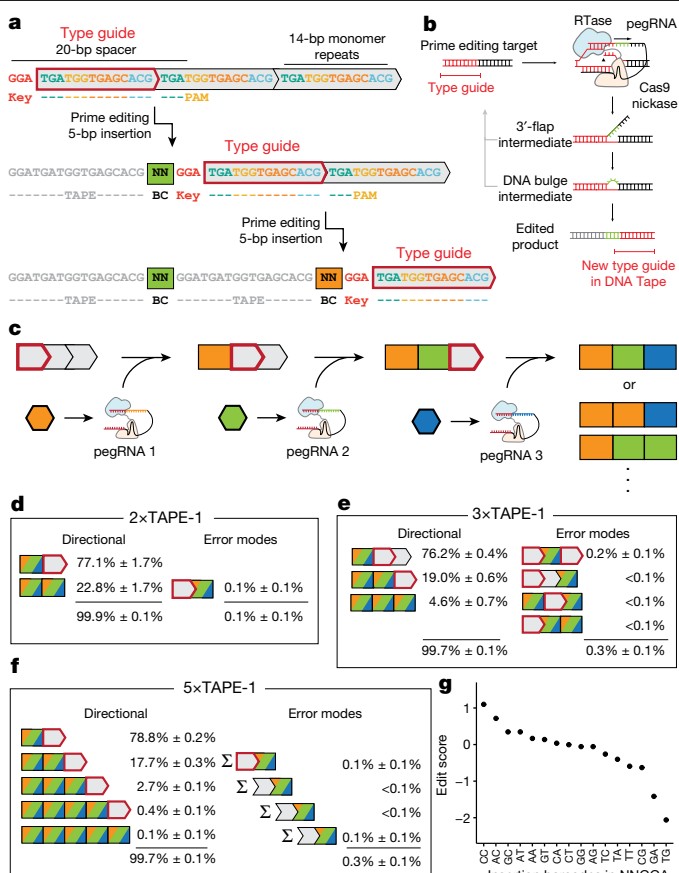

**Fig. 1 | Sequential genome editing with DNA Typewriter. a**, Schematic of two successive editing events at the type guide, which shifts in position with each editing event. The DNA Tape consists of a tandem array of CRISPR–Cas9 target sites (grey boxes), all but the first of which are truncated at their 5′ ends and therefore inactive. The 5-bp insertion includes a 2-bp pegRNA-specific barcode as well as a 3-bp key that activates the next monomer. Because genome editing is sequential in this scheme, the temporal order of recorded events can simply be read out by their physical order along the array. **b**, Schematic of prime editing with DNA Typewriter. Prime editing recognizes a CRISPR–Cas9 target and modifies it with the edit specified by the pegRNA[2]. With DNA Typewriter, an insertional editing event generates a new prime editing target at the subsequent monomer. **c**, Schematic of ordered recording via DNA Typewriter. Individual pegRNAs are potentially event driven[36] or constitutively expressed, together with the PE2 enzyme. **d–f**, Specificity of genome editing on versions of TAPE-1 with two (**d**), three (**e**) or five (**f**) monomers. Cells bearing stably integrated TAPE-1 target arrays were transfected with a pool of plasmids expressing pegRNAs and PE2. Each class of outcomes is inclusive of all possible NNGGA insertions; collectively, the classes shown include $2^n - 1$ possible outcomes, where *n* is the number of monomers. We observe that editing of any given target site is highly dependent on the preceding sites in the array having already been edited. **g**, Edit scores of 16 barcodes used in the experiment with 5×TAPE-1. Edit scores for each insertion are calculated as the $\log_2$-scaled ratio between the insertion frequencies and the abundances of pegRNAs in the plasmid pool, averaged over *n* = 3 transfection replicates.

of plasmids designed to express PE2 (pCMV-PE2-P2A-GFP; Addgene, 132776) and 16 pegRNAs, each programmed to insert an NNGGA barcode to TAPE-1, and collected the cells after 4 days. The TAPE-1 region was PCR amplified from genomic DNA and sequenced.

For each TAPE-1 array, we categorized sequencing reads into those in which (1) no editing occurred; (2) the observed pattern was consistent with sequential, directional editing; and (3) the observed pattern was inconsistent with sequential, directional editing (Fig. 1d–f and Supplementary Table 1). Overall editing rates were modest, as only 4.7% ± 0.5%,

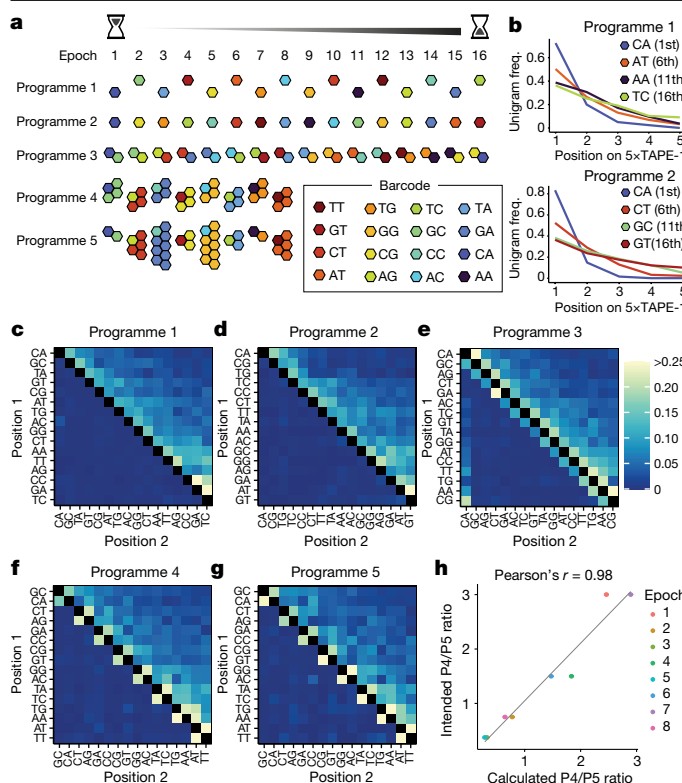

**Fig. 2 | Transfection programmes for 16 sequential epochs. a**, Schematic of five transfection programmes over 8 or 16 epochs. For programmes 1 and 2, pegRNAs with single barcodes were introduced in each epoch for 16 epochs. The specific orders aimed to maximize (programme 1) or minimize (programme 2) the edit distances between temporally adjacent transfections. For programme 3, pegRNAs with two different barcodes were introduced at a 1:1 ratio for 16 epochs, with one barcode always shared between adjacent epochs (and between epochs 1 and 16). For programmes 4 and 5, pegRNAs with two different barcodes were introduced either at a constant ratio (1:3) or at varying ratios in each epoch (1:1, 1:2, 1:4 or 1:8) for eight epochs, respectively. **b**, Barcode frequencies across five insertion sites in 5×TAPE-1 in programmes 1 and 2 following epoch 16. Barcodes introduced in early epochs are more frequently observed at the first site. **c–g**, Bigram transition matrices for programmes 1 (**c**), 2 (**d**), 3 (**e**), 4 (**f**) and 5 (**g**). Barcodes are ordered from early (left/top) to late (right/bottom). **h**, Calculated versus intended relative frequencies between programmes 4 and 5. Programme ratios were calculated by combining sequencing reads from n = 3 independent transfection experiments.

5.2% ± 0.6% and 5.9% ± 0.8% of all reads for 2×TAPE-1, 3×TAPE-1 and 5×TAPE-1, respectively, exhibited any editing. However, within the set of reads showing edits, the data were overwhelmingly consistent with sequential, directional editing. For example, with 2×TAPE-1, the second monomer was edited in 22.8% ± 1.7% of reads in which the first monomer was also edited (Fig. 1d). By contrast, the second monomer was only edited in 0.6% of reads in which the first monomer was not edited. This observation strongly suggests that edits of the second monomer were dependent on an edit of the first monomer having already occurred. Furthermore, this confirms that the 3-bp mismatch at the PAM-distal end of 'inactive' spacers of the TAPE-1 design is sufficient to inhibit prime editing. Data obtained from 3×TAPE-1 and 5×TAPE-1 were also consistent with sequential genome editing. For example, 98.5% (3×TAPE-1) and 99.0% (5×TAPE-1) of reads that were edited at the second monomer were also edited at the first monomer, while 97.6% (3×TAPE-1) and 98.8% (5×TAPE-1) of reads that were edited at the third monomer were also edited at the first and second monomers (Fig. 1e,f). These results were consistent across three transfection replicates (Supplementary Table 1).

An interesting phenomenon was that, while the observed editing rate of the first TAPE-1 monomer was ~6%, the editing rates of the second or third TAPE-1 monomer, conditional on the preceding monomers already being edited, were ~20% (Extended Data Fig. 1a). A simple explanation for this ~14% greater 'elongation' than 'initiation' of editing is that some integrated tapes are more amenable to prime editing than others, resulting in an excess of fully unedited tapes. However, we also observed a similar pattern with episomal tapes, as well as following multiple sequential transfections with pegRNA- and PE2-expressing plasmids to edit integrated tapes (7–15% increase in the conditional editing efficiency of the second site). Factors that might contribute to the observed 'pseudo-processivity' include heterogeneous susceptibility of cells to transfection, chromatin context[20,21] and cell cycle phase, but the primary explanation remains unclear. We also observed modest reductions in the conditional editing efficacy after the second site (1–10% decreases), which might be explained simply by each site being 'active' for less time than its predecessor.

We next analysed the distribution of the 16 NNGGA barcode insertions, focusing on 5×TAPE-1. Their frequencies were correlated across three replicates as well as between the first and second target sites (Pearson's r = 0.97–0.99; Extended Data Fig. 1b,c). The observed variation was partly explained by the relative abundances of the individual pegRNAs in the plasmid pool (Pearson's r = 0.87; Extended Data Fig. 1d). To explore whether the sequence of the insertion itself influences editing efficiency, we repeated the experiment with an equimolar pool of 16 pegRNA-expressing plasmids that had been individually cloned and purified (rather than cloned as a pool). For each of the NNGGA insertions in each experiment, we calculated 'edit scores' as the $\log_2$-scaled insertion frequencies normalized by the abundances of pegRNAs in the corresponding plasmid pools (Fig. 1g). The maximal edit score difference between the best barcode (CCGGA with an edit score of 0.98) and the worst barcode (TGGGA with an edit score of −2.38) was 3.36, that is, a nearly ten-fold difference in editing efficiency. However, 10 of 16 barcodes exhibited efficiencies within a two-fold range. Edit scores were well correlated between 5×TAPE-1 edited by the 16 pegRNA plasmids pooled before versus after cloning (Spearman's p = 0.97; Extended Data Fig. 1e), in line with an insertion sequence-dependent bias. Indeed, when we used the relative efficiencies observed in the 'post-cloning pooling' experiment to correct the TAPE-1 unigram barcode frequencies measured in the 'pre-cloning pooling' experiment, the correlation of the frequencies with the abundance of the corresponding pegRNAs in the plasmid pool improved (Pearson's r = 0.87→0.94; Extended Data Fig. 1d) and vice versa (Pearson's r = 0.27→0.67; Extended Data Fig. 1f).

## Enhanced prime editing of DNA Tape

Several strategies to improve the efficiency of prime editing through modular engineering were recently reported: (1) addition of degradation-resistant secondary structure to the 3′ end of the pegRNA[22] (resulting in enhanced pegRNAs, or epegRNAs); (2) introduction of human MLH1 dominant-negative peptide (hMLH1dn) to favour the intended edit[23]; and (3) modifications to the primary sequence of the prime editing enzyme[23] (resulting in PEmax). Combined deployment of these strategies has been reported to improve editing efficiency by ~3.5-fold in HEK293T cells and ~72-fold in HeLa cells, relative to PE2 and pegRNAs[23].

As our initial experiments with PE2 and pegRNAs resulted in only modest editing of the first site of TAPE-1 (~6%), we sought to incorporate these new strategies. We cloned a pool of U6-driven epegRNAs, each programmed to insert an NNGGA barcode to TAPE-1, and transfected them into HEK293T cells in which 5×TAPE-1 was integrated (5×TAPE-1(+) HEK293T) along with a plasmid expressing PEmax and hMLH1dn (pCMV-PEmax-P2A-hMLH1dn; Addgene, 174828). After 4 days, we collected genomic DNA and then PCR amplified and sequenced TAPE-1. We observed 18.1% ± 0.5% editing of the first site (Extended Data Fig. 2a),

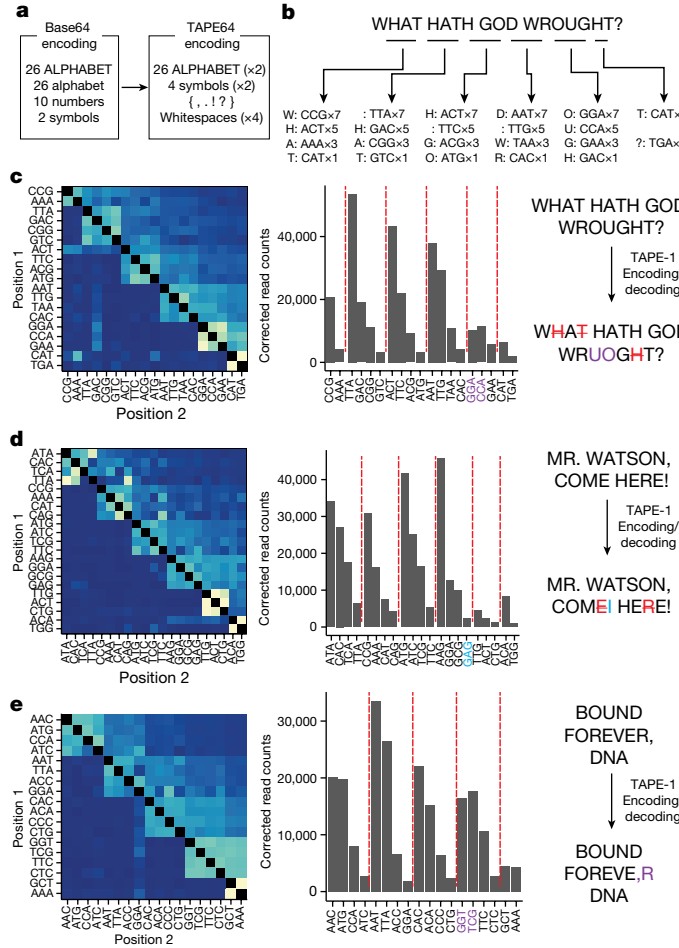

**Fig. 3 | Recording and decoding short digital text messages with DNA Typewriter. a**, Base64 binary-to-text was modified to assign 64 NNNGGA barcodes for TAPE-1 to 64 text characters. **b**, Illustration of the encoding strategy for "WHAT HATH GOD WROUGHT?", which has 22 characters including whitespaces. The message is grouped into sets of four characters, converted to NNN barcodes according to the TAPE64 encoding table, and plasmids corresponding to each set are mixed at a ratio of 7:5:3:1 for transfection. To encode 22 characters, we sequentially transfected 5 sets of 4 characters and 1 set of 2 characters 3 days apart into PE2(+) 5×TAPE-1(+) HEK293T cells. **c**–**e**, Decoding of three messages based on sequencing of the following 5×TAPE-1 arrays: "WHAT HATH GOD WROUGHT?" (**c**), "MR. WATSON, COME HERE!" (**d**) and "BOUND FOREVER, DNA" (**e**). For each message, the full set of NNNGGA insertions was first identified and cotransfected sets of characters were then identified from the bigram transition matrix (left). Within each set of characters inferred to have been cotransfected, ordering was based on corrected unigram counts (middle), resulting in the final decoded message (right). Misordered characters within each recovered message are coloured purple, missing characters are coloured red with strikethrough, and unintended characters are coloured light blue. Both two-dimensional histogram and corrected read counts were calculated by combining sequencing reads over n = 3 independent transfection experiments. Read counts were corrected using the edit score for each insertion barcode.

a nearly three-fold increase relative to PE2 and pegRNAs, while editing remained overwhelmingly sequential (>99.5%). We then cloned four more pools, encoding 6-bp (NNNGGA) to 9-bp (NNNNNNGGA) barcodes. The epegRNA–PEmax–hMLH1dn prime editing system achieved reasonably high efficiencies for longer insertions (for example, 10.6% ± 0.5% for 9-bp insertions; Extended Data Fig. 2a). Edit scores for pegRNA–PE2 versus epegRNA–PEmax–hMLH1dn were highly correlated (Spearman's p = 0.96 for NNGGA and Spearman's p = 0.88 for NNNNGGA; Extended Data Fig. 2b–e). The edit scores for epegRNAs were

more uniform than those for standard pegRNAs, as 14 of 16 NNGGA barcodes exhibited efficiencies within a two-fold range (Extended Data Fig. 2c) and 59 of 64 NNNGGA barcodes exhibited efficiencies within a four-fold range (Extended Data Fig. 2e). We also calculated edit scores for more than 1,900 barcodes in NNNNNNGGA (or 6N+GGA) TAPE-1-targeting epegRNAs in a single experiment (Extended Data Fig. 2f–i), markedly expanding the number of unique symbols that can be encoded and deployed to write to a shared DNA Tape by two orders of magnitude relative to our original NNGGA experiment. Overall, 1,509 of the 1,908 6N+GGA barcodes exhibited efficiencies with edit scores between −1 and 1, that is, a four-fold range (Extended Data Fig. 2h).

To evaluate the compatibility of DNA Typewriter with cell types other than HEK293T cells, we integrated the 5×TAPE-1 target into mouse embryonic fibroblasts (MEFs) and mouse embryonic stem (mES) cells using the piggyBac transposase system and transfected cells with either a pool of 16 NNGGA epegRNAs or a pool of 64 NNNGGA epegRNAs with PEmax- and hMLH1dn-expressing plasmids through electroporation with DNA plasmids. After 4 days, we collected genomic DNA and then amplified and sequenced TAPE-1. We observed 7.0–18.1% editing of the first site after 4 days (Extended Data Fig. 2j). In mESCs, where prolonged culturing was permitted compared with MEFs, we performed a second transfection with the same set of epegRNA-, PEmax- and hMLH1dn-expressing plasmids, 4 days after the first transfection. The cumulative editing of the first site increased to 28.7% ± 2.8% when the sample was collected another 4 days after the second transfection. Of note, the edit scores for NNGGA and NNNGGA pegRNAs in mESCs were reasonably well correlated with those measured in HEK293T cells (Extended Data Fig. 2k,l), suggesting that measurements of relative pegRNA efficiencies made in HEK293T cells are applicable to other cell types. Collectively, these results demonstrate that the performance of DNA Typewriter can be improved using methods that enhance prime editing and, furthermore, that the method can be used in primary and stem cells. Overall, we suspect that the range and efficiency of DNA Typewriter will be tightly coupled to that of prime editing, which has also been demonstrated to work in human induced pluripotent stem cells (iPSCs) and primary human T cells[23].

## Screening additional DNA Tape sequences

Our TAPE-1 construct exhibited sequential, directional editing, wherein the editing of any given site along the array was strongly dependent on all preceding sites having already been edited. This behaviour is consistent with DNA Typewriter's design, as the key sequence must be inserted 5′ to any given monomer within the DNA Tape to complete the spacer that is recognized by any of the gRNAs used. However, performance would presumably be corrupted by non-specific editing, for example, if a guide were able to mediate edits to a non-type-guide monomer despite several mismatches at the 5′ end of the spacer[24,25].

Although TAPE-1 exhibited reasonable efficiency and specificity, we sought to explore whether this would be the case for other spacers. To this end, we designed and synthesized 48 TAPE constructs (TAPE-1 through TAPE-48), each derived from one of eight basal spacers that previously demonstrated reasonable efficiency for prime editing[2,26,27] and one of six design rules that vary monomer sequence, key sequence and key/monomer length (Extended Data Fig. 3a). In each of these 48 constructs, a 3×TAPE region was accompanied by a pegRNA-expressing cassette designed to target it with a 4- to 6-bp insertion (16 possible 2-bp barcodes followed by a 2- to 4-bp key sequence). We then transiently transfected HEK293T cells with PE2-encoding plasmid and a pool of 48 pegRNA-by-3×TAPE constructs and collected cells after 4 days. The 3×TAPE region was PCR amplified from genomic DNA and sequenced.

We calculated two quantities for each 3×TAPE array: (1) efficiency, calculated by summing all edited reads and dividing by the total number of reads, and (2) sequential error rate, calculated by summing all

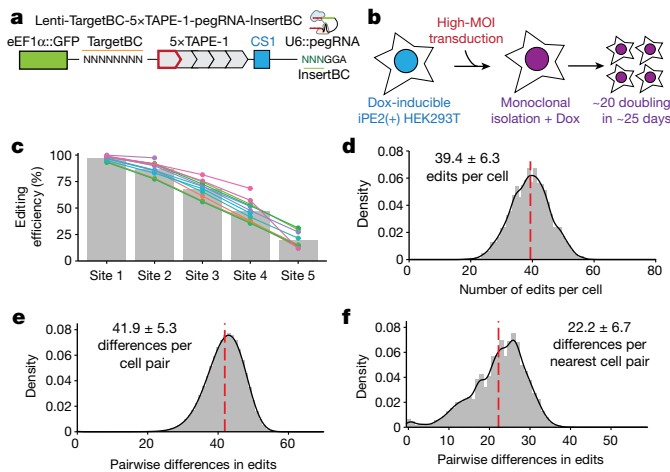

**Fig. 4 | Reconstruction of a monophyletic cell lineage tree using DNA Typewriter and scRNA-seq. a**, Schematic of the lentiviral vector used in the DNA Typewriter-based lineage tracing experiment[38]. The integration cassette includes a 5×TAPE-1 sequence associated with an 8-bp random barcode (TargetBC) and a pegRNA expression cassette. The pegRNA targets TAPE-1 and inserts 6 bp, in which the first 3 bp is the random barcode (InsertBC) and the last 3 bp is the key sequence of GGA for TAPE-1. Each TargetBC-5×TAPE-1 array is embedded in the 3′ UTR of the eGFP gene with an RNA capture sequence at its 3′ end and transcribed from the eEF1α promoter. **b**, Schematic of the monophyletic lineage tracing experiment. A HEK293T line with Dox-inducible PE2 expression was transfected with the lentiviral construct shown in **a** at a high MOI. A monoclonal line was then established and expanded in the presence of Dox. During expansion, pegRNAs expressed by TargetBC-defined integrants compete to mediate insertions at the type guides of TAPE-1 arrays within the same cell. **c**, Cumulative editing of each site within TAPE-1. Each coloured line shows the cumulative editing rate for 1 of 13 TargetBCs. Grey bars denote the cumulative editing of TAPE-1 sites across all 13 independent TargetBCs within the $n = 1$ single-cell experiment. **d**, Histogram of the number of edits across 59 editable sites in each cell. The red dashed line denotes the average. **e**, Histogram of the number of differences across the 59 editable sites for all possible pairs of the 3,257 sampled cells. The red dashed line denotes the average. **f**, Distribution of the number of pairwise differences between each cell and its 'nearest neighbour' among the 3,257 sampled cells.

edited reads inconsistent with sequential, directional editing and dividing by the total number of edited reads (Extended Data Fig. 3b). Of note, our initial TAPE-1 construct had one of the lowest sequential error rates among the 48 tested DNA Tapes. The only construct that had a lower sequential error rate than TAPE-1 was TAPE-6, which was derived from the same basal spacer (HEK3) but had a 4-bp rather than a 3-bp key sequence. Indeed, across the full experiment, a longer key sequence was associated with a lower sequential error rate (Extended Data Fig. 3c). Performance differences between basal spacers were modest, with DNA Tapes based on the HEK3 and FANCF spacers exhibiting the best combination of efficiency and specificity (Extended Data Fig. 3d). Among FANCF-based spacers, TAPE-27 exhibited over 50% greater efficiency than TAPE-1 but also had a two-fold-higher sequential error rate (Extended Data Fig. 3b). Performance characteristics were highly consistent when we repeated the experiment with integration rather than transient transfection of the constructs (Extended Data Fig. 3e).

Overall, these results show considerable variation in efficiencies and sequential error rates, specific to particular 13- to 15-bp TAPE sequences. Although a single well-performing monomer such as either TAPE-1 or TAPE-27 is sufficient to construct a generic substrate to which thousands of distinct symbols can be written, additional screening might yield monomers with even better performance characteristics and would also facilitate modelling of the sequence determinants of monomer performance[24–26,28].

## Recording complex event histories

We next asked whether we could apply DNA Typewriter to record, recover and decode complex event histories. We prepared a set of synthetic signals by individually cloning 16 individual pegRNA-expressing plasmids, each encoding a unique 2-bp barcode insertion to TAPE-1. We also prepared a polyclonal population of HEK293T cells with integrated 5×TAPE-1 to serve as the substrate for recording. Finally, we designed a set of five 'transfection programmes'–complex event histories that we could attempt to record and then subsequently decode (Fig. 2a).

At the beginning of each epoch of each transfection programme, one or more pegRNA plasmids were introduced to a population of HEK293T cells with integrated 5×TAPE-1 (5×TAPE-1(+) HEK293T) via transient transfection of plasmids expressing the corresponding pegRNA(s) and PE2. After each transfection, cells were passaged the next day into a new plate and excess cells were collected for genomic DNA. 5×TAPE-1 from each epoch of each programme was amplified and sequenced. Successive epochs occurred at 3-day intervals.

Programmes 1 and 2 each consisted of a distinct, non-repeating sequence of transfection of the 16 pegRNAs, that is, one per epoch. The specific orders aimed to maximize (programme 1) or minimize (programme 2) the edit distances between temporally adjacent signals. On the basis of sequencing of 5×TAPE-1 after epoch 16, we observed that barcodes introduced in the early epochs were more frequent at the first target site (site 1) than barcodes introduced at late epochs (Fig. 2b). This is expected, as each editing round shifts more of the type guides to site 2 (and subsequently to site 3 to site 5) (Extended Data Fig. 4a), with minimal effects on the integrity of the 5×TAPE-1 array (Extended Data Fig. 4b). A trivial decoding approach would be to simply arrange barcodes in order of decreasing site 1 unigram frequency, but for both programmes 1 and 2 this resulted in an incorrect order (Extended Data Fig. 4c).

However, inference can be improved by leveraging the sequential aspect of DNA Typewriter, for instance, by analysing bigram frequencies or pairwise appearance of events as used to infer orders from the CRISPR–Cas spacer acquisition process (Cas1–Cas2 system used in bacteria)[11,29]. For example, if signal B preceded signal A, then we expect many more B–A bigrams than A–B bigrams at adjacent edited sites in 5×TAPE-1. In Fig. 2c,d, we show heatmaps of bigram frequencies measured from all four pairs of adjacent editing sites on 5×TAPE-1, arranged by the true order in which the signals were introduced for programmes 1 and 2. Indeed, the bigram frequencies appear to capture event order information, as evidenced by the gross excess of observations immediately above versus immediately below the diagonal (for example, in programme 1, CA-GC >> GC-CA). One way to leverage this information is by enumerating 'ordering rules' among all events for possible permutations and then checking which the observed data best match[11,29]. However, the number of ordering rules for $n$ events increases to the order of $n^2$ (for ordering of 16 events, there are 136 ordering rules, or $(n^2 + n)/2$ in general), while the number of possible permutations increases to $n$ factorial. As a more computationally efficient approach, we implemented the following algorithm: (1) initialize with the event order inferred from site 1 unigram frequencies; (2) iterate through adjacent epochs from beginning to end and swap signals A and B if the bigram frequency of B–A is greater than that for A–B; and (3) repeat step 2 until no additional swaps are necessary. For both programmes 1 and 2, this algorithm resulted in correct ordering of the 16 signals, out of 16 factorial or 21 trillion possibilities (Extended Data Fig. 4c). This inference was robust to the sequencing depth, as the correct order could be reconstructed from as few as 2,500 reads of the 5×TAPE-1 amplicon (Extended Data Fig. 4d).

The dearth of bigrams inconsistent with the true order, illustrated by the lack of signal below the diagonal in the programme 1 and programme 2 heatmaps (Fig. 2c,d), indicates minimal interference between adjacent epochs; that is, transfected pegRNAs from adjacent epochs did not overlap in their activities. To evaluate performance in the presence of such overlap, we designed programme 3, in which two

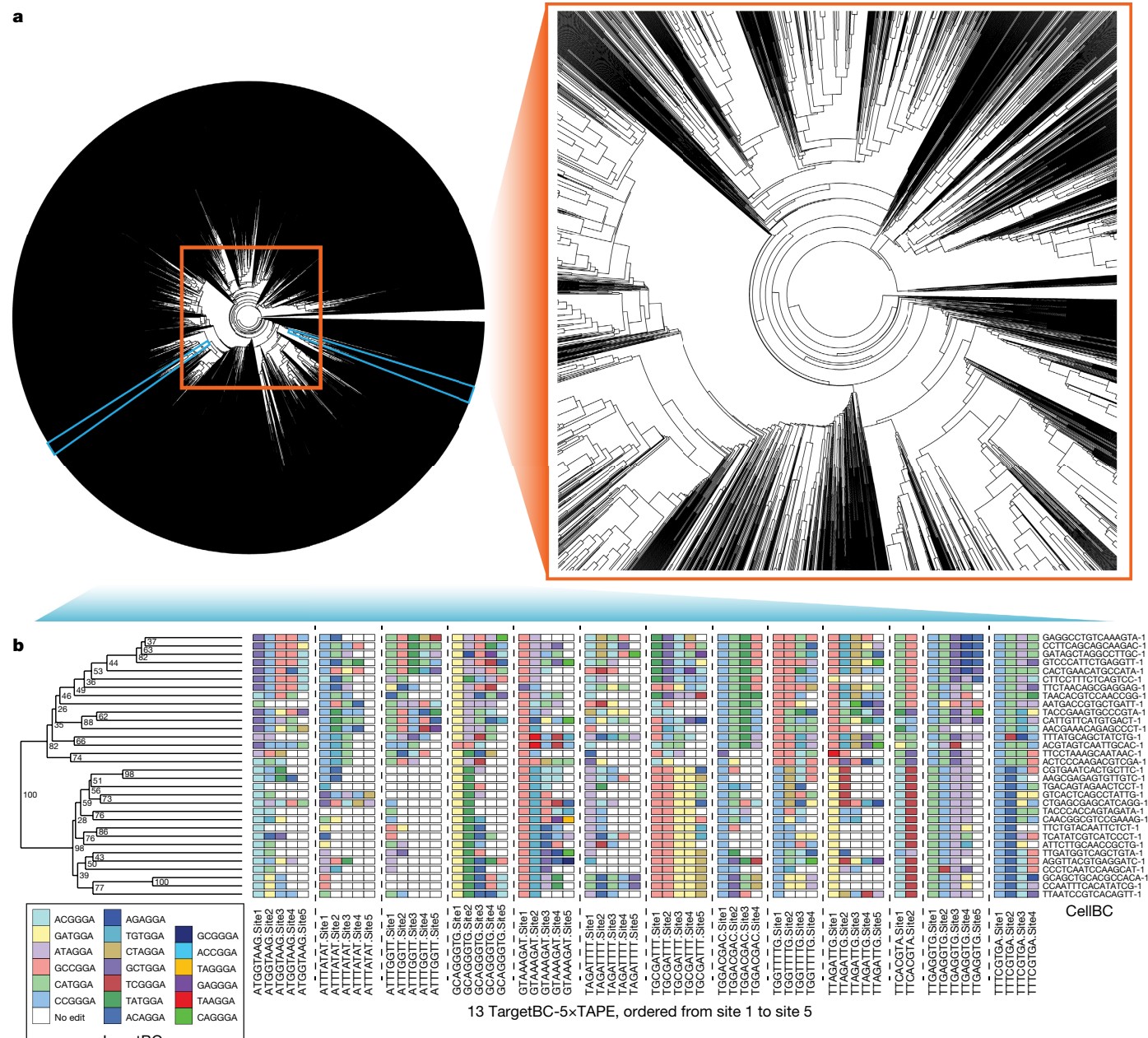

**Fig. 5 | Reconstruction of a monophyletic cell lineage tree using DNA Typewriter. a**, A monophyletic lineage tree of the 3,257 cells with all 13 TargetBC Tape arrays recovered. The UPGMA clustering method was used to construct the tree from a distance matrix that takes into account the order of edits within the TAPE-1 arrays, by discounting matches for which earlier sites along the same DNA Tape were not also identically edited. **b**, A lineage tree constructed by order-aware UPGMA for a subset of 32 cells drawn from the larger tree, specifically the two 16-cell clades marked with light blue in the circular tree. Numbers next to branching points denote bootstrap values out of 100 resamplings. The 59 sites of the 13 TargetBC-associated Tape arrays are represented to the right, with InsertBCs coloured by edit identity. Cells are identified by the 16-bp CellBCs (10x Chromium v3 chemistry) listed on the far right. A higher-resolution version of the entire tree of 3,257 cells in the same format is provided in Supplementary Fig. 1.

barcodes are introduced in each epoch but adjacent epochs always share one barcode (Fig. 2a). Concurrent transfection of two pegRNAs with distinct barcodes is evident in the resulting bigram frequency matrix, specifically by the signal both immediately above and below the diagonal (Fig. 2e). Our aforementioned decoding algorithm performed slightly worse on programme 3, with a single swap between epochs 4 and 5 required to revise the inferred order to the correct order (Extended Data Fig. 4c).

Finally, we asked whether the relative strength of signals could be inferred from symbols recorded to DNA Tape. For this, we designed programmes 4 and 5, which have the same order of barcodes–a pair

in each epoch–but with each pair at different ratios in the two programmes. In programme 4, pegRNAs encoding each pair of barcodes were always mixed at a 1:3 ratio, whereas in programme 5 the same pairs for each epoch were mixed at a 1:1, 1:2, 1:4 or 1:8 ratio (Fig. 2a). For both programmes, the resulting bigram frequency matrix was consistent with expectation and the order of events was accurately inferred (Fig. 2f,g and Extended Data Fig. 4c). However, in addition, we were able to compare the relative ratios at which each pair of barcodes was introduced within each epoch between programmes 4 and 5 and found these to be well correlated with expectation (Fig. 2h and Extended Data Fig. 4e,f). Taken together, these results show that DNA Typewriter

can record, recover and decode complex event histories including the order, overlap and relative strength of signals.

## Recording and recovering short texts

We next designed a strategy to record and decode short text messages in populations of cells with DNA Typewriter. In brief, we modified the Base64 binary-to-text encoding scheme by assigning each of the 64 possible 3-mers to 6-bit binaries. The Base64 scheme encodes uppercase and lowercase English characters, numbers from 0 to 9, and two symbols. In our TAPE64 scheme, we encoded uppercase English characters, four symbols and a whitespace, with two-fold or four-fold redundancy (Fig. 3a and Supplementary Table 2).

We selected three messages to encode: (1) "WHAT HATH GOD WROUGHT?", the first long-distance message transmitted by Morse code in 1844; (2) "MR. WATSON, COME HERE!", the first message transmitted by telephone in 1876; and (3) "BOUND FOREVER, DNA", a translation of a lyric from the 2017 song *DNA* by the K-pop music group BTS. Each message was split into sets of four characters. Plasmids encoding a given set of pegRNAs were concurrently transfected along with a plasmid encoding PE2 into 5×TAPE-1(+) HEK293T cells at a ratio of 7:5:3:1, such that the ratio encoded the order of the four characters within each set (Fig. 3b). As such, each full message could be recorded by five to six consecutive transfections spaced by 3-day intervals.

To recover and decode the recorded messages, we collected populations of cells corresponding to each message and amplified and sequenced the DNA Tape region. From the resulting reads, we first identified all characters in the message by examining NNNGGA insertions at site 1 of 5×TAPE-1. We then grouped these characters into sets by hierarchical clustering (Extended Data Fig. 5a), while also ordering the sets relative to one another by applying the algorithm used for the previous experiment to the bigram transition matrix (Fig. 3c–e). Finally, we arranged the four characters within each set by decreasing the order of their edit score-corrected frequencies, as within each set earlier characters were encoded at a higher plasmid concentration.

For all three messages, our reconstructions of the original text were reasonable but imperfect. From the first message, 17 out of 22 characters were correctly recovered and ordered, with 3 deletion errors and 1 swap between adjacent characters to yield "WA HATH GOD WRUOGT?" (Fig. 3c). Of note, the deletion errors were due to repeated use of pegRNA barcodes ACT, CAT and GAC to encode multiple 'H' or 'T' characters that, as such, were not expected to be recovered separately. These deletion errors are the result of our encoding scheme, which used only 64 unique pegRNAs. We anticipate that greater information content per edit can be achieved with pegRNAs with longer barcodes; for example, 6-bp barcodes would have allowed each instance of repeated characters to be represented by different insertions, thereby avoiding this kind of error. In line with our previous analysis on decoding complex event histories, this inference was robust to sequencing depth, as undersampling did not appreciably add more errors to decoded messages (Extended Data Fig. 5b). From the second message, 20 out of 22 characters were correctly recovered and ordered, with two deletions and one insertion to yield "MR. WATSON, COMI HEE!" (Fig. 3d). From the third message, 16 out of 18 characters were correctly recovered and ordered, with a single swap between adjacent characters to yield "BOUND FOREVE,R DNA" (Fig. 3e). Despite these errors, our experiment demonstrates the potential of DNA Typewriter to digitally record the content and order of information to the genomes of populations of mammalian cells.

## Ordered recording of cell lineage

Beginning with genome editing of synthetic target arrays for lineage tracing (GESTALT), several approaches have been developed that leverage stochastic genome editing to generate a combinatorial diversity of mutations that irreversibly accumulate to a compact DNA barcode during in vivo development[6,30]. Such stochastically evolving barcodes mark cells and enable inference of their lineage relationships on the basis of patterns of shared mutations. However, despite their promise, GESTALT and similar recorders remain sharply limited by several technical challenges, including (1) a failure to explicitly record the order of editing events, which renders phylogenetic reconstruction of cell lineage highly challenging[31,32]; (2) a reliance on DSBs and NHEJ to introduce edits (DSBs frequently delete or corrupt consecutively located targets within a barcode); and (3) the decreasing number of target sites available to CRISPR–Cas9 as sites are irreversibly edited, which effectively makes it impossible to sustain continuous lineage recording over long periods of time without sacrificing resolution.

The ordered manner in which edits accrue with DNA Typewriter, the use of a prime editor with a Cas9 nickase to insert one of many possible symbols at the type guide, the predefined sequences and locations of potential edits, and the fact that one and only one monomer is an active type guide at any given moment have the potential to address all of these limitations at once. To demonstrate this potential, we sought to record cell lineage during the expansion of a monoclonal cell line, leveraging DNA Typewriter in combination with single-cell RNA-seq (scRNA-seq). First, we constructed a HEK293T cell line with doxycycline (Dox)-inducible PE2 expression (iPE2(+) HEK293T). We also designed and cloned a lentiviral construct that includes (1) the 5×TAPE-1 sequence, associated with a random 8-bp barcode region (TargetBC) at its 5' end; (2) a transcription cassette for TargetBC-5×TAPE-1 with a reverse transcription capture sequence for enrichment during scRNA-seq; and (3) a constitutive pegRNA expression cassette that targets TAPE-1 for a 6-bp insertion (NNNGGA, referred to below as InsertBC; GGA is the key sequence for TAPE-1) (Fig. 4a). Lentiviral transduction of this construct into the cell line at a high multiplicity of infection (MOI) was followed by serial dilution to isolate a monoclonal cell line that grew from 1 cell to ~1.2 million cells via ~20 doublings over 25 days in the presence of Dox (Fig. 4b and Extended Data Fig. 6a). After collection, we used scRNA-seq to recover and sequence multiple TargetBC-5×TAPE-1 arrays from each of ~12,000 cells.

The frequency distribution of recurrently observed TargetBCs and InsertBCs in these data suggested that the MOI for this monoclonal cell line was ~19 (Extended Data Fig. 6b,c and Methods). However, the DNA Tapes associated with some TargetBCs were recovered more effectively than others (Extended Data Fig. 6b), presumably owing to site-of-integration effects on expression. To minimize complications related to missing data, we focused our analysis on cells for which we recovered DNA Tape sequences from all of the 13 most frequently observed TargetBCs, excluding one DNA Tape sequence with a corrupted type guide (TargetBC ATAAGCGG). Although the sequencing error rate was estimated to be very low (Extended Data Fig. 6d,e), accumulation of errors across edited sites might affect lineage reconstruction. We therefore also required that all edits to these DNA Tapes be among the 19 most frequently observed InsertBCs.

Applying these filters left 3,257 cells, for each of which we recovered intact TAPE-1 sequences for each of the 13 prioritized TargetBCs. Although nine of these TAPE-1 sequences were the expected five monomers in length, three were four monomers in length (TargetBCs TGGAC-GAC, TTTCGTGA and TGGTTTTG) and one was two monomers in length (TargetBC TTCACGTA). Because of their consistent length across the dataset, we infer that these TargetBC-specific contractions are due to pre-existing heterogeneity in the TAPE-1 lentiviral library before integration, rather than having been caused by editing. Thus, the TAPE-1 arrays on which we focused our analyses included 13 active type guides and 59 editable sites. With 59 editable sites and 19 potential edits per site, the overall complement of assayed DNA Tape in each cell has on the order of $10^{75}$ possible states.

During monoclonal expansion, the generation of lineage barcodes in each cell was efficient, such that the vast majority of the cells assayed contained a unique editing pattern across the 59 sites (3,236/3,257 or

99.4%; 9 patterns recurred in 2 cells and 1 pattern recurred in 3 cells). After 25 days of editing, the first sites of active TAPE-1 arrays were edited to near saturation (mean, 96.8%) while the fifth sites were only modestly edited (mean, 19.7%) (Fig. 4c). Across all 13 DNA Tape arrays, the number of edits accruing per cell resembled a Poisson distribution, with the mean number of discrete events per cell ($\mu = 39.4$) roughly equalling the variance ($\sigma^2 = 40.0$) (Fig. 4d). Assuming 20 cell divisions, this corresponds to an average of two edits accruing per cell division. The mean number of pairwise differences between cells, including sites at which one cell was edited and the other was unedited, was 41.9 ± 5.3 (Fig. 4e).

We next sought to construct a cell lineage tree. In contrast to GESTALT and other CRISPR–Cas9-based lineage recording systems, edits accruing to the multicopy DNA Tape derive from a finite set of pegRNA-specified symbols, analogous to the finite set of nucleotides or amino acids used to build conventional phylogenetic trees. However, in further contrast to GESTALT but also to conventional phylogenetics, DNA Typewriter provides explicit information regarding the order in which differences accrued. To leverage this, we constructed a 3,257 × 3,257 similarity matrix by calculating, for all possible pairs of cells, the number of shared edits across the 59 sites. However, for shared edits at any given site to be counted, we required that all earlier sites along that DNA Tape also be identically edited (Methods). Across all 5.3 million pairwise comparisons of cells, 24 million of 33 million shared edits met this criterion; those that did not presumably correspond to coincident occurrences of the same edit at the same site in different cells and, as such, are appropriate to discount. After converting this similarity matrix to a distance matrix, we generated two phylogenetic trees, using either the unweighted pair group method with arithmetic mean (UPGMA) or the neighbour-joining hierarchical clustering method. When comparing these two methods, UPGMA resulted in a tree with a lower parsimony score of 123,625, compared with the score of 124,997 for the tree constructed using neighbour-joining hierarchical clustering. A compact representation of the UPGMA tree is shown in Fig. 5a, with the full tree in Supplementary Fig. 1.

To assess robustness, we first focused on two distantly related clades of 16 cells from the global UPGMA tree, merged them into a new set of 32 cells and then performed conventional bootstrapping, treating the sites associated with each of the 13 TargetBCs as independent groups, sampling 13 TargetBC groups with replacement, and then constructing and comparing UPGMA-based trees (Methods). Across 100 resamplings, all 31 branchings were observed multiple times, 20 with bootstrap values over 50%, with a bootstrap value of 100% for the separation between the two distantly related clades (Fig. 5b). Bootstrap analysis of an additional clade of 81 cells is shown in Extended Data Fig. 6f; for this clade, all 80 branchings were observed multiple times, 38 with bootstrap values over 50%. Finally, we performed bootstrap analysis of the entire matrix, resulting in a tree in which 76% of branches were seen multiple times and 25% had bootstrap values over 50% (Supplementary Fig. 1).

In summary, over the course of 25 days of expansion of a monoclonal cell line from 1 to ~1.2 million cells, we observed the ordered accumulation of 39.4 ± 6.3 edits to 59 sites located within 13 DNA Tape arrays. Although the number of active type guides at these arrays declined (from 13 in the founding cell to a mean of 8.6 active type guides per cell after 25 days), we did not exhaust the recording capacity of the system (only 1 of the 3,257 sampled cells was edited at all 59 sites). To further assess whether editing was maintained throughout the experiment, we examined the number of pairwise differences between each cell and its nearest neighbour within the sampled set of 3,257 cells (Fig. 4f). On average, cells were separated from their nearest neighbour by 22.8 edits (or, assuming a constant rate of ~2 edits per generation, 11 to 12 generations). We interpret this as strong support for the conclusion that editing of the DNA Tapes was maintained throughout clonal expansion.

## Editing and recovering longer DNA Tape

As illustrated by this lineage tracing experiment, we can deploy and recover at least a dozen DNA Tapes in each cell, which substantially increases information capacity. However, even with multiple DNA Tapes, the maximum potential recording duration of each DNA Tape remains directly proportional to the number of consecutive monomers on each DNA Tape. Although 5×TAPE-1 appears to be very stable within cells as well as throughout amplification and sequencing (Extended Data Fig. 4b), longer tandem arrays might introduce additional technical challenges, for example, by being difficult to synthesize, clone and maintain, prone to instability during in vivo DNA replication or repair as well as during in vitro PCR, and difficult to accurately and fully sequence.

To evaluate the extent to which such issues might be limiting in practice, we sought to generate a synthetic minisatellite in the form of 12 or 20 repeats of the 14-bp TAPE-1 monomer. 12×TAPE-1 was synthesized as single-stranded DNA (IDT), and 20×TAPE-1 was synthesized as a plasmid (GenScript). PCR amplicons of each array were cloned into the piggy-Bac vector through Gibson assembly. Of note, cloned constructs were used 'as is', even though it is possible that some degree of variation in repeat number was already present (Extended Data Fig. 7a,b). We integrated piggyBac vectors bearing ~12×TAPE-1 or ~20×TAPE-1 into HEK293T cells expressing both PE2 and pegRNAs targeting TAPE-1 for NNNGGA insertions (PE2(+) 3N-TAPE-1-pegRNA(+) HEK293T) in triplicate. We cultured these cell lines for 40 days before collecting genomic DNA. PCR amplification of TAPE-1 was followed by standard library construction and sequencing on the Pacific Biosciences Sequel platform to obtain circular consensus sequencing (CCS) reads. On average, we recovered 8.4 ± 3.3 repeats of TAPE-1 monomers from 12×TAPE-1 and 12.5 ± 4.3 repeats from 20×TAPE-1. In each case, there was a sharp drop-off after the intended length of 12 or 20 monomers, suggesting that, regardless of the mechanism, these longer arrays are more prone to contraction than expansion (Extended Data Fig. 7c). Of note, the editing rates were the same for the constructs (4.5 ± 1.3 edits and 4.5 ± 1.5 edits for the 12×TAPE-1 and 20×TAPE-1 arrays, respectively; Extended Data Fig. 7d). This is expected, as each DNA Tape has exactly one active type guide and, as such, the rates at which they are written to should be independent of their length.

We grouped CCS reads within each replicate on the basis of a degenerate 8-bp barcode (TargetBC), as these presumably derived from the same integration. On average, each TargetBC group had 3.1 ± 3.4 and 3.8 ± 5.7 reads for ~12×TAPE-1 and ~20×TAPE-1, respectively. Within TargetBC groups, shorter arrays appeared more stable, with a greater proportion matching the maximum length within that group (Extended Data Fig. 7e,f). Of representative CCS reads for 4,784 and 6,254 integrated arrays for 12×TAPE-1 and 20×TAPE-1, respectively, the overwhelming majority (>99.5%) exhibited clear patterns of sequential, directed editing (Extended Data Fig. 7g,h). In terms of the maximum extent to which any given DNA Tape was edited, we observed one TargetBC for which 14 distinct 3-bp insertion events were recorded along a 14-monomer DNA Tape.

This experiment illustrates that it is possible to construct and use synthetic minisatellites corresponding to at least 20 monomers as a DNA Tape and that sequential recording of at least 14 consecutive events with DNA Typewriter is possible. Nonetheless, further experiments are required to quantify the extent to which variation in synthetic minisatellite length is due to (1) piggyBac vector heterogeneity, that is, variation that existed before integration; (2) DNA replication and microsatellite instability in HEK293T cells; (3) DNA repair subsequent to prime editing-induced nicks; and/or (4) PCR amplification artefacts. Of note, the observed variation in array length tended to occur within the unedited portion of the DNA Tape (Extended Data Fig. 7g,h). We have yet to observe any clear examples of 'information erasure', possibly because the edits themselves disrupt the tandem repeats, inhibiting processes that might otherwise lead to erasure from spreading proximal to the type guide.

## Discussion

Digital systems represent information through both the content and order of discrete symbols, with each symbol drawn from a finite set. Digital systems are ancient and include written text, Morse code and binary data as well as, of course, genomic DNA. In this proof-of-concept study of DNA Typewriter, we demonstrate how sequential genome editing of a monomeric array constitutes an artificial digital system that is operational within living eukaryotic cells, capable of 'writing' thousands of discrete symbols to DNA in an ordered fashion.

DNA Typewriter improves on existing CRISPR-based molecular recorders in important ways (Supplementary Table 3). The sequential editing achieved by DNA Typewriter resembles Cas1–Cas2-based recording[10,11,16], which at present is limited to bacterial systems. In DOMINO[19] and CAMERA[14], base editors are used to record biological signals to 'preprogrammed logic circuits' composed of multiple targets for base editing. Although these methods are conceptual predecessors to DNA Typewriter, there are critical differences. In particular, with all three methods, a recording event creates a new target for further editing (that is, the type guide). However, with DOMINO and CAMERA, each logic circuit is designed to record a specific order. By contrast, a single DNA Typewriter construct can potentially record any order. For example, to distinguish pairwise orderings within a set of $n$ events, DOMINO or CAMERA would require $n$-choose-2 recording logic circuits or a system that contains on the order of $n^2$ unique gRNAs and their targets. By contrast and as demonstrated here (Fig. 2), DNA Typewriter requires only a single target array such as 5×TAPE-1, along with $n$ unique pegRNAs that encode different insertions but share the same target.

How might we write biologically specific information using DNA Typewriter? Here we use pegRNAs to encode symbols (that is, insertional barcodes), but these pegRNAs are introduced by artificial transduction or stochastic expression. However, several groups have engineered gRNAs whose activity is dependent on the binding of specific small molecules or ligands[33–35]. Also, we recently developed ENGRAM, a prime editing-based system in which biological signals of interest such as NF-κB and Wnt signals are coupled to the production of specific pegRNAs[36]. These pegRNAs mediate the insertion of signal-specific barcodes to a DNA-based recording site, providing quantitative information with respect to the strength and/or duration of the signal(s). At least in principle, such strategies are compatible with the current implementation of DNA Typewriter, potentially enabling the temporal dynamics of multiple biological signals or other cellular events to be recorded and resolved. In this context, the use of longer and therefore more diverse insertion barcodes could enable extensive multiplexing, although this might come at the expense of recording efficiency. A further caution is that we estimate the rate of prime editing to be on the order of days, such that DNA Typewriter may be most useful for recording information about biological processes that unfold over a timescale of days or weeks, rather than minutes or hours.

One such process is biological development, wherein the unfolding of a cell lineage tree is of fundamental interest. In a proof-of-concept experiment, we show how DNA Typewriter overcomes the major limitations of earlier editing-based lineage recorders such as GESTALT[6,30] by reducing ambiguity about the order in which editing events occurred, eschewing DSBs and thereby minimizing the risk of inter-target deletion, predefining the locations to which edits accrue, predefining the 'symbol set' from which edits are drawn and stabilizing the rate of editing by ensuring one and only one type guide per active DNA Tape. These attributes clearly paid off in our proof-of-concept experiment, as we were able to sustain a seemingly steady accumulation of edits to multicopy DNA Tape across 25 days of in vitro expansion, from a single cell to over 1 million cells. Although this is longer than the gestation period of a mouse, we do not exhaust the recording capacity of the system. Furthermore, the resulting data are sufficiently rich and complete that we can build and characterize cell lineage trees from these data with conventional phylogenetic algorithms (for example, UPGMA and NJ), with only minor modifications directed at leveraging information about the order of edits, not available in other contexts in which phylogenetics is applied. In this experiment, the number of edits accruing per cell resembled a Poisson distribution. Further experiments are needed to assess the extent to which this rate of accrual is a function of absolute time, the cell cycle or some combination thereof. However, as it has been shown that prime editing continues to take place in non-mitotic cells such as neurons[2], we suspect that it is primarily a function of time.

What are the limits of this approach? Under the assumption that we can achieve similar performance in vivo (multiple efficiently recovered DNA Tapes per cell; steady accrual of edits over several weeks; multiple edits per lineage per cell division), we can readily conceive of a technical path to Sulstonesque reconstructions[37] of the cell lineage histories of non-transparent model organisms, for example, flies, mice, zebrafish and macaques. We further envision that a single synthetic DNA construct that encodes a prime editing enzyme, multiple recording arrays and a combination of stochastic and signal-specific pegRNAs could be used to simultaneously record both lineage and biological signals in any multicellular system, that is, a molecular 'flight recorder' locus. A single-locus design would be less affected by site-of-integration effects, such as we have observed with multiple DNA Tape constructs integrated across the genome. Alternatively, if genomic sites with a high prime editing efficiency can be identified, such sites might be leveraged to boost information capture. A separate risk is that prime editing efficiency might vary substantially across cell types. However, any such variation could potentially be ameliorated by technical improvements to system components[22,23], by increasing recording capacity and/or by modelling it during tree reconstruction. Although challenging to engineer, a generic recorder locus would allow us to take full advantage of DNA as an in vivo digital recording medium, for example, not only to characterize wild-type development, but also to enable systematic comparison of the developmental histories of wild-type and mutant individuals.

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

## Methods

### Plasmid cloning

Both pegRNA and DNA Tape constructs were cloned using either Gibson assembly (Gibson Assembly Master Mix, New England Biolabs) or ligation after restriction (T4 DNA ligase, New England Biolabs). For the Gibson assembly protocol, inserts of interest, usually ordered in the form of single-stranded DNA (IDT; Ultramer, up to 200 bp, or oPool, up to 350 bp), were amplified using PCR (KAPA HiFi polymerase) and converted into double-stranded DNA molecules. For ligation, single-stranded DNAs (IDT) were annealed with 4-bp overhangs on both ends of the double-stranded DNAs, with these overhangs acting as a substrate for T4 DNA ligase. Cloning backbones were digested with either BsaI-HFv2 or BsmBI-v2 (NEB), gel purified and mixed with inserts in the Gibson assembly reaction. A small amount (1–2 μl) of Gibson assembly reaction mix or T4 ligation mix was added to an NEB Stbl cell (C3040) for transformation with cells grown at 30 °C or 37 °C for plasmid DNA preparation (Qiagen Miniprep). The resulting plasmids were sequence verified using Sanger sequencing (Genewiz). The pegRNA plasmids used in transient transfection experiments were cloned using plasmid backbone pU6-pegRNA-GG-acceptor (Addgene, 132777), following the protocol outlined in ref. [2]. The resulting pegRNA expression cassette had a U6 promoter and poly(T) terminator. For epegRNA cloning, another fragment including the evoPreQ1 sequence was added, with each strand of oligonucleotides purchased phosphorylated from IDT. The Lenti-TargetBC-5×TAPE-1-pegRNA-InsertBC construct was cloned on the basis of the CROP-seq vector[39] (CROP-seq-guide-Puro; Addgene, 86708). The vector was modified to include a GFP-TargetBC-5×TAPE-1-CaptureSequence1 sequence, and the sequence downstream of the U6 promoter had been modified to allow insertion of the InsertBC-pegRNA sequence. Plasmids encoding DNA Typewriter constructs (piggyBac-5×TAPE-1-BlastR), lineage tracing constructs (Lenti-TargetBC-5×TAPE-1-pegRNA-InsertBC) and pegRNAs (pU6-CApegTAPE1) have been submitted to Addgene (accessions 175808, 183790 and 175809).

### Tissue culture, transfection, lentiviral transduction and transgene integration

The HEK293T cell line was purchased from the American Type Culture Collection and maintained by following the recommended protocol from the vendor. Primary MEFs were purchased from Millipore-Sigma (PMEF-CFL; EmbryoMax Primary Mouse Embryonic Fibroblasts, strain CF1, not treated, passage 3). Both HEK293T and MEF cells were cultured in DMEM with high glucose (Gibco), supplemented with 10% FBS (Rocky Mountain Biologicals) and 1% penicillin-streptomycin (Gibco). mEScells(E14tg2a) were a gift from C. Schröter. mEScells were cultured in Ndiff 227 medium (Takara) supplemented with 1% penicillin-streptomycin, 3 μM CHIR99021 (Millipore-Sigma), 1 μM STEMGENT PD0325901 (Reprocell) and 1,000 units of ESGRO recombinant mouse LIF protein (Sigma-Aldrich). For culturing of both MEFs and mEScells, wells in the culture plates were coated with 0.1% gelatin in a 37 °C incubator for 1 h. Cells were grown with 5% $CO_2$ at 37 °C. Cell lines were used as received without authentication or a test for mycoplasma.

For transient transfection, HEK293T cells were cultured to 70–90% confluency in a 24-well plate. For prime editing, 375 ng of Prime Editor-2 enzyme plasmid (Addgene, 132776) and 125 ng of pegRNA plasmid were mixed and prepared with transfection reagent (Lipofectamine 3000) following the recommended protocol from the vendor. Cells were cultured for 4 to 5 days after the initial transfection unless noted otherwise, and genomic DNA was collected following cell lysis and the protease protocol from ref. [2].

Both MEFs and mEScells were transfected using 4D-Nucleofector (Lonza Bioscience). For MEFs, about 200,000 cells were resuspended in 20 μl Nucleofector buffer with supplement, mixed with 800 ng of DNA plasmids (600 ng of pCMV-PEmax-P2A-hMLH1dn and 200 ng of epegRNA plasmid), loaded onto a 16-well strip cuvette and electroporated using programme CM137 in the 4D-Nucleofector. For mEScells, about 50,000 cells were resuspended in 20 μl Nucleofector buffer with supplement, mixed with 800 ng of DNA plasmids (600 ng of pCMV-PEmax-P2A-hMLH1dn and 200 ng of epegRNA plasmid), loaded onto a 16-well strip cuvette and electroporated using programme CG104 in the 4D-Nucleofector. Cells were cultured for four more days before genomic DNA collection or the subsequent transfection in the case of mEScells.

For lentivirus generation, approximately 300,000 HEK293T cells were seeded in each well of a six-well plate and cultured to 70–90% confluency. The lentiviral plasmid was transfected into cells along with the ViraPower lentiviral expression system (Thermo Fisher), following the recommended protocol from the vendor. Lentivirus was collected following the same protocol, concentrated overnight using Peg-it Virus Precipitation Solution (SBI) and used within 1–2 days to transduce HEK293T cells without a freeze–thaw cycle. To achieve high MOI, we used the MagnetoFection protocol (OZ Bioscience). For the lineage tracing experiments, transduced cells were serially diluted and seeded in 96-well plates to identify monoclonal lines. Dox concentrations were maintained by including 10 mg l$^{-1}$ Dox in the initial culture and replenishing it every 5 days, to account for the 24- to 48-hour half-life of Dox in culture medium.

For transposase integration, 500 ng of cargo plasmid and 100 ng of Super piggyBac transposase expression vector (SBI) were mixed and prepared with transfection reagent (Lipofectamine 3000) following the recommended protocol from the vendor and then transfected into confluent 24-well plates. A monoclonal cell line with Dox-inducible expression of PE2 was generated by integrating the coding sequence for PE2 using the piggyBac transposase system and selecting clones by prime editing activity, as previously described[27].

### Genomic DNA collection and sequencing library preparation

The targeted region from collected genomic DNA was amplified using two-step PCR and sequenced using an Illumina sequencing platform (NextSeq or MiSeq). The first PCR (KAPA Robust polymerase) included 1.5 μl of cell lysate and 0.04 to 0.4 μM of forward and reverse primers in a final reaction volume of 25 μl. In the first PCR, samples were incubated for 3 min at 95 °C; 15 s at 95 °C, 10 s at 65 °C and 90 s at 72 °C for 25–28 cycles; and 1 min at 72 °C. Primers included sequencing adaptors at their 3′ ends, appending them to both termini of the PCR products amplified from genomic DNA. After the first PCR step, products were assessed on a 6% TBE gel, purified using 1.0× AMPure beads (Beckman Coulter) and added to the second PCR that appended dual sample index sequences and flow cell adaptors. The second PCR programme was identical to the first except that we ran it for only 5–10 cycles. Products were again purified using AMPure beads and assessed on a TapeStation (Agilent) before being denatured for the sequencing run.

To append 10-bp unique molecular identifiers (UMIs), we performed PCR in three steps: first, genomic DNA was linearly amplified in the presence of 0.04 to 0.4 μM of a single forward primer in two PCR cycles using KAPA Robust polymerase. Specifically, we programmed the UMI-appending linear PCR to incubate samples for 3 min and 15 s at 95 °C; 1 min at 65 °C followed by 2 min at 72 °C for 5 cycles; 15 s at 95 °C; and 1 min at 65 °C followed by 2 min at 72 °C for 5 cycles. Second, this reaction was cleaned up using 1.5× AMPure beads, followed by a second PCR with forward and reverse primers: 3 min at 95 °C; 15 s at 95 °C, 10 s at 65 °C and 90 s at 72 °C for 25–28 cycles; and 1 min at 72 °C. In this PCR, the forward primer bound upstream of the UMI sequence and was not specific to the genomic locus. Finally, after PCR amplification, products were cleaned up using AMPure magnetic beads (1.0×, following the protocol from Beckman Coulter) and added to the third and last PCR that appended dual sample index sequences and flow cell adaptors. The run parameters for the third PCR were the same as for the second PCR except that only 5–10 cycles were used. TAPE construct sequences and PCR primer sequences are provided in Supplementary Tables 4 and 5, respectively.

For long-read amplicon sequencing library preparation, we used a one-step PCR protocol: the first PCR (KAPA Robust polymerase) included 1.5 µl of cell lysate and 0.04 to 0.4 µM of forward and reverse primers with Pacific Bioscience sample index sequences in a final reaction volume of 25 µl. We programmed the first PCR to incubate samples for 3 min at 95 °C; 15 s at 95 °C, 10 s at 65 °C and 3 min at 72 °C for 25–28 cycles; and 1 min at 72 °C. After the first PCR step, products were purified using 0.6× AMPure beads (Beckman Coulter), assessed on a TapeStation (Agilent) and sequenced on the Sequel platform (Pacific Biosciences; Laboratory of Biotechnology and Bioanalysis, Washington State University).

### Genomic DNA amplicon sequencing data processing and analysis

Sequencing reads from the Illumina MiSeq and NextSeq platforms were first demultiplexed using BCL2fastq software (Illumina). For the experiments shown in Fig. 1 (and Extended Data Figs. 1–3), sequencing libraries were single-end sequenced to cover the DNA Tape from one direction. For the experiments shown in Figs. 2 and 3 (and Extended Data Figs. 4 and 5), sequencing libraries were paired-end sequenced to cover the entire array from both directions. Paired reads were then merged using PEAR[40] with default parameters to reduce sequencing errors. Insertion sequences, in the form of NNGGA (5-mer) to NNNNNNGGA (9-mer), were extracted from sequencing reads of the TAPE arrays, including 2×TAPE-1, 3×TAPE-1 and 5×TAPE-1, using pattern-matching software such as Regular Expression (package REGEX) in Python. Insertions (4–6 bp) in 3×TAPE-1 to 3×TAPE-48 were also extracted using REGEX pattern-matching software.

For the sequential transfection epoch experiment shown in Fig. 2, we first extracted 5-mer insertions from 5×TAPE-1 sequencing reads and used a $k$-means clustering algorithm to filter out possible PCR and sequencing errors with low read counts. Such filtering removed all reads that had the wrong key sequence (GGA in the case of TAPE-1), leaving a set of 16 possible 5-mer sequences in the form of NNGGA. Across five repeats of insertion sites in 5×TAPE-1, we calculated the separate unigram frequencies for each site, which were used to build the unigram order as shown in Extended Data Fig. 4c. Bigram frequencies for adjacent insertion sites (site 1 and site 2, site 2 and site 3, site 3 and site 4, and site 4 and site 5) were combined, normalized across the row and column, and used to build the bigram transition matrices shown in Fig. 2c–g. For ordering of barcodes according to their transfection history, we first generated a unigram order by sorting relative frequency at site 1, with barcodes assumed to have been transfected earlier if they appeared more frequently in site 1 than in the other sites. Using the resulting unigram order as the initial order, we implemented an iterative algorithm where we passed through the order from early to late, swapped the order if a bigram frequency was inconsistent with the order and restarted the pass unless there had been no swaps in a single pass.

For the short digital text encoding experiment shown in Fig. 3, we extracted 6-mer insertions, corrected the read counts of each 6-mer by editing efficiency (using separately measured insertion frequency and respective plasmid abundance, similarly to the process described in Extended Data Fig. 1d,f), used a $k$-means clustering algorithm to identify NNNGGA barcodes and built a bigram transition matrix as described in the paragraph above. We first analysed the bigram transition matrices using a hierarchical clustering algorithm with default parameters in R software (using a Euclidean distance measure and the complete linkage clustering method, as described in Extended Data Fig. 5). Putative sets of barcodes (cotransfection sets with generally 2–4 barcodes) were visually identified on the basis of the dendrogram and used to group barcodes in the output bigram order of the algorithm used above. The order within the cotransfection sets was determined using corrected unigram counts combined across all five sites, where more abundant barcodes were assigned to be earlier within the set. Barcodes were mapped back to the text following the encoding table (Supplementary Table 2).

For the long-read sequencing experiment described in Extended Data Fig. 7, 12×TAPE-1 and 20×TAPE-1 sequences were isolated from Pacific Biosciences CCS reads. The number of TAPE monomers and insertions was calculated using sequential text matching around insertions and the expected length of the array based on insertion counts. Reads without a match between expected length and observed length were filtered out. Each 12×TAPE-1 and 20×TAPE-1 construct is associated with an 8-bp degenerate barcode sequence (TargetBC). Assuming that the integration sites for each TargetBC were different, we grouped reads from any given replicate that had the same TargetBC. On the basis of our observation that array collapse is more frequent than array expansion, we selected the read with the maximum number of TAPE monomers from each set of reads that shared a TargetBC. If multiple reads were in a tie by this criterion, we selected the one (or one of the ones) with the most edits for presentation in Extended Data Fig. 7g,h. For presentation in Extended Data Fig. 7c–h, we selected reads that had at least three insertions and at most 12 or 20 TAPE-1 monomers (Extended Data Fig. 7c–f) or at most 25 TAPE-1 monomers (Extended Data Fig. 7g,h).

### Single-cell lineage tracing experiment and analysis

Monoclonal HEK293T cells containing 5×TAPE-1, iPE2 and multiple TargetBC-5×TAPE-1-pegRNA constructs were cultured for 25 days in the presence of 10 mg l$^{-1}$ Dox. Dox was replenished every 5 days, to account for the 24- to 48-hour half-life of Dox in culture medium. The initial culture in a 96-well plate was moved to a 24-well plate and subsequently to a 6-well plate, when the culture was 80–90% confluent. Once the monoclonal cell line reached confluence in the six-well plate (estimated to be 1.2 million cells), cells were frozen and thawed for a single-cell experiment in the absence of Dox. For preparation of cells for the single-cell experiment, cells were dissociated, pelleted by centrifugation at 200 RCF for 5 min and resuspended in a single-cell suspension in 0.04% BSA (NEB) in 1× PBS at a concentration of 1,000 cells per µl following the Cell Preparation Guide from 10x Genomics (manual part no. CG00053 Rev C). Cell numbers and the single-cell suspension were checked using both a manual haemocytometer and a Countess II FL Cell Counter (Thermo Fisher).

Single-cell suspensions of cells were directly used in the 10x Genomics experimental protocol (Chromium Next GEM Single-Cell 3′ Reagent Kit v3.1 with Feature Barcoding Technology for CRISPR Screening; manual part no. CG000205 Rev D). We strictly followed the protocol with recovery of 20,000 targeted cells (10,000 per reaction) until step 2.3. The protocol is written for the CRISPR Screening library, where Feature Barcode components including CRISPR gRNA sequences would be collected in step 2.3B, owing to its smaller size compared with the 3′ Gene Expression library (collected in step 2.3A). In our case, we expected our Feature Barcode components including TargetBC-5×TAPE-1 constructs tagged with 16-nucleotide 10x single-cell barcodes (CBC) and 12-bp UMIs from reverse transcription to be greater than 1 kb in length and therefore collected along with the 3′ Gene Expression library. Nonetheless, we collected both components (the eluates from steps 2.3A and 2.3B) and detected TargetBC-5×TAPE-1 constructs in both using quantitative PCR. Detection of TargetBC-5×TAPE-1 constructs from step 2.3B was unexpected but could have resulted from non-processive reverse transcription that generated shorter cDNA products. We combined the TargetBC-5×TAPE-1 constructs and used paired-end sequencing to obtain CBC, UMI and TargetBC-5×TAPE-1 sequences for each read, along with the 3′ Gene Expression library.

For the initial analysis, we used the CellRanger pipeline from 10x Genomics, which filtered out single-cell barcodes (CBC) and UMIs, recovering about 12,000 cells. We selected reads that contained approved CBC and UMI sequences and extracted TargetBC-5×TAPE-1 sequences from the CellRanger output BAM file. Reads with different UMIs were collapsed on the basis of shared CBC-TargetBC-5×TAPE-1

sequences, and any CBC-TargetBC-5×TAPE-1 reads that had fewer than two UMI sequences associated with them were removed. In cases where we observed the same CBC-TargetBC pairs but with different 5×TAPE-1 sequences, we took the consensus sequence with a larger number of associated UMIs.

For the monoclonal lineage tracing experiment, we corrected the observed TargetBC if it contained a single-nucleotide mismatch with respect to the approved list of the 19 most frequent 8-bp sequences. If the TargetBC differed from the list of sequences by more than 2 nucleotides, we removed the corresponding reads from further analysis. For detection of the 14-bp TAPE-1 sequence, a single-base-pair mismatch or substitution error was corrected to the TAPE-1 sequence. We also filtered out TargetBC-5×TAPE-1 arrays that included InsertBCs other than the top 19 most frequent ones. This resulted in a table where each row contained a CBC, TargetBC and up to five InsertBCs (unedited positions left blank) (Supplementary Data).

For lineage tree reconstruction, only cells (CBC) that included the top 13 most frequent TargetBCs were selected (3,257 cells). This 'top 13' list excluded the corrupted ATAAGCGG TargetBC (where the second TAPE-1 monomer appeared to have been contracted by 6 bp, inactivating the type guide). We calculated a $3,257 \times 3,257$ distance matrix by counting the number of shared InsertBCs across $13 \times 5 = 65$ sites, only counting them if they had the same InsertBC at previous sites (five possible sites per TargetBC; unedited sites were excluded), and then subtracting the count from the maximum number of shared InsertBCs (59, excluding 6 missing sites from three 4×TAPE-1 arrays and one 2×TAPE-1 array) to calculate the distance between a pair of cells. The resulting distance matrix was used as an argument in the UPGMA and neighbour-joining clustering functions in the R phangorn package[41]. Tree visualizations, bootstrapping analysis and parsimony analysis were performed using the R ape package[42] and included functions. Bootstrap resampling was done on blocks of sites within the same TargetBC-TAPE-1 array (that is, resampling with replacement of the intact TAPE-1 arrays associated with the 13 TargetBCs). We then used the same function to calculate the distance matrix as described above, counting InsertBCs as shared only if they had the same InsertBC at previous sites within the TargetBC-TAPE-1 array.

## Reporting summary

Further information on research design is available in the Nature Research Reporting Summary linked to this paper.

## Data availability

Raw sequencing data have been uploaded to the Sequence Read Archive with associated BioProject ID PRJNA757179.

## Code availability

Custom analysis code for this project is available at GitHub (https://github.com/shendurelab/DNATickerTape) and Figshare (https://doi.org/10.6084/m9.figshare.19607811).

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

**Acknowledgements** We thank the members of the Shendure laboratory, especially S. Srivatsan and J.-B. Lalanne, for helpful discussions and approaches for data analysis. We thank L. Starita for critical reading of the manuscript. We thank the members of the Allen Discovery Center for Cell Lineage Tracing, particularly A. Schier and M. Elowitz, for helpful discussions over several years. We thank D. Liu's laboratory for sharing the prime editing plasmids. This work was supported by a grant from the Paul G. Allen Frontiers Group (Allen Discovery Center for Cell Lineage Tracing to J.S.), the National Human Genome Research Institute (UM1HG011586 to J.S.; R01HG010632 to J.S.) and the Brotman Baty Institute for Precision Medicine. J.C. is a Howard Hughes Medical Institute Fellow of the Damon Runyon Cancer Research Foundation (DRG-2403-20). J.S. is an Investigator of the Howard Hughes Medical Institute.

**Author contributions** This project was initiated during discussions among J.C., W.C. and J.S. J.C. and J.S. designed experiments with input from W.C. and A.M. throughout the study. J.C., W.C., A.M., F.M.C., C.C.S., S.G.R. and S.D. contributed to reagent and material preparation. J.C. performed experiments with assistance from W.C., F.M.C., S.G.R., N.H., C.L., B.M. and R.M.D. Single-cell assays were performed by J.C. and S.G.R. with input from A.M. and F.M.C. J.C. and J.S. analysed the data with assistance from W.C., A.M., F.M.C. and S.G.R. J.C. and J.S. wrote the manuscript with input from all other authors.

**Competing interests** The University of Washington has filed a patent application partially based on this work in which J.C., W.C. and J.S. are listed as inventors. The remaining authors declare no competing interests.

**Additional information**
**Correspondence and requests for materials** should be addressed to Junhong Choi or Jay Shendure.

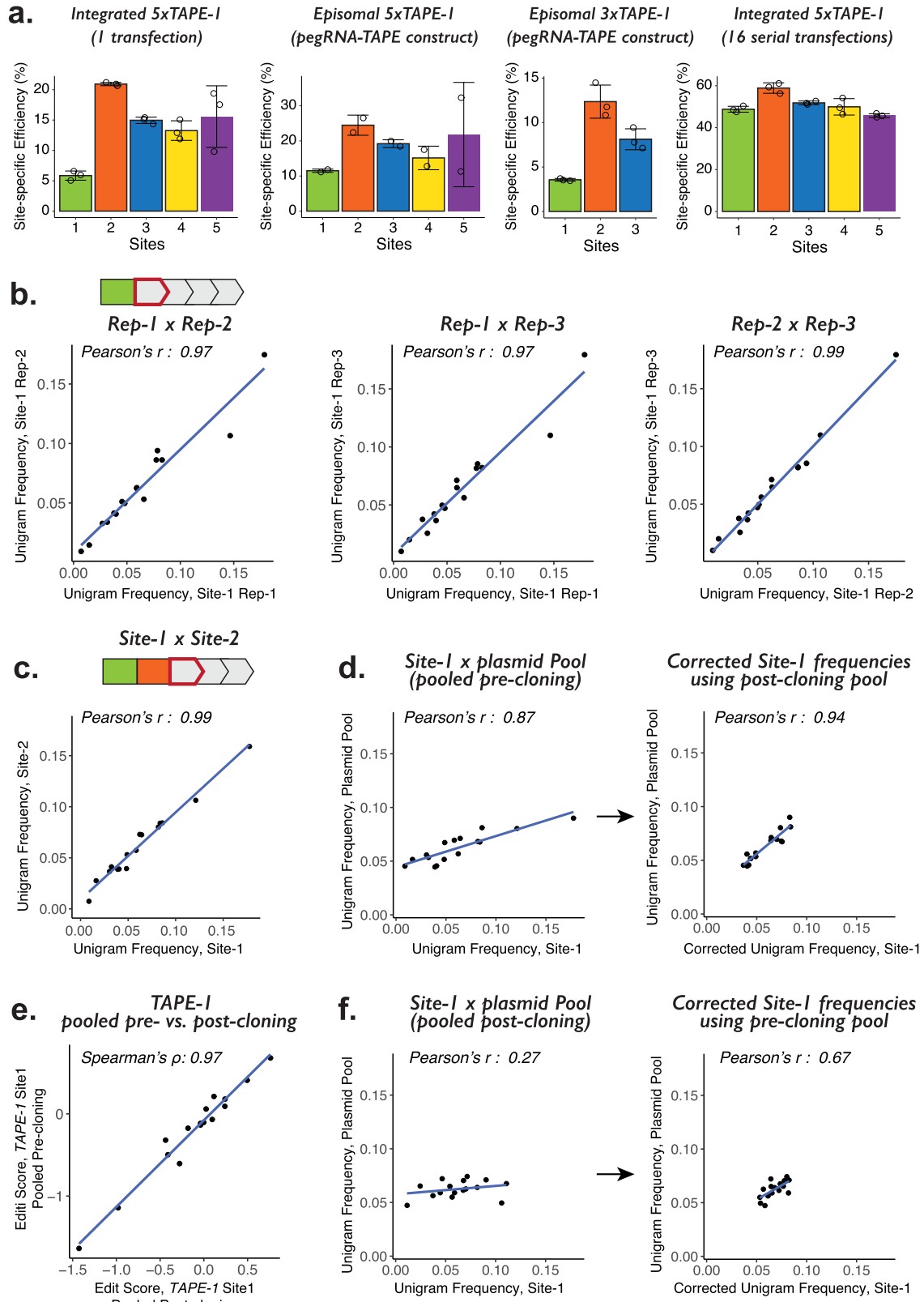

**Extended Data Fig. 1** | See next page for caption.

**Extended Data Fig. 1 | The relative insertional frequencies of k-mers to DNA Tape are determined by relative pegRNA abundances as well as by insertion-dependent sequence bias. a**. Conditional, site-specific editing efficiencies across 3 sites within the 3xTAPE-1 or 5 sites within the 5xTAPE-1, calculated as the number of reads that contain an edit in the indicated site over the total number of reads that contain an edit in the immediately preceding site, which activates the indicated site as a target for editing. The number of all 5xTAPE-1 (or 3xTAPE-1) reads were used for calculating the site-specific editing efficiency for the Site-1, which is activated by its own key sequence. The center and error bars are mean and standard deviations, respectively, from n = 2 transfection replicates for the second plot from the left and n = 3 transfection replicates for the other 3 plots. **b**. Pairwise scatterplots of unigram frequencies of NNGGA insertions at the initiating monomer of 5xTAPE-1 among three transfection replicates. **c**. Scatterplot of unigram frequencies, averaged across three transfection replicates, at the initiating vs. second monomer of 5xTAPE-1. **d**. Scatterplot of averaged unigram frequencies at the initiating monomer in "pre-cloning pooling" experiment vs. the abundances of NNGGA pegRNA-expressing plasmids (left). Insertional bias was corrected for with data from a separate experiment using NNGGA pegRNA-expressing plasmids that were pooled post-cloning, resulting in a better correlation with the abundances of pegRNAs in the plasmid pool (right). Corrections were done by dividing pre-cloning unigram frequencies by post-cloning unigram frequencies at the initiating monomer and multiplying by post-cloning pegRNA plasmid frequencies. **e**. Scatterplot of NNGGA edit scores calculated on the initiating monomer of the 5xTAPE-1 target edited by pegRNA-expressing plasmids pooled pre-cloning vs. post-cloning. Edit scores for each insertion are calculated as log2 of the ratio between insertion frequencies and the abundances of pegRNAs in the plasmid pool. Spearman's $p$ was used instead of Pearson's $r$. **f**. Scatterplot of averaged unigram frequencies at the initiating monomer in "post-cloning pooling" experiment vs. the abundances of NNGGA pegRNA-expressing plasmids (left). Correcting for insertional bias with pre-cloning unigram frequencies improves the correlation (right).

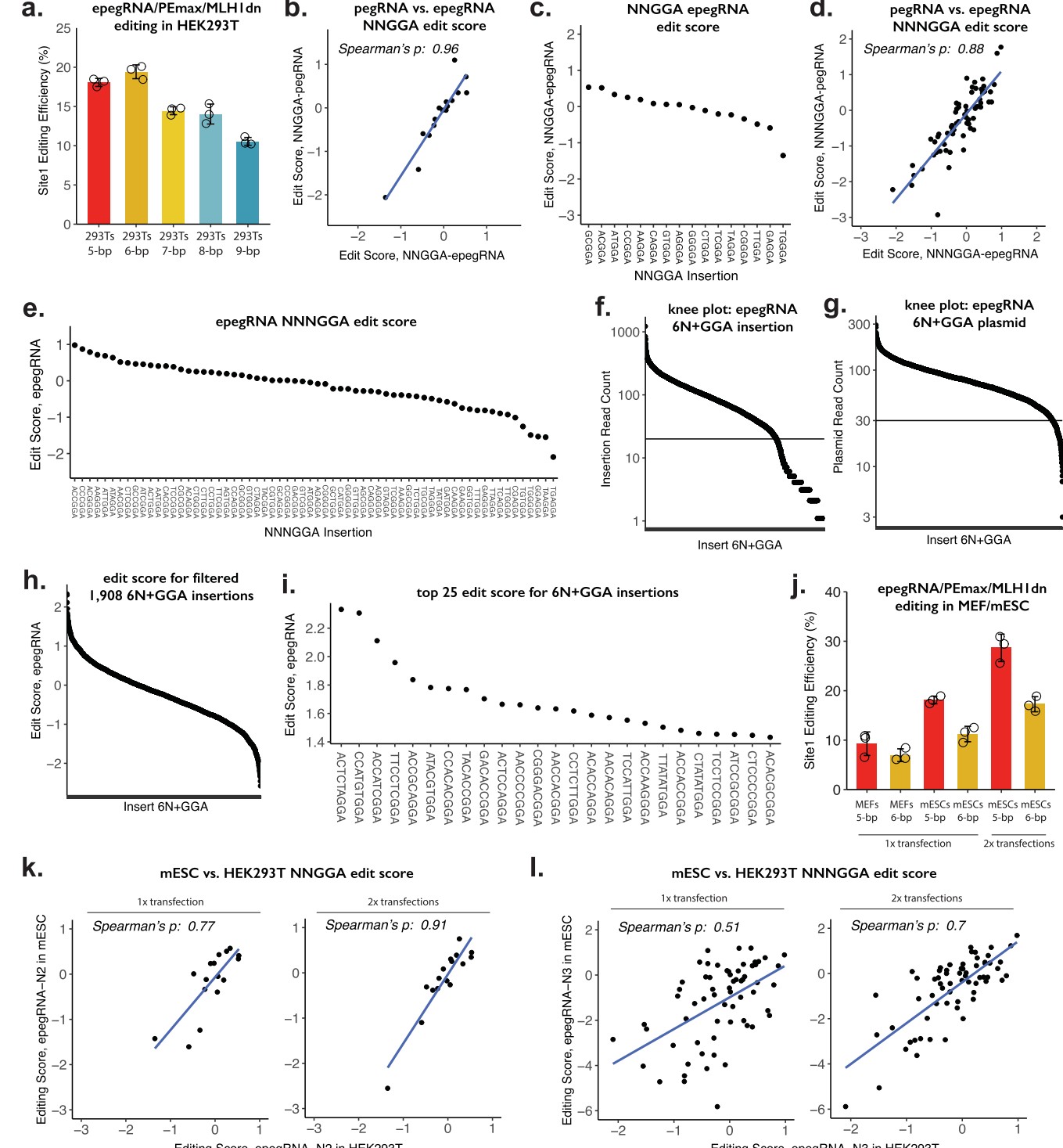

**Extended Data Fig. 2 |** See next page for caption.

**Extended Data Fig. 2 | Enhancements of prime editing facilitate DNA Typewriter's range and efficiency. a**. Editing efficiencies at the first site of 5xTAPE-1 integrated in HEK293T cells. A pool of plasmids expressing TAPE-1 targeting epegRNAs were transfected with the pCMV-PEmax-P2A-hMLH1dn plasmid. Five pools with different insertion lengths ranging from 5-bp (NNGGA) to 9-bp (NNNNNNGGA or 6N+GGA) were tested separately. The center and error bars are mean and standard deviations, respectively, from n = 3 transfection replicates. **b**. Scatterplot of 16 NNGGA edit scores with pegRNAs vs. epegRNAs. **c**. Edit scores for 16 NNGGA insertions with epegRNA. Edit scores for each insertion are calculated as log2 of the ratio between insertion frequencies and the abundances of pegRNAs in the plasmid pool. **d**. Scatterplot of 64 NNNGGA edit scores with pegRNAs vs. epegRNAs. **e**. Edit scores for 64 NNNGGA insertions with epegRNAs. **f**. Knee plot of read-counts for 4,096 possible 6N+GGA insertions, across three replicates. A minimum threshold of requiring at least 20 reads for a given insertion in each of the three transfection replicates was determined based on this plot. **g**. Knee plot of read-counts for 4,096 possible 6N+GGA-inserting pegRNAs from the pool of plasmids. A minimum threshold of 30 reads for each insertion plasmid was determined based on this plot. **h**. Edit scores for 1,908 6N+GGA insertions. Only insertions that appeared more than 20 reads in each of three transfection replicates and more than 30 reads in the sequencing of the plasmid pool were considered. Edit scores for each insertion are calculated as log2 of the ratio between insertion frequencies and the abundances of pegRNAs in the plasmid pool. **i**. Top 25 edit scores for 6N+GGA insertions. **j**. Editing efficiencies at the first site of 5xTAPE-1 integrated in the mouse embryonic fibroblasts (MEFs) or mouse embryonic stem cells (mESCs). For mESCs, up to two sequential transfections of a pool of epegRNA-expressing plasmids were tested. The error bars are standard deviations from n = 3 transfection replicates. **k,l**. Scatterplot of 16 NNGGA (**k**) and 64 NNNGGA (**l**) edit scores with epegRNAs in mESCs vs. HEK293T cells. Edit scores were calculated after one transfection (left) or two serial transfections (right) of the same pool of pCMV-PEmax-P2A-hMLH1dn/ U6-epegRNA plasmids. The edit score calculated with two serial transfections showed higher correlations (Spearman's $p$) with the edit score measured in HEK293Ts, probably due to better coverage of the insertion pools. Edit scores shown in this figure are calculated by combining sequencing data across n = 3 transfection replicate experiments.

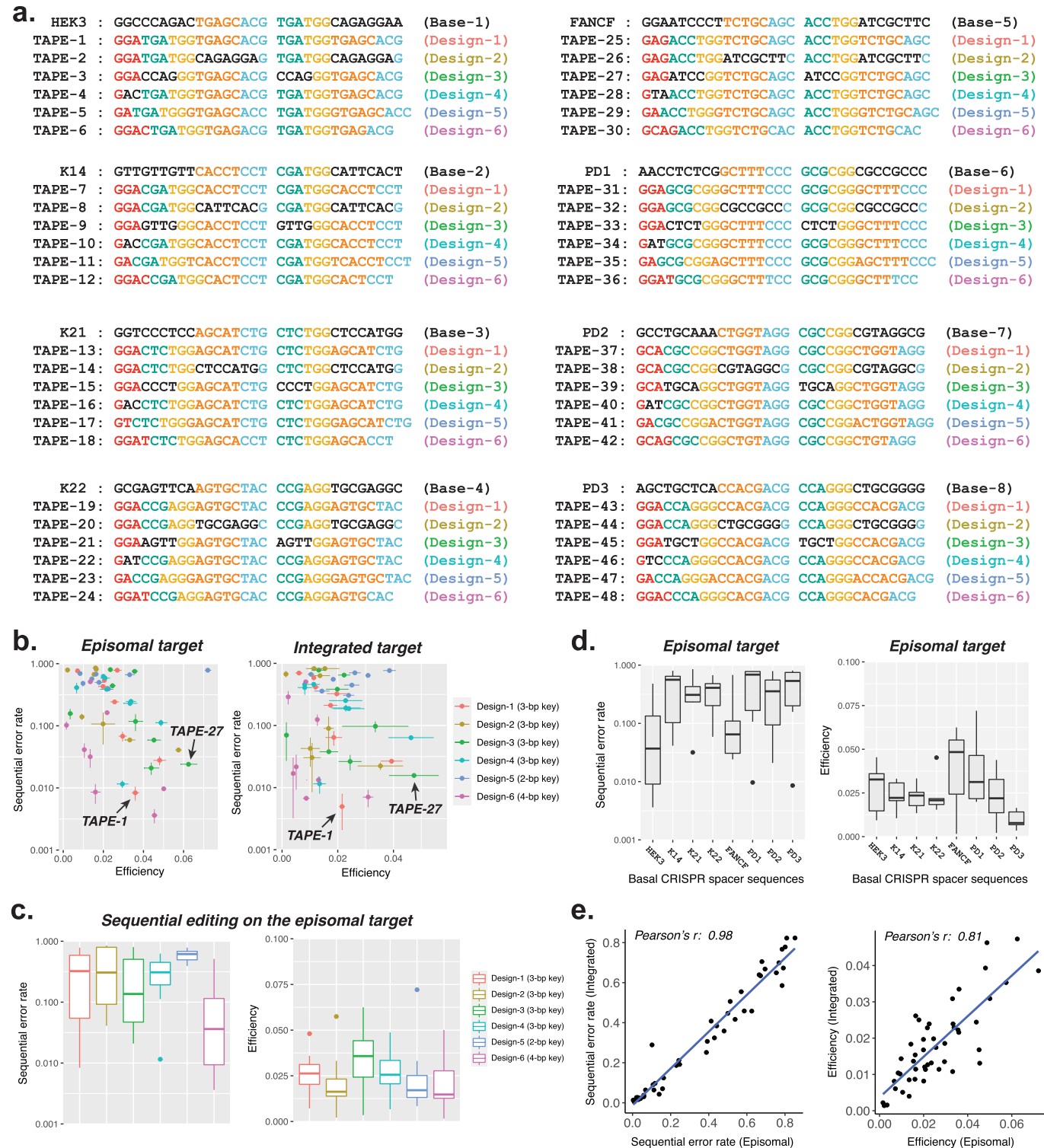

**Extended Data Fig. 3** | See next page for caption.

**Extended Data Fig. 3 | Characterising diverse DNA Tape designs for efficiency and directional accuracy. a**. Deriving 48 TAPE designs from the eight basal CRISPR spacer sequences that previously demonstrated reasonable prime editing efficiencies[2,26,27] via six distinct sequence shuffling procedures. **b**. Efficiency (fraction of edited reads out of all reads) vs. sequential error rate (fraction of edited reads inconsistent with sequential, directional editing out of all edited reads) for 48 3xTAPE constructs on episomal DNA (left) and piggyBAC transposon integrated DNA (right). Both horizontal and vertical error bars are standard deviations from n = 3 transfection replicates. **c**. Boxplots of the efficiencies and sequential error rates of 3xTAPE constructs derived from 8 basal sequences for each of 6 design procedures. Each data point is either mean efficiencies or mean sequential error rates over n = 3 independent transfection experiments with 8 basal sequences in each experiment. In general, a longer key sequence was associated with a lower error rate, while a longer insertion did not appreciably impact efficiency (*e.g.* NNGGAC with Design-6 vs. NNGA with Design-5). **d**. Boxplots of sequential error rates (left) and efficiencies (right) of 3xTAPE constructs grouped by their basal CRISPR target sequences. Each data point is either mean efficiencies or mean sequential error rates over n = 3 independent transfection experiments with 6 design procedures in each experiment. Boxplot elements in **c,d** represent: Thick horizontal lines, median; upper and lower box edges, first and third quartiles, respectively; whiskers, 1.5 times the interquartile range; circles, outliers. **e**. Correlation between the sequential error rate (left) and editing efficiency (right) of each 3xTAPE construct either in the context of episomal DNA vs. integrated DNA. Each data point is both mean efficiencies and mean sequential error rates over n = 3 independent transfection experiments with 48 designs in each experiment.

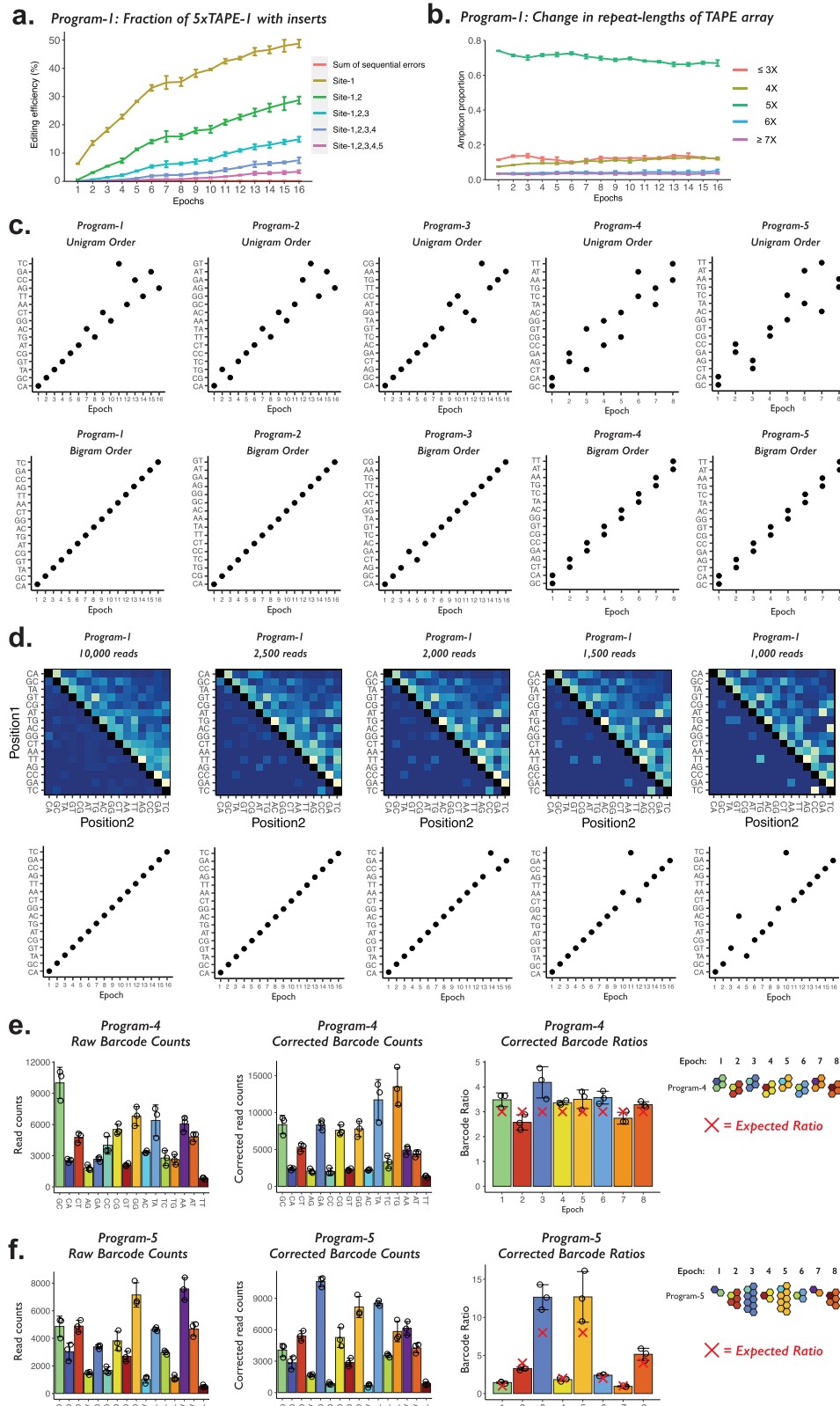

**Extended Data Fig. 4** | See next page for caption.

**Extended Data Fig. 4 | Inferred event order and magnitude from sequential transfections. a**. Sequential editing efficiency and sum of sequential errors from five sites in 5xTAPE-1 across 16 transfection epochs of Program-1. **b**. Repeat-length change of 5xTAPE-1 array sampled over 16 transfection epochs. **c**. For each of the five transfection programs, the event orders are inferred using "Unigram" (top) and "Bigram" (bottom) information. **d**. Undersampling analysis of Program-1. From the original 277,397 sequencing reads used for Program-1, we undersampled to 10,000, 2,500, 2,000, 1,500, or 1,000 reads. For each sampling point, the bigram transition matrix (top) was plotted and order of events (bottom) were inferred using bigram information. In **c,d**, sequencing reads from n = 3 independent transfection experiments are combined. **e,f**. For Program-4 (**e**) and Program-5 (**f**), the absolute barcode read counts (left) are corrected based on the edit score of 16 NNGGA barcodes (middle), and used to calculate the relative magnitude of two co-transfected barcodes (right). The expected barcode ratios are marked with a red "X" mark in each epoch. The center and error bars in panels (**a**), (**b**), (**e**), and (**f**) are mean and standard deviations, respectively, from n = 3 transfection replicates.

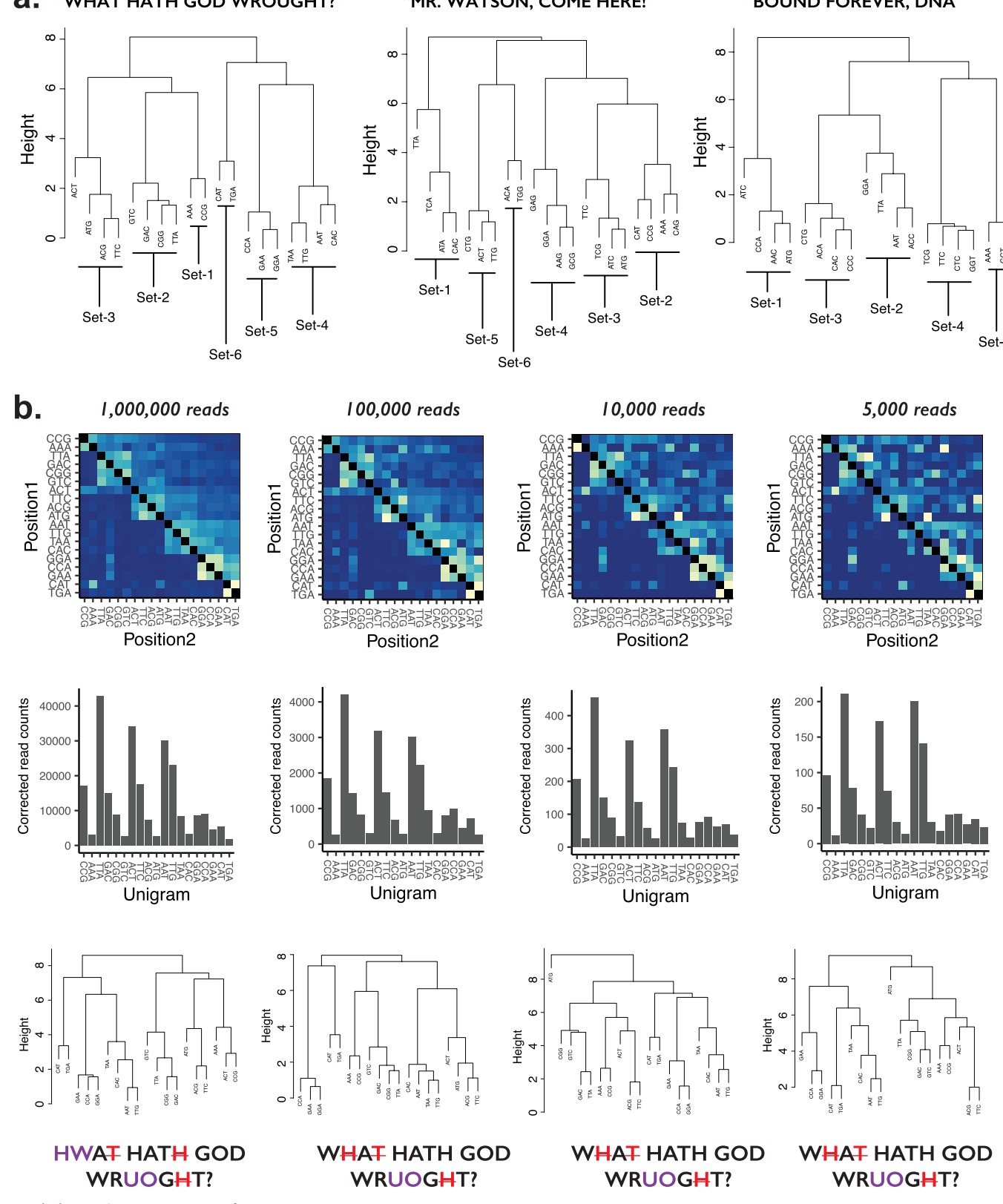

**Extended Data Fig. 5** | See next page for caption.

**Extended Data Fig. 5 | Inferring the barcode overlap in each message.**
**a**. Hierarchical clustering analyses of identified unigram barcodes based on the bigram matrices. For each message, the normalised bigram matrix was converted to a distance matrix using the euclidean distance measure. The resulting distance matrix was then used for clustering 3-mer barcodes using the complete-linkage clustering method, resulting in a cluster dendrogram for each message. Based on these dendrograms, groups of 2 to 4 barcodes were manually grouped as putative co-transfection sets, and ordered within the set based on unigram frequencies. Sets were ordered relative to one another using the normalised bigram matrix, following the sorting algorithm described in the text. **b**. Undersampling analysis of the short text "WHAT HATH GOD WROUGHT?". From the original 1,256,996 sequencing reads, we undersampled to 4 sampling points: 1,000,000, 100,000, 10,000, and 5,000 reads. For each sampling point, the bigram transition matrix (top), the corrected unigram counts (middle), and the hierarchical clustering (bottom) were plotted. From these, the original short text was inferred at the end. Both 2D histogram and corrected read counts are calculated by summing the sequencing reads over n = 3 independent transfection experiments. Read counts are corrected using the edit score for each insertion barcode.

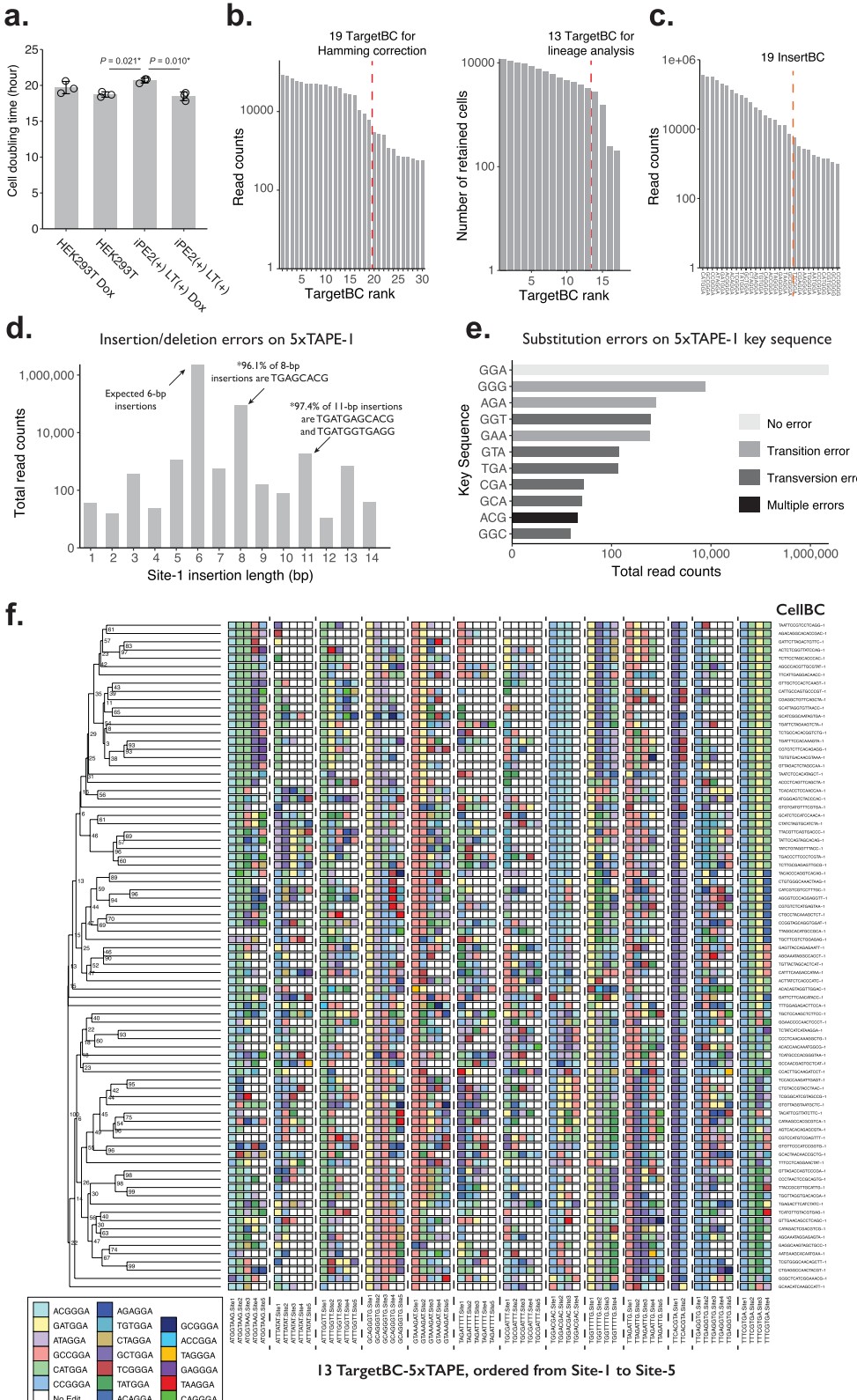

**Extended Data Fig. 6** | See next page for caption.

**Extended Data Fig. 6 | Characterising the monoclonal lineage tracing experiment. a**. Cell doubling times measured for HEK293T and the monoclonal lineage tracing cell line (iPE2(+) LT(+)), with or without Doxycycline (Dox). The presence of Dox lengthened the cell doubling time, possibly negatively affecting the cell physiology. *P* values were obtained using the two-tailed Student's t-test with Bonferroni correction: only *P < 0.05 are shown. The center and error bars are mean and standard deviations, respectively, from n = 3 independent experiments. **b**. Determining a set of valid TargetBCs based on frequencies. The Y-axis is on a log10-scale. Recovered TargetBCs were first ranked by their read counts to estimate multiplicity of infection (MOI) (left). Any additional TargetBCs that are 1-bp Hamming distance away from the set of 19 were corrected. We then retained 3,257 cells for which we recovered 13 of the most frequent TargetBCs (excluding one tape sequence with a corrupted type-guide) for lineage analysis (right). **c**. Read counts of InsertBCs observed in TAPE-1 arrays. The Y-axis is on a log10-scale. For the 3,257 selected cells, we additionally required that all observed edits were amongst the 19 most frequent InsertBCs in the overall dataset, as we presume this to be the valid set of pegRNA-defined insertional edits. **d**. Characterization of indel error rates of prime editing on TargetBC-5xTAPE-1 arrays. The Y-axis is on a log10-scale.

Correct length insertions with prime editing are > 100-fold more likely than an insertion of a different length product. Furthermore, some of the apparent longer insertions are likely to correspond to a contraction of TAPE-1 monomer within 5xTAPE-1 before the integration, such as contraction of TGATGGTGAGCACG TAPE-1 monomer to the observed TGAGCACG 8-bp sequence appearing between two TAPE-1 monomers. **e**. Characterization of substitution error rates during prime editing-mediated insertion of the GGA key sequence on TargetBC-5xTAPE-1 arrays. The X-axis is on a log10-scale. Correct insertions are > 100-fold more likely than insertions with substitution errors. The most frequent class of errors are transition errors, and these may be occurring during PCR amplification or sequencing-by-synthesis of cDNA amplicons, rather than during prime editing. Data in panel (**b**) to (**e**) is generated from n = 1 monoclonal lineage experiment, followed by n = 1 single-cell RNA-seq data collection. **f**. A lineage tree constructed by order-aware UPGMA for a clade of 81 cells drawn from the larger tree. Numbers next to branching points denote bootstrap values out of 100 resamplings. The 59 sites of 13 TargetBC-associated tape arrays are represented to the right, with InsertBCs colored by edit identity. Cells are identified by the 16-bp CellBCs (10X Chromium v3 chemistry) listed on the far right.

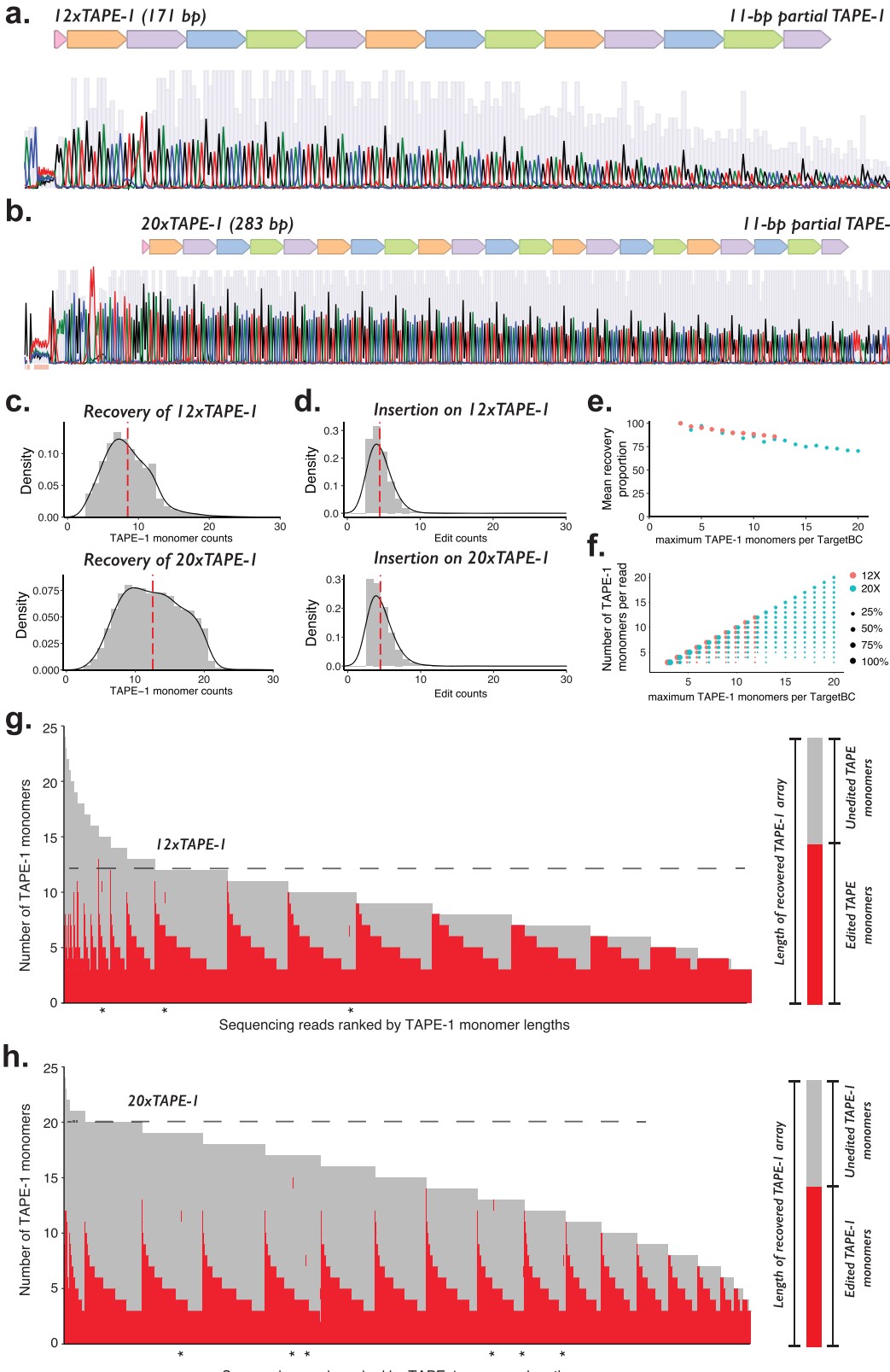

**Extended Data Fig. 7** | See next page for caption.

**Extended Data Fig. 7 | Editing and recovering longer TAPE arrays.**
**a-b**. Sanger sequencing traces for cloned (**a**) 12xTAPE-1 and (**b**) 20xTAPE-1 constructs. Each TAPE-array includes the 3-bp key sequence (GGA for TAPE-1), 12 or 20 repeats of 14-bp TAPE-1 monomer, and a 11-bp partial TAPE-1 monomer to serve as a prime-editing homology sequence for the last editing site. Nucleotides A, C, G, and T, in Sanger sequencing traces are colored green, blue, black, and red, respectively. Grey bars in the background are proportional to quality (Phred-scale) for each base call. **c-h**. Integration, editing, and recovery of 12x and 20xTAPE-1 arrays. Each construct was integrated into PE2(+) 3N-TAPE-1-pegRNA(+) HEK293T cell line in triplicate, cultured for 40 days for prolonged editing, and recovered via PCR and long-read sequencing on the PacBio platform. Circular consensus sequencing (CCS) reads that had at least 3 NNNGGA insertions and no small indel errors were grouped based on the site of integration (using 8-bp TargetBC barcodes), and a read with the maximum number of TAPE-1 monomers (and within that set, the read with the maximum number of edits) was selected per TargetBC. **c**. Histogram of the number of TAPE-1 monomers recovered from -12xTAPE-1 (top) and -20xTAPE-1 (bottom) integrants. **d**. Histogram of number of edits recovered from -12xTAPE-1 (top) and -20xTAPE-1 (bottom) integrants. **e**. For TargetBC groups with a given maximum number of TAPE-1 monomers (X-axis), we show the mean proportion with the same number of monomers as the maximum (Y-axis), for both 12xTAPE-1 (red) and 20xTAPE-1 (blue) integrants. We conclude from this that shorter arrays are more stable, and that the length-dependent stability is consistent between the two experiments. **f**. Similar to (**e**), but showing the full distribution of monomer lengths (Y-axis) for each TargetBC group with a given maximum number of TAPE-1 monomers (X-axis), for both -12xTAPE-1 (red) and -20xTAPE-1 (blue) integrants. The size of dots are proportional to these proportions. Data shown in panels (**c**) to (**f**) are generated by combining sequencing reads from n = 3 transfection replicate experiments. **g,h**. Recovery of (**g**) -12x-TAPE-1 and (**h**) -20x-TAPE-1 arrays after prolonged editing. Edited portions of each TAPE-array are colored red and overwhelmingly exhibit sequential editing. Very rarely, we observe instances of non-sequential editing, *e.g.* internal monomers that are edited. These are marked with asterisks below the corresponding column.

# Reporting Summary

## Statistics

For all statistical analyses, confirm that the following items are present in the figure legend, table legend, main text, or Methods section.

| n/a | Confirmed | |
|---|---|---|
| ☐ | ☒ | The exact sample size (*n*) for each experimental group/condition, given as a discrete number and unit of measurement |
| ☐ | ☒ | A statement on whether measurements were taken from distinct samples or whether the same sample was measured repeatedly |
| ☒ | ☐ | The statistical test(s) used AND whether they are one- or two-sided *Only common tests should be described solely by name; describe more complex techniques in the Methods section.* |
| ☒ | ☐ | A description of all covariates tested |
| ☒ | ☐ | A description of any assumptions or corrections, such as tests of normality and adjustment for multiple comparisons |
| ☐ | ☒ | A full description of the statistical parameters including central tendency (e.g. means) or other basic estimates (e.g. regression coefficient) AND variation (e.g. standard deviation) or associated estimates of uncertainty (e.g. confidence intervals) |
| ☐ | ☒ | For null hypothesis testing, the test statistic (e.g. *F*, *t*, *r*) with confidence intervals, effect sizes, degrees of freedom and *P* value noted *Give P values as exact values whenever suitable.* |
| ☒ | ☐ | For Bayesian analysis, information on the choice of priors and Markov chain Monte Carlo settings |
| ☒ | ☐ | For hierarchical and complex designs, identification of the appropriate level for tests and full reporting of outcomes |
| ☐ | ☒ | Estimates of effect sizes (e.g. Cohen's *d*, Pearson's *r*), indicating how they were calculated |

*Our web collection on statistics for biologists contains articles on many of the points above.*

## Software and code

Policy information about availability of computer code

| Data collection | We used next-generation sequencing platform (Illumina Nextseq 550 and Miseq, and Pacific Bioscience Sequel) and associated software within the instruments for data collection. |
|---|---|
| Data analysis | We have used BCL2fastq (version 2.20) for data analysis. Custom Python and R analysis scripts are included with the manuscript. R packages Ape (5.6-1) and Phangorn (2.8.1) are used to analyze the lineage recording data. |

For manuscripts utilizing custom algorithms or software that are central to the research but not yet described in published literature, software must be made available to editors and reviewers. We strongly encourage code deposition in a community repository (e.g. GitHub). See the Nature Portfolio guidelines for submitting code & software for further information.

## Data

Policy information about availability of data

All manuscripts must include a data availability statement. This statement should provide the following information, where applicable:
- Accession codes, unique identifiers, or web links for publicly available datasets
- A description of any restrictions on data availability
- For clinical datasets or third party data, please ensure that the statement adheres to our policy

Raw sequencing data have been uploaded on Sequencing Read Archive (SRA) with associated BioProject ID PRJNA757179.

# Field-specific reporting

Please select the one below that is the best fit for your research. If you are not sure, read the appropriate sections before making your selection.

☒ Life sciences    ☐ Behavioural & social sciences    ☐ Ecological, evolutionary & environmental sciences

For a reference copy of the document with all sections, see nature.com/documents/nr-reporting-summary-flat.pdf

# Life sciences study design

All studies must disclose on these points even when the disclosure is negative.

| | |
|---|---|
| Sample size | No sample-size calculation was performed. For experiments involving in vitro cell lines, we have performed experiments in three technical replicates for transfection and the measurement of doubling times, which is sufficient to estimate the technical variation observed in genome editing. Values (editing efficiencies) measured in technical replications were used to calculate mean and standard deviations. |
| Data exclusions | Sequencing data exclusion criteria is outlined in the method section, including filtering out the substandard data in single-cell measurements, following the general practice in the field. |
| Replication | All but one genome editing experiments were performed at least with three technical replicates (independent transfections of cells cultured separately previous to transfections). Single-cell RNA-seq experiment with lineage recording was done in a single replicate experiment. All attempts at replication for genome editing presented in this study were successful. |
| Randomization | This is not relevant to our study, because we only use human and mouse cells cultured in vitro for measuring genome editing efficiencies. |
| Blinding | Blinding was not relevant to our study, because the results derived from sequencing experiments are quantitative and did not require a subject interpretation or judgment. |

# Reporting for specific materials, systems and methods

We require information from authors about some types of materials, experimental systems and methods used in many studies. Here, indicate whether each material, system or method listed is relevant to your study. If you are not sure if a list item applies to your research, read the appropriate section before selecting a response.

## Materials & experimental systems

| n/a | Involved in the study |
|---|---|
| ☒ | ☐ Antibodies |
| ☐ | ☒ Eukaryotic cell lines |
| ☒ | ☐ Palaeontology and archaeology |
| ☒ | ☐ Animals and other organisms |
| ☒ | ☐ Human research participants |
| ☒ | ☐ Clinical data |
| ☒ | ☐ Dual use research of concern |

## Methods

| n/a | Involved in the study |
|---|---|
| ☒ | ☐ ChIP-seq |
| ☒ | ☐ Flow cytometry |
| ☒ | ☐ MRI-based neuroimaging |

# Eukaryotic cell lines

Policy information about cell lines

| | |
|---|---|
| Cell line source(s) | HEK293T (ATCC), MEF (PMEF-CFL; Millipore-Sigma), mESCs (E14tg2a; Dr. Christian Schröter) |
| Authentication | All cell lines were used as received without further authentication. |
| Mycoplasma contamination | Cell lines were not tested with mycoplasma contamination. |
| Commonly misidentified lines (See ICLAC register) | No commonly misidentified cell lines were used in the study. |

