## [Peer Review File · Nature]

Manuscript Title: A time-resolved, multi-symbol molecular recorder via sequential genome editing.

Reviewer Comments & Author Rebuttals

Reviewer Reports on the Initial Version:

Referees' comments:

Referee #1 (Remarks to the Author):

Choi et al presents a CRISPR-Cas prime editing-based approach to record temporal profiles of molecular and cellular events in mammalian cells, which include presence of plasmids expressing certain prime-editing guide RNAs (pegRNAs) and cellular divisions. This framework, called DNA Ticker Tape, uses prime editing's capability to insert short DNA sequences to the specific sites of the genome, where the insertion event disrupts the current target site and activates the next target site along the tandem array of partial CRISPR-Cas target sites. The authors first validate the concept by sequential genome editing of 2-5x arrays with a specific prime editing target sequence (TAPE-1). While they successfully show sequential and directional editing of TAPE-1 arrays with modest editing rates (5.3-7.2%) and low error rates (0.07-0.36%), it is unclear whether the sequential genome editing performance is generalizable considering the huge (>10%) error rates for the most other TAPE designs, which is crucial to record multiple events in complex biological contexts. The authors then demonstrate temporal recording of transfection events of plasmids expressing 16 barcoded pegRNAs. The data provides an exciting possibility to reconstruct temporal profiles of past multiple molecular events in mammalian cells, although the data presented thus far does not yet support their claims for 'signal' recording. Next, they used the system to write short texts into the genomes of mammalian cells by transfecting plasmids expressing character-encoded pegRNAs over the course of 5-6 time points. Finally, the authors demonstrated recording of cellular lineage histories using their sequential genome editing framework with constitutively expressed pegRNAs. The ordered nature of DNA Ticker TAPE framework enabled simple reconstruction of cell lineage histories without a need for special computational methods that are necessary for most DSB-based unordered lineage recording systems.

Overall, the paper provides a new strategy of using CRISPR-Cas prime editing to record molecular events into the genomes of mammalian cells with temporal information. The key novelty of the work is using prime editor for development of a molecular recorder and their successful demonstration of temporal recording over the course of relatively longer period than other previous works up to 16 epochs (~48 days). However, the TAPE design with overlapping CRISPR-Cas target sites as a moving DNA writing address is similar to the DOMINO system (Farzadfard et al., Mol Cell 2019), where CRISPR-Cas base editor are used for sequential genome editing-based recording of actual biological signals with much higher efficiencies. Discussion of the uniqueness of the current approach could better highlight novelty. Furthermore, current demonstration of the DNA Ticker Tape technique is fairly underwhelming for their unnecessarily overstated claims on its potential applications and advantages for recording temporal profiles of thousands of distinct biological signals and cellular lineages. The authors did not elaborate on describing what they found from the data and just put them on the supplementary tables which are not labeled with details and is therefore not straightforward to digest. Nevertheless, this CRISPR-Cas prime editing-based temporal recording is likely to be an exciting foundation towards a highly efficient and programmable platform for temporal biological recording and lineage tracing in mammalian cells and potentially multicellular organisms.

Several points should be addressed in the current version of the manuscript.

Major comments:

1. The authors' first TAPE-1 construct happen to be with one of the lowest error rates among the 48 TAPE constructs characterized later (Figure 2). Interestingly, so many other TAPE designs show very high error rates (fraction of edited reads inconsistent with sequential and directional genome editing) with some of them even reaching above 80% that it does not seem to be generalizable at all to various target sequences. While they mention that the length of key sequences in the TAPE sites is associated with error rate, it is unclear what other factors are affecting this huge range of error rates across different TAPE sequences. Is this just something intrinsic to CRISPR-Cas9 target specificity or prime editing specificity for different target sequences and mismatches? Can TAPE performance be predicted based on sequence of the target sites and can highly efficient TAPE designs with lower error rates be forward designed for multiplex recording of biological events across many different TAPEs?

2. Also, it seems that even very small variations (2-3 bps) in the barcode sequences within the pegRNA sequences can affect prime editing efficiencies significantly (Supplementary Figure 1). Probably for this reason, when the authors record presence of two different barcodes at different ratios over time for the temporal transfection programs 4 and 5 in Figure 3 to ask whether the relative strength of signals could be inferred from DNA Ticker Tape, only the relative ratios between the programs 4 and 5 are compared (Figure 3h), not between inferred and actual ratios of barcodes for each epoch within each program. This is only described in the Supplementary Table 3, which is not labeled with details. I suspect that the system may not be very accurate at recording relative strength of signals, probably due to the confounding effect of barcode sequences on prime editing efficiencies. This necessitates characterization of prime editing efficiencies for all barcoded pegRNAs beforehand for multiplex signal recording, limiting the scalability of the system. The authors should temper these overstated claims on scalability (or provide evidence to support these claims), elaborate on the advantages and limitations of the system, and plot the data they have in supplementary tables whenever possible.

3. While the authors claim capability of signal recording throughout the text, what they experimentally demonstrate is constitutively expressed pegRNAs on the transiently transfected plasmids. While the results in Figure 3 show that this is on the correct path, but the next set of advances are really what is going to push this work to a regime that is biologically useful. It is not clear why text messages and not biological signals were chosen as a demonstration of recording for Figure 4. Could experiments be done for cells to sense and record biological signals over time using several inducible promoters for expression of barcoded pegRNAs? Without such demonstrations, the authors should not describe their transfection events as signaling events and the claim for biological 'signal' recording should be revised.

4. For text message recording experiments, the character deletion errors are mostly due to repeated use of pegRNA barcodes for repeated characters. Why does this still happen when the authors tried to avoid this error mode by using two different barcode sequences alternatively for the same characters? This error mode needs to be discussed more in details as it is relevant to the potential performance of the system for recording actual biological signals that fluctuate and repeat constantly.

5. For lineage recording experiment, the starting single cell contains 23 inserted TAPE-1 arrays with 23 barcoded pegRNA cassettes in its genome and all the arrays are being independently edited by all the 23 pegRNAs over the course of 32 days. The authors' current analysis for some of the selected individual TAPE-1 arrays show potential utility of the system to track cellular lineages by simply ordering of the barcodes inserted into the arrays. However, analysis on individual TAPE-1 arrays is not sufficient for their experimental design and the whole 23 different TAPE-1 arrays

together should be considered for constructing cellular lineages. Perhaps all these TAPE-1 arrays could be linked in single cells? Alternatively better defined experiments could be done with single TAPE-1 array with multiple barcoded pegRNA cassettes integrated into the genome of the starting single cell. Again, the authors just put most of the data for this experiment in Supplementary Table 4. It would be informative to plot and describe how all 23 individual barcoded TAPE-1 array populations behaved during the 32 days of cell divisions.

Minor comments:

6. The authors may want to place their work in the correct context of other works. While many molecular recording techniques for mammalian cells records signals stochastically to unordered target sites, several recent approaches (Tang and Liu, *Science* 2018; Farzardfard et al., *Mol Cell* 2019; Loveless et al., *Nat Chem Biol* 2021, etc) write biological information in sequential and unidirectional manner. These techniques should be discussed and compared with DNA Ticker Tape more critically. Multiplexability is the key advantage of prime editing-based approach over the previous approaches, which could be emphasized. How many similar pegRNA cassettes with only several mismatches can practically implemented in the living cells? How stable are they in the genome? What are needed to address these potential limitations?

7. Again, columns in most supplementary tables should be relabeled clearly (or put notes to describe what each column is showing) and it would be nice to plot the data whenever possible.

8. For Figure 1, it seems they did 3 replicate experiments. It would be nice to include standard deviations for the numbers in Figure 1.

9. It would be informative to see how many sequencing reads are required for correct ordering of temporal profiles (Figure 3) or correct calling of text encoded (Figure 4).

Referee #2 (Remarks to the Author):

Summary: Here, Choi et al. create and validate a molecular construct designed to be targeted by Cas9 in an ordered manner by leveraging insertion events created via prime editing. In a simple statement, this process has one active Cas9 target that perfectly matches the protospacer, and when Cas9 binds and nicks the DNA, prime editing from the Cas9-RT fusion (prime editing enzyme) templates the pegRNA to insert sequence at the cut site. This sequence insertion “moves” the active Cas9 target down one site – effectively respacing the full target and enabling a new PAM site within the inactive third site to be used for the next editing reaction. Choi and colleagues demonstrate effective and elegant proof-of-principle experiments for their encoding strategy. These experiments determine the efficiency of prime editing to occur sequentially and the accuracy of it as an information encoder. While the rate of editing is quite low, the subset of TAPES that are edited do provide proof-of-concept. Throughout the manuscript, they demonstrate its efficacy as a transient transfection, transposon-mediated insertions, and as a lentiviral system, which indicates it could be broadly utilized across many types of cell types and in vitro experiments that are amenable to these techniques. As an application of this technology, Choi et al. employ an in vitro lineage tracing experiment, demonstrating their ability to use this as a lentiviral system for tracking lineage relationships from a single cell over time. Overall, although this is an exciting system, the manuscript as presented covers two distinct applications of this approach, it would be ideal to gain some biological insight from these applications.

Major Comments:

- Choi and colleagues demonstrate a truly elegant idea, with a solid set of validation experiments for this technique as a data storage device (Figs 1 – 4), however, the proof of concept experiments rely on relatively ‘clean’ gene expression, raising questions on how well this approach would work

in a real, noisy biological context. Moreover, we are curious how this method would generally perform in a less robust cell line. Since HEK cells are very resilient, they appear to be well suited to a DNA-based assay with many rounds of passaging, transfections, and/or long-term dox treatments. This does, however, raise questions of the broad applicability of this approach to other cells – particularly primary cells. The authors may consider deploying their method to a well-characterized cell differentiation paradigm to confirm known events or reveal new insights.

- The authors comment on the possibility of recording sequential multiplexed signals. However, again due to the low editing rate, the practicality of this seems questionable without some kind of demonstration. Even with a dual cassette, would it not be difficult to parse out cells that received a signal but failed to accrue edits vs. those that received both (the effects of which would be compounded after each signaling epoch)? We assume the authors intend to capture information about single cells.

- The lineage tracing portion of the manuscript requires further development. No biological insights are provided, though the authors indicate that it could be performed via interrogation of sequential signals that are linked to pegRNA expression. Indeed, further biological interrogation is necessary to demonstrate the ability of this system to record valuable information over time. Though the lineage tracing experiment is intriguing, it fails to demonstrate the complexity of a lineage tracing system or create a lineage tree from all of the TAPes within the monoclonal cell population. Further experiments to deduce the complexity of labeling are also necessary to determine the efficacy of the system to individually label cells in a unique manner, increasing the resolution of the tree.

- With regards to the lineage tracing application, a “clone mixing” experiment of multiple monoclonal populations to determine if multiple unique trees can be determined would go a long way to demonstrating the efficacy of this as a single cell lineage tracing system. Provided this approach is worthwhile, understanding cell states that are derived from lineage identities would be useful as interrogated biological information from the lineage, as previous publications, such as GESTALT or scGESTALT have done.

- The example of lineage recording is interesting and shows potential, but it is ultimately incomplete. The authors validate their ability to partition each original TargetBC by InsertBCs, presenting a strong foundation for lineage analyses. However, they do not present a feasible means to unify this data into a collective lineage tree, which is presumably the final goal? Without being able to identify individual cells within the monoclonal population, it is unclear how the authors propose consolidating the information across 23 TargetBCs from a single clone into a composite tree (or how they would leverage the lineage information after constructing such a tree?). Scaling this up to more than one clone would seem to present multiple challenges as well. Some clarification of this from the authors would be helpful.

Specific Comments:

- Page 2, last pp: This is an excellent description of the system that is implemented. However, the associated Figure 1a could be improved. The inclusion of a more detailed drawing that could include the function of the PE enzyme, how it integrates the barcode+key sequence and moves the write head would be more effective for the reader.

- Page 3, pp3: “Overall editing rates were modest, as only 5.3%, 7.2%, and 6.6% of all reads for 2xTAPE-1, 3xTAPE-1, and xTAPE-1, respectively, exhibited any editing.” These editing rates are low – can the efficiency be increased? If the utility of this becomes an information encoding process, it would be better to make it more efficient to demarcate past events in cells to increase the likelihood of capture in final cells of interest – especially with rare cell types. Further, with lineage tree construction, is there a reason that some cells may be biased to recording while others are not, as this would skew how effectively the labeling may occur in specific populations?

- Page 3, last pp: This is an honest discussion of the technology, with specific weaknesses

discussed. Would it be possible for the authors to determine a way to reduce the site of integration effects with alternative methods? For example, safe harbor loci, insulator elements, etc?

- Page 5, Fig 1: We find the ordering of this figure is slightly confusing – the panels for b and c would make more sense as panel a. Adding more detail to this figure that includes the sequence, such as a Cas9 and RT fusion construct drawn, highlighting the cut and editing functions, and ultimately showing the shift in write head would be more helpful. Panels d, e, and f are intuitive, they communicate the ideas well.

- Page 7 (Fig2c-2e): It is a bit unclear when reading the manuscript which figure panels fall into the episome style versus integration style of experiment. It would be a little easier to read if that was explicitly stated.

- Page 7, last pp: While the experiment increases the prospective number of tags that can be created (moving from 4^2 to 4^3 , and ultimately 4^6) to add complexity to the library of barcodes that can be integrated, the efficiency decreased markedly. One concern is that this could limit the potential of this technology to be a high-resolution, high-capture lineage recorder since the added complexity isn't compatible to labeling more cells. While the resulting lineage relationships detected may be more detailed, this seems like it would benefit from increased labeling efficiency with less signals to encode more meaningful information – such as if a cell had specific gene expression. The transcriptional history of a cell is difficult to capture, though this process could offer a means to that end.

- Page 9, pp 1: Considering the hamming distance between any two barcodes could be as little as 1, how does sequencing error impact the confidence of calling which barcode was integrated into the system?

- Page 9, pp 2: This approach of the multiple programs to record information and understand the behaviors of prime editing to encode the information into the TAPE is impressive.

- Page 10, last sentence: "Taken together, these results show that DNA Ticker Tape can record, recover, and decode complex event histories including the order, overlap, and relative strength of signals." This is an excellent summary of the potential of the technology. This is also why it may be less attuned for high-resolution lineage tracing than it is for other information encoding. High-resolution lineage tracing approaches generally leverage the random encoding of very complex libraries of barcodes, either generated via scarring or randomized synthesis. This statement outlines why it would be so well suited to not creating RANDOM events, but better at recording SPECIFIC events.

- Page 11, Fig3c-3g: The heatmaps depicted here are intuitive and illustrate the relationships between the PE program and editing events very well.

- Page 12, "Recording and decoding short text messages" and Fig 4: The concept of recording and decoding the sentences is clever – a fun and whimsical read as a proof-of-principle experiment.

- Page 15: Based on the frequencies of usage of InsertBC barcodes, we calculated their Shannon entropy to be 3.64 and 3.70 bits for Site-1 and Site-2, respectively, out of a theoretical maximum of 4.46 bits if the probabilities of the 22 InsertBC barcodes were equal." This sentence is somewhat unclear and difficult to read. Perhaps outlining the relationships between the InsertBCs and the pegRNA and the insertions would help?

- Page 16, first 3 lines: Since the damage to the tape was discovered, this is a type of informational dropout that could occur, though the DNA Ticker Tape system is generally robust to the types of dropout generally observed. We are unsure if the "damaged tape" occurs at a low enough rate since we only see data from one monoclonal experiment. We would like to see this

performed in many more cells to determine how often it occurs and if this off-target damage makes information dropout more comparable to other single cell lineage tracing methods.

- Page 16, first full pp: When discussing prime editing error rates, are there more comprehensive statistics that could be shared? We again worry about the power of this experiment since it was inferred from one clonally-related cell population.

- Page 16, pp 3, "Identity by state vs. identity by descent": Given that this is a single comparison of a monoclonal population, would it be easier to find recurrence if multiple, independent clones are compared? Then if cells with different pools of barcodes are used (though with the same transduced barcode pool) and compared, repeated barcode patterns would determine how effective this method is at creating discernable lineage trees or repeating patterns. This "clone mixing" experiment would be a way to determine the efficacy of the method by measuring repeats in clonal-subclonal signatures.

- Page 19: The synthetic minisatellite experiment is challenging due to the nature of the repeats of the DNA. Even after sequential editing, is it possible that shortening of the minisatellite sequence could occur, leading to information dropout? Could an optimal length of TAPE be determined, to maximize the ordered, recordable information but minimize the possibility of TAPE contraction?

- Finally, most of the manuscript is extremely well written, but there are some vague phrases and points that can be clarified about the analysis and system (see above). Overall, the figures are comprehensive, but the reader would greatly benefit from a more technical drawing of the process of the "write head" movement from site 1 to site 2. Fully drawing out the prime editing enzyme and demonstrating the process on the sequence to show the key and barcode integration would be helpful for an audience not intimately familiar with prime editing. Other figures would also do well to be altered to gain more clarity, and place sequences into supplemental tables where they can be easily assessed.

Referee #3 (Remarks to the Author):

This work describes a molecular system for logging temporal events in the genomes of eukaryotic cells. The technology has the potential to make a large impact in the understanding of biological systems. One can imagine immediate uses in improved lineage recording, but the greater potential advances will likely come from the ability to encode temporal sequences of molecular events as they happen (pathway activation, gene expression, cell-cell interactions) and decode them by sequencing later on. The fact that the technology is not yet fully developed enough to enable those applications does not detract from the strong impact of this work. However, because it will require further development, it is critical that this work better describe the fundamental parameters of the system and accurately contextualize the experiments.

1. The authors argue in the introduction that one drawback of existing Cas9 molecular recorders is toxicity, "Frequent DSBs are toxic." The authors should quantify the impact of their recording system on the cell physiology. At minimum they should quantify cell viability and doubling time with all components expressed over extended periods of time.

2. The authors describe the system as being used to record signals of interest: "Signals of interest are coupled to the expression of specific prime editing guide RNAs." However, the signal of interest

used in this work is transfection of barcoded pegRNA plasmids, which is of limited interest beyond demonstrating the potential of the technology in future applications. The work would be more accurately described without invoking the vague 'signals of interest' that makes it seem like biological events might have been recorded. Unless the authors would like to drive the system with an actual signal (rather than the physical delivery of the material to be encoded), I would recommend revising the language to more literally describe the results. For instance, "Each signal of interest is coupled to the expression or activity of a prime editing guide RNA (pegRNA)," could become something like, "To demonstrate the potential application, we encoded successive transfection events using specific primer editing guide RNAs (pegRNAs)." For another, "At the beginning of each epoch of each transfection program, one or more synthetic signals was introduced to a population..." should just be "...one or more pegRNA plasmids was introduced..." This applies throughout the manuscript to distinguish between what could be done with the system from what has been done with the system.

3. As a related point, the authors note while discussing the use of this system that "engineered guide RNAs whose activity is dependent on the binding of specific small molecules or ligands" could be recorded, but that "the further development of such signal conversion systems is a non-trivial challenge." It is hard to know how much of a challenge this might be without additional experimental insight into the sensitivity and dynamic range. How abundant are these pegRNAs as compared to endogenous transcripts, as assessed by qPCR or some other quantitative metric? What is the dynamic range, i.e. the relative editing efficiencies at different concentrations of pegRNA? Those numbers are critical to enable the necessary further development of the system into an event recorder.

4. Although the overall concept is communicated well and easy to understand, the molecular implementation took a lot of effort to figure out. It is unclear why 1b and 1c both need to exist as they show largely the same thing. In figure 1c, the schematic could be expanded and better annotated. It would be useful to annotate the write-head onto the figure. It would also be useful to depict the pegRNA with sequence. The text and legend refer to base positions of the PAM, but there are multiple PAMs. I could not figure out what purpose the mismatch bases are serving, which should be better explained.

5. The analysis of the experiment in Figure 1 is lacking in some basic descriptive results. A plot of the relative editing efficiency for each barcoded pegRNA should be included in the main figure, depicting the variability across replicates. The critical value I would suggest plotting is editing score. Given the large range of editing efficiencies with the different barcodes, it would be useful to have the barcode itself on that plot as well, so that one might, at the minimum, select useful barcodes in future iterations. It would be ideal to investigate if there are any rules to the high editing or low editing barcodes. Is there any chance that the high editing barcodes actually include prime editing errors, inflating their frequency?

6. The authors point out a pseudo-processivity where the editing frequency is increased from the first to the second position. It appears that it might also decline beyond the second position, apparent in the supplemental data and consistent with results further into the paper. Data for editing efficiency by position for Figure 1 and 3 should be plotted and discussed in terms of the accuracy and temporal limits of the recording.

7. Since the interpretation of the pseudo-processivity is inaccessibility of some editing sites when integrated randomly into the genome, the authors should look for this effect in the episomal plasmid editing (in figure 2). If their interpretation is correct, there should be no difference in editing efficiency by position in the episomal plasmid. If there is, they need to revise their interpretation.

8. In the lineage section it is stated that "Prime editing errors are rare, but do occur." What is rare? We need a quantification of these errors and error types, with particular focus on errors that

could be mistaken as recording data.

9. In Figure 2, most TAPes have error rates that look to be >50%. What sort of errors are these, prime editing errors or directional errors? The authors should include this categorization for all the designs tested.

10. For the experiments presented in Figure 3c,d, TAPes with an early barcode in position 1 (e.g. CA) should have every later barcode at nearly the same frequency in subsequent positions since they were all present for the same amount of time. Yet, the frequency falls off the further in time a barcode gets from the first barcode. That decline in frequency would indicate that TAPes are only recording a few signals before they stop recording, and therefore the recording is constrained in time. It is possible that this graph is not plotting exactly what I think it is. Either the result should be discussed or the analysis should be modified/clarified. Generally, the graphs in Figure 3c-g are very difficult to understand. The authors should consider a different type of plot or some sort of schematic to explain. The schematic in 3a is excellent.

11. To the point above, it is unclear if only the first and second position of the 5XTAPE are being analyzed for the order, or if all positions are being used. A useful analysis would be to quantify how much order information is contained in the TAPE by position in the TAPE, to help inform the design of future experiments.

12. The temporal recovery analysis used in Fig 3 is nearly identical in principle to the temporal analysis performed in Shipman et al., Science 2016 and Shipman et al., Nature 2017 to recover timing information from data encoded in CRISPR arrays. The data generated in those studies of an earlier temporal DNA molecular recorder is identical in structure to the data generated here. In those papers, CRISPR arrays were analyzed by taking pairwise comparisons of the order of individual spacers within the array (e.g. Shipman et al., Science 2016 Fig 3), just as the bigram frequencies are used in this work. This text should point out the similarities in data and analysis to help contextualize the work and establish standard analyses in this field.

13. This sentence doesn't make sense: "Correspondingly, although there are 4^3 or 64 potential InsertBC barcodes, only 22 were observed at appreciable frequencies, suggesting that one recurred by chance..." Why one?

14. The first citation to DNA based recordings is to the "theoretically promising but methodologically underdeveloped...DNA memory device," with a citation to a twenty year old patent. These devices have been physically demonstrated in the meantime, so this historical context is perhaps not the most relevant. In particular, the cited Roquet et al. 2016, Sheth et al. 2017, and Shipman et al. 2016 all demonstrate the physical recording of events over time in DNA. Sheth et al. and Shipman et al. use systems that generate incredibly similar data to the one presented here, with the major difference being eukaryotic vs prokaryotic. Building from that literature directly would better contextualize this work. Additionally, while this system enables recordings in eukaryotic cells, it presumably would not work in prokaryotic cells so distinguishing the parallel context for each approach would be useful.

15. Regarding the name DNA Ticker Tape, the use of the term Ticker Tape is so common in the field of molecular recording and has already been associated with other DNA based recorders that it seems likely to cause confusion. Would previous similar technologies that use DNA for recording, and are self-described as molecular ticker tapes, or biological ticker tapes not fall into this general umbrella? It would be better to include something to distinguish the pegRNA/prime approach within the name for clarity.

16. The pegRNA plasmid is not described, including no description of the promoter used.

Author Rebuttals to Initial Comments:

Response to Reviewers

We thank the three reviewers for their highly constructive comments on the submitted version of the manuscript. Towards addressing their comments, major new experiments and analyses that are included in the revised version of the manuscript include:

1. The application of DNA Ticker Tape in conjunction with single cell RNA-seq to reconstruct a monophyletic lineage of 3,257 cell. From the resulting data, we document the Poisson-like accumulation of sequential edits to multi-copy tape across 20 generations and 25 days of *in vitro* clonal expansion. Despite its length, this experiment did not exhaust the recording capacity of the system, and it was evident that editing was ongoing even at its conclusion. The completeness of the data matrix (3,257 cells x 13 tapes bearing 59 editable sites, on average 39.4 ± 6.3 of which were edited in one of exactly 19 ways) allowed us to use “off the shelf” methods (UPGMA, NJ), with only minor modifications to take advantage of the ordering information, for phylogenetic reconstruction.
2. The incorporation of several “enhanced prime editing” strategies, which substantially improve the efficiency of writing to DNA Ticker Tape.
3. The demonstration of sequential editing of DNA Ticker Tape in cell types other than HEK293T, including mouse embryonic fibroblasts and mouse embryonic stem cells.
4. The multiplex measurement of the relative editing efficiencies of nearly 2,000 insertion barcodes to DNA Ticker Tape.
5. Additional experiments and/or analyses to characterise the prime editing error frequency, contraction, and expansion of DNA Ticker Tape arrays upon frequent editing, and the impact of ongoing prime editing on the cell proliferation rate.

A point-by-point response, which includes summaries of changes made in the revision response to these comments, is provided below. The original reviewer comments are in **blue text**, while our responses are in **black text**.

Referee #1 (Remarks to the Author):

Choi et al presents a CRISPR-Cas prime editing-based approach to record temporal profiles of molecular and cellular events in mammalian cells, which include presence of plasmids expressing certain prime-editing guide RNAs (pegRNAs) and cellular divisions. This framework, called DNA Ticker Tape, uses prime editing’s capability to insert short DNA sequences to the specific sites of the genome, where the insertion event disrupts the current target site and activates the next target site along the tandem array of partial CRISPR-Cas target sites. The authors first validate the concept by sequential genome editing of 2-5x arrays with a specific prime editing target sequence (TAPE-1). While they successfully show sequential and directional editing of TAPE-1 arrays with modest editing rates (5.3-7.2%) and low error rates (0.07-0.36%), it is unclear whether the sequential genome editing performance is generalizable considering the huge (>10%) error rates for the most other TAPE designs, which is crucial to record multiple events in complex biological contexts.

Thank you for this comment. A key advantage of DNA Ticker Tape relative to most other molecular recorders is that the recording substrate is generic. Biological events are encoded by the separately expressed pegRNAs, all of which compete to write to a shared tape. In the revised manuscript, we demonstrate up to 7-9 bp insertions to DNA Ticker Tape (*i.e.* a 4-6 bp barcode and a 3 bp key), such that as many as 256 (4^4) to 4,096 (4^6) distinct pegRNAs could potentially be used to write distinct event- or signal-specific insertions to a single TAPE-1 (or TAPE-27) array. If additional “memory” was needed, more TAPE-1 arrays could be used, each preceded by a distinct target barcode. In the revised manuscript, we explicitly demonstrate this, by leveraging DNA Ticker Tape in conjunction with single cell RNA-seq, recovering 13 uniquely barcoded tape arrays from each of thousands of cells. We argue that the fact that so many distinct barcodes can be written to a common tape design is a major strength of the method.

As such, while it is accurate that most of the other tape monomers that we tested exhibited substantially worse performance than TAPE-1, this does not preclude us from recording multiple events in complex biological contexts. Rather, only a single, high-performing tape monomer is needed for this goal, which we already have in TAPE-1 (and TAPE-27 as a similarly performing alternative).

Our intention with Fig. 2 of the original manuscript (now **Supplementary Fig. 3**) was to explore whether the performance of TAPE-1 was general. Instead, we found that we had gotten lucky in choosing a tape monomer with strong performance characteristics from the outset. Nonetheless, the fact that most of the other tape monomers that we evaluated do not perform nearly as well does not limit the potential of the method, given that we have a few high-performing monomers in hand.

The authors then demonstrate temporal recording of transfection events of plasmids expressing 16 barcoded pegRNAs. The data provides an exciting possibility to reconstruct temporal profiles of past multiple molecular events in mammalian cells, although the data presented thus far does not yet support their claims for ‘signal’ recording. Next, they used the system to write short texts into the genomes of mammalian cells by transfecting plasmids expressing character-encoded pegRNAs over the course of 5-6 time points. Finally, the authors demonstrated recording of cellular lineage histories using their sequential genome editing framework with constitutively expressed pegRNAs. The ordered nature of DNA Ticker TAPE framework enabled simple reconstruction of cell lineage histories without a need for special computational methods that are necessary for most DSB-based unordered lineage recording systems.

Overall, the paper provides a new strategy of using CRISPR-Cas prime editing to record molecular events into the genomes of mammalian cells with temporal information. The key novelty of the work is using prime editor for development of a molecular recorder and their successful demonstration of temporal recording over the course of relatively longer period than other previous works up to 16 epochs (~48 days). However, the TAPE design with overlapping CRISPR-Cas target sites as a moving DNA writing address is similar to the DOMINO system (Farzadfard et al., Mol Cell 2019), where CRISPR-Cas base editor are used for sequential genome editing-based recording of actual biological signals with much higher efficiencies. Discussion of the uniqueness of the current approach could better highlight novelty.

We thank the reviewer for the suggestion that we better highlight the uniqueness of DNA Ticker Tape in relation to these previously reported molecular recording systems.

DOMINO, as well as CAMERA, utilises base editing (cytidine deaminase fused with Cas9 nickase) to program sequential edits (Farzadfard et al. 2019; Tang and Liu 2018). In each of these methods, an edited product from the first round serves both as the marker of the event occurrence, as well as a target for the next round of editing.

In contrast, with prime editing-mediated insertions to DNA Ticker Tape, the nucleotides encoding the recorded event (*i.e.* the 5' portion of the inserted k-mer) are distinct from the nucleotides that activate the target for the next round of editing (*i.e.* the 3' “key” portion of the inserted k-mer). In other words, with DNA Ticker Tape, the target sequence for the second round of editing does not overlap with the nucleotides encoding the recorded event from the first round of editing. This is a clear difference between DNA Ticker Tape and DOMINO/CAMERA.

This difference becomes particularly important when we consider what would be needed to record the order of 3 or more events. DOMINO and CAMERA allow us to interrogate the order of events with essentially pre-programmed logic circuits (*e.g.* whether we observe A-before-B or B-before-A in each circuit). However, if we were seeking to record the order of more than two kinds of events, the complexity of the necessary logic circuit increases exponentially (*e.g.* for n events, getting the pairwise order among them correct would require n -choose-2 or on the order of n -squared pre-programmed logic circuits).

In contrast, with DNA Ticker Tape, a single circuit (*e.g.* an array of TAPE-1 monomers, together with one pegRNA per signal type) can be used to record the order of many events. Each new kind of signal added to the system requires only one additional pegRNA -- a linear increase in circuit complexity. Furthermore, adding new signal types does not require any changes to the TAPE-1 array.

In the revision, we have sought to now introduce both DOMINO and CAMERA in the Introduction. We also more explicitly discuss the key differences of current CRISPR-based molecular recorders with respect to DNA Ticker Tape in the Discussion and new **Supplementary Table 3**.

Introduction, paragraph 4: “Finally, at least two groups have independently developed “logic-circuit architectures” that use sequential base editing to record the order and identity of biological signals both in bacterial and mammalian cells (DOMINO¹⁹ and CAMERA¹⁴). However, because base editors are currently limited to writing single base substitutions to predefined targets, the order of signals can only be recorded on pre-programmed circuits, rendering multiplex recording challenging.”

Discussion, paragraph 4: “In DOMINO¹⁹ and CAMERA¹⁴, base editors are used to record biological signals to “pre-programmed logic circuits” composed of multiple targets for base editing. Although these methods are conceptual predecessors to DNA Ticker Tape, there are critical differences. In particular, with all three methods, a recording event creates a new target for further editing (*i.e.* the write-head). However, with DOMINO and CAMERA, each logic circuit is designed to record a specific order. In contrast, a single DNA Ticker Tape construct can potentially record any order. For example, to distinguish pairwise orderings within a set of n events, DOMINO or CAMERA would require n -choose-2 recording logic circuits or a system that contains the order of n^2 number of unique gRNA and their targets. In contrast and as demonstrated here (**Figure 2**), DNA Ticker Tape requires only a single target array such as 5xTAPE-1, along with n unique pegRNAs that encode different insertions but share the same target.”

Molecular recording methods	TRACE; Record-seq; Shipman et al.	GESTALT	HOMING; stgRNA	CHYRON	CAMERA; DOMINO	DNA Ticker Tape
DNA editing modality	Cas1-Cas2 integrase	Cas9 nuclease	Cas9 nuclease	Cas9 nuclease and Terminal deoxynucleotidyl Transferase (TdT)	Base editors (e.g. , Cas9 nickase with cytidine deaminase)	Prime editors (e.g. , Cas9 nickase with reverse-transcriptase)
Information content per recording	Up to 28-bp preset spacer sequence (<56 bits)	Stochastic NHEJ/MMEJ DNA repair outcome	Stochastic NHEJ/MMEJ DNA repair outcome	8.4-bp insertion on average (<17 bits)	Up to 2-bp within the target editing window (2 bits)	Up to 6-bp preset insertion sequence (<12 bits)

Possible symbols per target	>1 trillion	1	1	1	1	10-1000
Recording operation	Expansion of array during spacer acquisition	Unordered indels at adjacent targets (del > ins)	Accumulated indels at self-targeting site (del > ins)	Accumulated indels at self-targeting site (ins > del)	5+ sequential single base-pair edits	14+ sequential insertional edits w/ pegRNAs
Limiting factor in multiplexing	Number of spacer precursors	Array size	Number of stgRNAs	Number of stgRNAs	Number of sgRNAs and targets	Number of pegRNAs
Temporal inference	Position within the target array	Shared edits among target arrays	Position within the stgRNA	Position within the stgRNA	Programmed to each target-sgRNA circuit	Position within the target array
Necessary cellular components	Cas1-Cas2, Spacer DNA oligos	Cas9, sgRNAs, target arrays	Cas9, stgRNAs	Cas9, TdT, stgRNAs	Base editor, sgRNAs	Prime editor, pegRNAs, DNA Ticker Tape target
Demonstrated host range	Escherichia coli	Eukaryotic	Eukaryotic	Eukaryotic	Prokaryotic and eukaryotic	Eukaryotic
Key limitations	Portability to eukaryotic cells	Inability to capture the event order; Large deletion on target arrays	Inability to form target arrays; Reduced activity for longer stgRNA	Inability to form target arrays; Reduced activity over time	Limited diversity in editing per target	High multiplicity of integration necessary for highly diverse edits
References	(Shipman et al. 2016; Shipman et al. 2017; Schmidt et al. 2018; Sheth et al. 2017)	(McKenna et al. 2016; Chan et al. 2019; Bowling et al. 2020)	(Perli et al. 2016; Kalhor et al. 2017; Kalhor et al. 2018; Park et al. 2021)	(Loveless et al. 2021)	(Tang and Liu 2018; Farzadfar et al. 2019)	This paper

Supplementary Table 3. Comparison of example CRISPR-based molecular recording methods to DNA Ticker Tape. The overall table structure was adapted from Table 1 of Sheth and Wang (2017).

Finally, we argue that the editing efficiencies demonstrated by DOMINO and CAMERA are similar to what we achieve with DNA Ticker Tape via prime editing. For example, our editing efficiencies using a single transient transfection of PE2 and pegRNAs, allowing for editing over four days, were on the order of 5-10%, without accounting for any inefficiencies in transfection. CAMERA similarly achieves 5-15% editing in mammalian cells, without sequential editing (see Fig. 5C,D of that paper). DOMINO achieves precise sequential editing with an efficiency on the order of 10% after integrating all editing systems to mammalian cells and allowing for prolonged editing over 15 days (see Fig. 5D of that paper). In our revised manuscript, we have tested recently reported approaches to improve the efficiency of prime editing (Nelson et al. 2021; P. J. Chen et al. 2021), which have increased the editing efficiency on DNA Ticker Tape to nearly 20% over four days. As such, the efficiencies of these three methods are now comparable, at least in the mammalian context. We anticipate that prime editing will continue to improve at a fast pace, and any additional improvements in efficiency from contributions across the field can be leveraged to further enhance DNA Ticker Tape performance.

Furthermore, current demonstration of the DNA Ticker Tape technique is fairly underwhelming for their unnecessarily overstated claims on its potential applications and advantages for recording temporal profiles of thousands of distinct biological signals and cellular lineages.

This is a fair point, and we have revised the manuscript to remove or tone down such overstatements. We have furthermore sought to explicitly emphasise that our proof-of-concept demonstrations make use of artificial events (e.g. the transfection of specific pegRNA-expressing plasmid) rather than *bona fide* biological signals.

The authors did not elaborate on describing what they found from the data and just put them on the supplementary tables which are not labeled with details and is therefore not straightforward to digest.

Thank you for pointing this out. In the revision, we have included new supplementary figures that match Supplementary Tables 2-4, in order to make these results more accessible to readers. Supplementary Tables 2 and 3 have been converted to **Supplementary Figure 4**. The experiment previously represented in

Supplementary Table 4 and 5 is no longer in the paper, but the equivalent information for the more extensive lineage tracing experiment that has replaced it is now provided in **Supplementary Files 1-2**.

Nevertheless, this CRISPR-Cas prime editing-based temporal recording is likely to be an exciting foundation towards a highly efficient and programmable platform for temporal biological recording and lineage tracing in mammalian cells and potentially multicellular organisms.

We thank the reviewer for these positive comments on the foundational potential of DNA Ticker Tape for recording biological signals and cell lineage in mammalian systems.

Several points should be addressed in the current version of the manuscript.

Major comments:

1. The authors' first TAPE-1 construct happens to be with one of the lowest error rates among the 48 TAPE constructs characterized later (Figure 2). Interestingly, so many other TAPE designs show very high error rates (fraction of edited reads inconsistent with sequential and directional genome editing) with some of them even reaching above 80% that it does not seem to be generalizable at all to various target sequences.

As discussed above, we agree that the clear majority of tape monomers that we evaluated exhibited poor performance characteristics, and in retrospect we got quite lucky with TAPE-1 (although we note that TAPE-27 has similar if not better performance characteristics, so the screen was worthwhile). However, as also discussed above, only a single, high-performing TAPE monomer is needed for DNA Ticker Tape to be useful for recording complex event histories, as thousands of distinct signal types can be written to this shared TAPE composed of a high-performing monomer. Additional memory can be realised via longer tape or additional instances of the same high-performing TAPE design.

While they mention that the length of key sequences in the TAPE sites is associated with error rate, it is unclear what other factors are affecting this huge range of error rates across different TAPE sequences. Is this just something intrinsic to CRISPR-Cas9 target specificity or prime editing specificity for different target sequences and mismatches?

We agree that the 2-3 orders of magnitude of variation in specificity is surprising. However, a simple explanation is that some spacer sequences are much more tolerant of "PAM-distal" mismatches, as has been documented for both conventional CRISPR/Cas9 genome editing (Hsu et al. 2013; Cho et al. 2014) and prime editing (Kim et al. 2020), resulting in errors when 5'-truncated TAPE monomers (*i.e.* lacking the correct key sequences) are prime edited despite not being the position of the "write head".

Can TAPE performance be predicted based on sequence of the target sites and can highly efficient TAPE designs with lower error rates be forward designed for multiplex recording of biological events across many different TAPEs?

It is challenging to make generalisations with the limited number of basal TAPE sequences that we tested. However, given the success of sequence-based prediction on Cas9 target specificity and efficiency (Hsu et al. 2013; Doench et al. 2016) from models trained on thousands of spacers, it is entirely plausible that TAPE performance could similarly be predicted based on the monomer sequence. We have added this point to the end

of the Results section subtitled **“Screening additional monomers for their performance as DNA Ticker Tape”**, which now reads:

“Overall, these results show considerable variation in efficiencies and error rates, specific to particular 13- to 15-bp TAPE sequences. Although a single well-performing monomer such as either TAPE-1 or TAPE-27 is sufficient to construct a generic substrate to which thousands of distinct symbols can be written, additional screening might yield monomers with even better performance characteristics, and would also facilitating modelling of the sequence determinants of monomer performance^{25-27,29”}

2. Also, it seems that even very small variations (2-3 bps) in the barcode sequences within the pegRNA sequences can affect prime editing efficiencies significantly (Supplementary Figure 1). Probably for this reason, when the authors record presence of two different barcodes at different ratios over time for the temporal transfection programs 4 and 5 in Figure 3 to ask whether the relative strength of signals could be inferred from DNA Ticker Tape, only the relative ratios between the programs 4 and 5 are compared (Figure 3h), not between inferred and actual ratios of barcodes for each epoch within each program. This is only described in the Supplementary Table 3, which is not labeled with details.

We thank the reviewer for pointing this out, which made us realise that we should expand our presentation of this aspect of the data. We have added a new **Supplementary Figure 4** that graphically shows the data previously contained in Supplementary Tables 2-3. In this figure, we present the barcode counts before and after the correction using previously measured editing efficiencies, and how barcode ratios calculated with the corrected counts accurately capture the intended barcode ratios within the transfection. The relevant parts of this new figure are recapitulated below:

Supplementary Figure 4c. For each of the five transfection programs, the event orders are inferred using “Unigram” (top) and “Bigram” (bottom) information.

Supplementary Figure 4e,f. For Program-4 (e) and Program-5 (c), the absolute barcode read counts (left) are corrected based on the edit score of 16 NNGGA barcodes (middle), and used to calculate the relative magnitude of two co-transfected barcodes (right). The expected barcode ratios are marked with a red 'X' mark in each epoch.

In addition, we appended an additional panel to Figure 1 (new **Figure 1g**) to explicitly show the edit scores for all possible 16 barcodes tested in 5-bp insertion on DNA Ticker Tape:

Figure 1g. Edit score of 16 barcodes used in the experiment with 5xTAPE-1. Edit scores for each insertion are calculated as \log_2 of the ratio between insertion frequencies and the abundances of pegRNAs in the plasmid pool, averaged over 3 transfection replicates.

I suspect that the system may not be very accurate at recording relative strength of signals, probably due to the confounding effect of barcode sequences on prime editing efficiencies. This necessitates characterization of prime editing efficiencies for all barcoded pegRNAs beforehand for multiplex signal recording, limiting the scalability of the system. The authors should temper these overstated claims on scalability (or provide evidence to support these claims), elaborate on the advantages and limitations of the system, and plot the data they have in supplementary tables whenever possible.

As the reviewer points out, characterization of prime editing efficiencies for all barcoded pegRNAs beforehand to obtain edit scores/correction factors is necessary. However, we would like to highlight that characterising

prime editing efficiencies for barcoded pegRNAs is not difficult. For example, in the submitted manuscript, we characterised edit scores for 16 NNGGA and 64 NNNGGA insertions to the TAPE-1 construct. We expect that a single experiment is sufficient to measure the edit score of each barcode, as we have observed high correlation between two replicates using different pools of pegRNA plasmids (Spearman's ρ of 0.97; **Supplementary Figure 1e**) as well as in different cell lines (**Supplementary Figure 2k,i**).

In a new experiment, while using further optimised prime editing reagents (*i.e.* epegRNA / PEmax / MLH1dn), we pushed this further by measuring as many as 1,900 edit scores/correction factors in a single experiment using NNNNNNGGA (*i.e.* 6N + GGA key) insertions (new **Supplementary Figure 2f-i**). Briefly, we retained 6N+GGA insertion and plasmid reads that are above the thresholds, determined using the 'knee-plot' shown in the **Supplementary Figure 2f-g**, and selected 1,908 insertions that were sufficiently covered in all three replicates of editing experiment as well as the sequencing of the plasmid pool.

Therefore, we argue that the recording system based on DNA Ticker Tape is scalable, at least in terms of the number of barcodes that can be concurrently used in a single experiment, as these correction factors can be pre-determined (and indeed, already have been for thousands of barcoded pegRNAs through this experiment).

Supplementary Figure 2. Improved prime editing on the 5xTAPE-1 DNA Ticker Tape construct. **f.** Knee plot of read-counts for 4,096 possible 6N+GGA insertions, across three replicates. A minimum threshold of requiring at least 20 reads for a given insertion in each of the three transfection replicates was determined based on this plot. **g.** Knee plot of read-counts for 4,096 possible 6N+GGA-inserting pegRNAs from the pool of plasmids. A minimum threshold of 30 reads for each insertion plasmid was determined based on this plot. **h.** Edit scores for 1,908 6N+GGA insertions. Only insertions that appeared more than 20 reads in each of three transfection replicates and more than 30 reads in the sequencing of the plasmid pool were considered. Edit scores for each insertion are calculated as \log_2 of the ratio between insertion frequencies and the abundances of pegRNAs in the plasmid pool. **i.** Top 25 edit scores for 6N+GGA insertions.

3. While the authors claim capability of signal recording throughout the text, what they experimentally demonstrate is constitutively expressed pegRNAs on the transiently transfected plasmids. While the results in

Figure 3 show that this is on the correct path, but the next set of advances are really what is going to push this work to a regime that is biologically useful. It is not clear why text messages and not biological signals were chosen as a demonstration of recording for Figure 4. Could experiments be done for cells to sense and record biological signals over time using several inducible promoters for expression of barcoded pegRNAs? Without such demonstrations, the authors should not describe their transfection events as signaling events and the claim for biological 'signal' recording should be revised.

As also noted above, we agree with the reviewer that our demonstration with transfection of pegRNAs is not a *bona fide* biological signalling event. We have revised 'biological signal' to '(biological) event' or related verbiage throughout the main text to describe the use of transfection in our demonstration (revised texts are underlined).

Abstract: "Recording is mediated by the activity of specific prime editing guide RNAs at the DNA Ticker Tape²."

Page 2: "Here we describe a DNA-based memory device that is: (1) highly multiplexable, *i.e.* compatible with the concurrent recording of at least thousands of distinct event types;"

Page 13: "At the beginning of each epoch of each transfection program, one or more pegRNA plasmids were introduced to a population of HEK293T cells with integrated 5xTAPE-1 (5xTAPE-1(+) HEK293T) via transient transfection of plasmids expressing the corresponding pegRNA(s) and PE2."

Page 13: "Programs 1 and 2 each consisted of a distinct, non-repeating sequence of transfection of the 16 pegRNAs, *i.e.* one per epoch."

Figure 2 legend: "**Figure 2. Transfection programs for 16 sequential epochs. a.** Schematic of five transfection programs over 8 or 16 epochs. For Programs 1 and 2, pegRNAs with single barcodes were introduced in each epoch for 16 epochs. The specific orders aimed to maximise (Program-1) or minimise (Program-2) the edit distances between temporally adjacent transfections."

Relevant to this comment, in parallel to DNA Ticker Tape, we have developed a prime editing-based strategy for cells to sense and record biological signals over time using enhancer or signal-responsive enhancers that drive Pol-2-mediated expression of barcoded pegRNAs. With this method, termed ENGRAM (ENhancer-driven Genomic Recording of transcriptional Activity in Multiplex), we demonstrate concurrent genomic recording of the relative activity of at least hundreds of enhancers, as well as time- and concentration-dependent genomic recording of Wnt, NF- κ B, and Tet-On activity. The preprint describing the ENGRAM method is available at the link below and is now cited in the Discussion.

<https://www.biorxiv.org/content/10.1101/2021.11.05.467434v1>

In future work, we anticipate integrating ENGRAM and DNA Ticker Tape.

4. For text message recording experiments, the character deletion errors are mostly due to repeated use of pegRNA barcodes for repeated characters. Why does this still happen when the authors tried to avoid this error mode by using two different barcode sequences alternatively for the same characters? This error mode needs to be discussed more in details as it is relevant to the potential performance of the system for recording actual biological signals that fluctuate and repeat constantly.

We believe that the deletion errors occur when the same character occurs more than twice. For example, there are four 'H' in the short text 'WHAT HATH GOD WROUGHT?'. Therefore, the two 3-bp barcodes assigned to 'H' -- 'ACT' and 'GAC' -- are each being used twice in the text. Similarly, there are three 'T' in the same text, such

that the 'CAT' assigned to 'T' has to be used twice and gives rise to a deletion error. We believe that these errors could be avoided by using longer barcoded characters to assign more barcodes to specific characters used in the text. For example, use of 6-bp barcodes, as done in the **Supplementary Figure 2**, would allow 4096 barcodes to be assigned to 31 characters in our text encoding scheme, such that each character could be used >100 times before this issue was encountered. We have modified the main text accordingly to include this interpretation (added text is underlined):

"From the first message, 17/22 characters were correctly recovered and ordered, with three deletion errors and one swap between adjacent characters to yield "WA HATH GOD WRUOGT?" (Figure 3c). Of note, the deletion errors were due to repeated use of pegRNA barcodes 'ACT', 'CAT', and 'GAC' to encode multiple 'H' or 'T' characters, and as such were not expected to be recovered separately. These deletion errors are the result of our encoding scheme which used only 64 unique pegRNAs; we anticipate that greater information content per edit can be achieved with pegRNAs with longer barcodes, e.g. 6-bp barcodes would have allowed each instance of repeated characters to be represented by different insertions, thereby avoiding this kind of error."

We agree with the reviewer that the actual biological signals may fluctuate and repeat constantly. We speculate that, if actual biological signals fluctuate faster than the temporal resolution of prime editing, we would observe overlap of signals as in Programs 3-5 in **Figure 2**.

5. For lineage recording experiment, the starting single cell contains 23 inserted TAPE-1 arrays with 23 barcoded pegRNAs cassettes in its genome and all the arrays are being independently edited by all the 23 pegRNAs over the course of 32 days. The authors' current analysis for some of the selected individual TAPE-1 arrays show potential utility of the system to track cellular lineages by simply ordering of the barcodes inserted into the arrays. However, analysis on individual TAPE-1 arrays is not sufficient for their experimental design and the whole 23 different TAPE-1 arrays together should be considered for constructing cellular lineages. Perhaps all these TAPE-1 arrays could be linked in single cells? Alternatively better defined experiments could be done with single TAPE-1 array with multiple barcoded pegRNA cassettes integrated into the genome of the starting single cell.

We thank the reviewer for this suggestion. As suggested, we sought to repeat the lineage tracing experiment in conjunction with single cell RNA-seq, as suggested by the reviewer. Specifically, we generated a monoclonal cell line with at least 13 independently inserted TargetBC-TAPE-1 arrays, edited over 25 days. These arrays were expressed as RNAs within each cell, carrying a capture sequence compatible with 10X Genomics' single-cell target-capture assay. In this new experiment, we demonstrate that we can efficiently capture at least 13 TargetBC-TAPE-1 arrays per cell, and use information across different TAPE-1 arrays to build a lineage tree. Specifically, 99.4% of resulting cells have a unique editing pattern across 59 TAPE editing sites, which we then use to build a monophyletic tree from the multi-target profiles of 3,257 resulting cells. We also show that conventional phylogenetic algorithms can be applied largely "off the shelf" (*i.e.* with only minor modifications to take advantage of the ordering information provided by DNA Ticker Tape) for lineage reconstruction.

Because this new experiment constitutes such a major part of the revised paper (entirely replacing the previous lineage tracing experiment), we do not recapitulate its results in their entirety here. Please see **Figure 4**, **Supplementary Figure 6**, and corresponding portions of the text, as well as **Supplementary Files 1-2**.

Again, the authors just put most of the data for this experiment in Supplementary Table 4. It would be informative to plot and describe how all 23 individual barcoded TAPE-1 array populations behaved during the 32 days of cell divisions.

Thank you for this suggestion, and we agree that the data would be better shown graphically. As noted above, the referenced experiment has been replaced in the manuscript with the new, single cell experiment described immediately above. The entire monophyletic tree derived based on editing patterns across 59 sites (editable positions within 13 TargetBC-TAPE-1 arrays) in each of 3,257 cells is shown in **Fig. 4f** and **Supplementary File 1**, with accompanying data table as **Supplementary File 2**. The specific request here is perhaps no longer relevant, because the referenced experiment has been removed from the revised manuscript, and the new phylogenetic trees integrate information from all 13 TargetBC-TAPE-1 arrays.

Minor comments:

6. The authors may want to place their work in the correct context of other works. While many molecular recording techniques for mammalian cells records signals stochastically to unordered target sites, several recent approaches (Tang and Liu, *Science* 2018; Farzardfard et al., *Mol Cell* 2019; Loveless et al., *Nat Chem Biol* 2021, etc) write biological information in sequential and unidirectional manner. These techniques should be discussed and compared with DNA Ticker Tape more critically. Multiplexability is the key advantage of prime editing-based approach over the previous approaches, which could be emphasized.

Please see our response to your second general comment (pages 2-4 above). In brief, we have sought to do a better job contextualising the advance represented by DNA Ticker Tape in relation to previous reports of molecular recorders that write in a sequential, unidirectional manner (revised introduction and discussion, as well as new **Supplementary Table 3**).

How many similar pegRNA cassettes with only several mismatches can practically implemented in the living cells? How stable are they in the genome? What are needed to address these potential limitations?

We have conducted experiments in which 23 (submitted paper) or 19 (revised paper; estimated MOI is 19, and we used 13 highly-expressed TargetBC-5xTAPE-1 arrays) nearly identical pegRNA cassettes (identical 5xTAPE-1, but differing in the adjacent TargetBC) were successfully integrated to a clonal mammalian cell line, in both cases via lentivirus-mediated random integration. Previous work by Kahlor *et al.* showed that as many as 60 nearly identical gRNA-expressing cassettes could be introduced via multiplex piggyBAC integration to mouse embryonic stem cells, which then went on to produce healthy mice (Kahlor et al. 2018). As such, the answer is at least dozens, and potentially even 100 or more with some effort.

In our view, the greater concern is the stability of individual TAPE-1 arrays, particularly when they are much longer than 5 monomeric units. As shown in **Supplementary Figure 7**, we did observe some variation in the length of longer repeats, albeit in a non-clonal experiment. Some of this variation is probably attributable to 'slippage' of what is essentially a synthetic minisatellite:

"This experiment illustrates that it is possible to construct and use synthetic minisatellites corresponding to at least 20 monomers as DNA Ticker Tape, and that sequential recording of at least 14 consecutive events is possible. Nonetheless, further experiments are required to quantify the extent to which variation in synthetic minisatellite length is due to: (1) piggyBAC vector heterogeneity, *i.e.* variation that existed prior to integration; (2) DNA replication and microsatellite instability in HEK293T cells; (3) DNA repair subsequent to prime editing-induced nicks; and/or (4) PCR amplification artefacts. Of note, the observed variation in array length tended to occur within the unedited portion of the tape (**Supplementary Figure 7g,h**). We have yet to observe any clear examples of "information erasure", possibly because the edits themselves disrupt the tandem repeats, inhibiting processes that might otherwise lead to erasure from spreading proximal to the write-head."

That being said, the HEK293T cell line in which we conducted these experiments is MMR-deficient, so much so that HEK293T cells can be difficult to authenticate via short tandem repeat (STR) analysis. In MMR-competent cell lines as well as *in vivo*, the vast majority of mammalian minisatellites are very stable (Vergnaud and Denoeud 2000), such that we do not anticipate that this will be a major issue (or at least should be no worse). In the worst case, we could simply rely on larger numbers of independently integrated 5xTAPE-1 constructs, as we successfully do in the new *in vitro* lineage tracing experiments, as we have found these to be sufficiently stable over extended periods of time, even in MMR-deficient HEK293T cells (see also, new **Supplementary Figure 4b**).

7. Again, columns in most supplementary tables should be relabeled clearly (or put notes to describe what each column is showing) and it would be nice to plot the data whenever possible.

Thank you for this suggestion. As discussed above, we have expanded the supplementary figures to graphically present data previously provided in the supplementary tables. Supplementary Tables 2 and 3 have been converted to **Supplementary Figure 4**. The experiment previously represented in Supplementary Table 4 and 5 is no longer in the paper, but the equivalent information for the more extensive lineage tracing experiment that has replaced it is now provided in **Supplementary Files 1-2**.

8. For Figure 1, it seems they did 3 replicate experiments. It would be nice to include standard deviations for the numbers in Figure 1.

Good point. Following the reviewer's suggestion, we now include standard deviations for all the numbers in **Figure 1d-f**.

Revised **Figure 1d-f**, including standard deviation from three transfection replicates.

9. It would be informative to see how many sequencing reads are required for correct ordering of temporal profiles (Figure 3) or correct calling of text encoded (Figure 4).

This is a terrific suggestion. To characterize the robustness of order inference, we undersampled the sequencing reads used to reconstruct the order of 16 transfection events in Program-1. In **Figure 2c** (previously Figure 3c),

277,397 5xTAPE-1 amplicon reads from a single MiSeq sequencing run (2.67% of overall sequencing run using Illumina MiSeq Reagent kit v3) were used to reconstruct the bigram transition matrix. We undersampled the sequencing reads to 10,000 reads, 2,500 reads, 2,000 reads, 1,500 reads, and 1,000 reads, and plotted both bigram transition matrix and the inferred order for each sampling point (now **Supplementary Figure 4d**). We observed that both the bigram transition matrices and the ordering of transfection events were robust to undersampling. The correct order could be inferred from the 2,500 reads, and started to break down below 2,000 reads.

Supplementary Figure 4d. Undersampling analysis of Program-1. From the original 277,397 sequencing reads used for Program-1, we undersampled to 10,000, 2,500, 2,000, 1,500, or 1,000 reads. For each sampling point, the bigram transition matrix (top) was plotted and order of events (bottom) were inferred using bigram information.

Supplementary Figure 5b. Undersampling analysis of the short text “WHAT HATH GOD WROUGHT?”. From the original 1,256,996 sequencing used in **Figure 3**, we undersampled to 1,000,000, 100,000, 10,000, and 5,000 reads. For each sampling point, the bigram transition matrix (top), the corrected unigram counts (middle), and the hierarchical clustering (bottom) were plotted. From these, the original short text was inferred at the end.

Referee #2 (Remarks to the Author):

Summary: Here, Choi et al. create and validate a molecular construct designed to be targeted by Cas9 in an ordered manner by leveraging insertion events created via prime editing. In a simple statement, this process has one active Cas9 target that perfectly matches the protospacer, and when Cas9 binds and nicks the DNA, prime editing from the Cas9-RT fusion (prime editing enzyme) templates the pegRNA to insert sequence at the cut site. This sequence insertion “moves” the active Cas9 target down one site – effectively respacing the full target and enabling a new PAM site within the inactive third site to be used for the next editing reaction. Choi and colleagues demonstrate effective and elegant proof-of-principle experiments for their encoding strategy. These experiments determine the efficiency of prime editing to occur sequentially and the accuracy of it as an information encoder. While the rate of editing is quite low, the subset of TAPes that are edited do provide proof-of-concept. Throughout the manuscript, they demonstrate its efficacy as a transient transfection, transposon-mediated insertions, and as a lentiviral system, which indicates it could be broadly utilized across many types of cell types and in vitro experiments that are amenable to these techniques. As an application of this technology, Choi et al. employ an in vitro lineage tracing experiment, demonstrating their ability to use this as a lentiviral system for tracking lineage relationships from a single cell over time. Overall, although this is an exciting system, the manuscript as presented covers two distinct applications of this approach, it would be ideal to gain some biological insight from these applications.

We thank the reviewer for these positive comments.

Major Comments:

1. Choi and colleagues demonstrate a truly elegant idea, with a solid set of validation experiments for this technique as a data storage device (Figs 1 – 4), however, the proof of concept experiments rely on relatively ‘clean’ gene expression, raising questions on how well this approach would work in a real, noisy biological context. Moreover, we are curious how this method would generally perform in a less robust cell line. Since HEK cells are very resilient, they appear to be well suited to a DNA-based assay with many rounds of passaging, transfections, and/or long-term dox treatments. This does, however, raise questions of the broad applicability of this approach to other cells – particularly primary cells. The authors may consider deploying their method to a well-characterized cell differentiation paradigm to confirm known events or reveal new insights.

The reviewer is correct to point out that HEK293T is indeed a very robust cell line, and used in many studies for “proof-of-concept” demonstrations of genome editing-based technologies, in part because of this robustness. Prime editing in general has been shown to work in a variety of cell lines, including immortalised cell lines such as HEK293T and K562, primary cell lines such as T cells and mouse cortical neurons, as well as mouse and human stem cells (Liu et al. 2020; Schene et al. 2020; P. J. Chen et al. 2021). Nonetheless, although we are not overly worried, we recognize the reviewer’s point that it would add value to the story to demonstrate the method in contexts other than HEK293T cells. In response to this comment and because it is most in line with our future plans in terms of applying the method, we tested DNA Ticker Tape in mouse embryonic fibroblasts (MEFs) and mouse embryonic stem cells (mESCs):

Supplementary Figure 2. Enhanced prime editing of DNA Ticker Tape in MEFs and mESCs. j. Editing efficiencies at the first site of 5xTAPE-1 integrated in the mouse embryonic fibroblasts (MEFs) or mouse embryonic stem cells (mESCs). For mESCs, up to two sequential transfections of a pool of epegRNA-expressing plasmids were tested. The error bars are standard deviations from 3 transfection replicates.

Leveraging a combination of improvements to prime editing including epegRNAs, PEmax, and human MLH1 dominant negative fragments (MLH1dn), we achieved high (10-20%) editing efficiency with a single-round of epegRNA/PEmax/MLH1dn transfection, similar to what we observe in HEK293Ts using the same combination of improvements (**Supplementary Figure 2a**). We also note that two sequential rounds of transfections increase the editing efficiency in mESCs, supporting our previous observation that prolonged expression of prime editing machinery can increase the editing efficiency of DNA Ticker Tape.

2. The authors comment on the possibility of recording sequential multiplexed signals. However, again due to the low editing rate, the practicality of this seems questionable without some kind of demonstration. Even with a dual cassette, would it not be difficult to parse out cells that received a signal but failed to accrue edits vs. those that received both (the effects of which would be compounded after each signaling epoch)? We assume the authors intend to capture information about single cells.

The reviewer's concerns regarding the low editing rate are valid, and directly related to a current limitation of the prime editing in general. However, even since our original submission, there have been several independent demonstrations of improvements to the efficiency of prime editing via clever engineering, as we describe further in our response to comment #7 from this reviewer below. We imagine that such efforts to improve prime editing efficiency will continue, as was the case for base editing (Richter et al. 2020; Koblan et al. 2018). Improvements in the prime editing efficiency would ideally be sufficient to quantify the strength and order of biologically-relevant signals at high temporal resolution within single cells.

Nonetheless, we expect that the primary use case of DNA Ticker Tape in signal ordering during biological processes such as development would not be strictly dependent on this. During development of most animals, e.g. mouse or zebrafish, many cells comprise each cell type. As such, ordering signalling events may be achieved from the ensemble of DNA Ticker Tape recordings within cells of each type, analogous to how we ordered the 16 sequential transfections by integrating information obtained across many cells. In the long view, we anticipate that collection of both signalling and lineage information will allow one to reconstruct a lineage tree that is "decorated" with the order/strength of various signals based on ensembles of cells of each type.

Finally, we would like to point out that the recording capacity of the system can be increased by integrating a dozen or more independently operational DNA Ticker Tapes to the genome of each cell, and then recovering these sets as part of a single cell RNA-seq profile. In response to the next comment, we now demonstrate this in the context of lineage reconstruction.

3. The lineage tracing portion of the manuscript requires further development. No biological insights are provided, though the authors indicate that it could be performed via interrogation of sequential signals that are linked to pegRNA expression. Indeed, further biological interrogation is necessary to demonstrate the ability of this system to record valuable information over time. Though the lineage tracing experiment is intriguing, it fails to demonstrate the complexity of a lineage tracing system or create a lineage tree from all of the TAPes within the monoclonal cell population. Further experiments to deduce the complexity of labeling are also necessary to determine the efficacy of the system to individually label cells in a unique manner, increasing the resolution of the tree.

We thank the reviewer for this suggestion. As suggested, we sought to repeat the lineage tracing experiment in conjunction with single cell RNA-seq, as suggested by the reviewer. Specifically, we generated a monoclonal cell line with at least 13 independently inserted TargetBC-TAPE-1 arrays, edited over 25 days. These arrays were expressed as RNAs within each cell, carrying a capture sequence compatible with 10X Genomics' single-cell target-capture assay. In this new experiment, we demonstrate that we can efficiently capture at least 13 TargetBC-TAPE-1 arrays per cell, and use information across different TAPE-1 arrays to build a lineage tree. Specifically, 99.4% of resulting cells have a unique editing pattern across 59 TAPE editing sites, which we then use to build a monophyletic tree from the multi-target profiles of 3,257 resulting cells. We also show that conventional phylogenetic algorithms can be applied largely "off the shelf" (*i.e.* with only minor modifications to take advantage of the ordering information provided by DNA Ticker Tape) for lineage reconstruction.

Because this new experiment constitutes such a major part of the revised paper (entirely replacing the previous lineage tracing experiment), we do not recapitulate its results in their entirety here. Please see **Figure 4**, **Supplementary Figure 6**, and corresponding portions of the text, as well as **Supplementary Files 1-2**.

In our admittedly biased view, if one considers the completeness of information available for this set of 3,257 cells (59 sites altogether, each edit drawn from one of 19 possibilities, and temporally ordered editing at each group of up to 5 sites), the Poisson-like rate at which edits appear to accumulate, and the fact that we do not exhaust the recording capacity of the system after 20 generations and 25 days of continuous editing, we believe that this experiment represents a major leap forward in terms of the prospects for high-resolution CRISPR-based lineage tracing.

The reviewer is correct that we do not demonstrate biological recording here (e.g. signals beyond stochastic information useful for lineage). However, in parallel to DNA Ticker Tape, we have developed a prime editing-based strategy for cells to sense and record biological signals over time using enhancer or signal-responsive enhancers that drive Pol-2-mediated expression of barcoded pegRNAs. With this method, termed ENGRAM (ENhancer-driven Genomic Recording of transcriptional Activity in Multiplex), we demonstrate concurrent genomic recording of the relative activity of at least hundreds of enhancers, as well as time- and concentration-dependent genomic recording of Wnt, NF- κ B, and Tet-On activity. The preprint describing the ENGRAM method is available at the link below and is now cited in the Discussion.

<https://www.biorxiv.org/content/10.1101/2021.11.05.467434v1>

In future work, we anticipate integrating ENGRAM and DNA Ticker Tape.

4. With regards to the lineage tracing application, a “clone mixing” experiment of multiple monoclonal populations to determine if multiple unique trees can be determined would go a long way to demonstrating the efficacy of this as a single cell lineage tracing system. Provided this approach is worthwhile, understanding cell states that are derived from lineage identities would be useful as interrogated biological information from the lineage, as previous publications, such as GESTALT or scGESTALT have done.

Following the reviewer’s suggestion, we performed a new “polyphyletic” experiment in which multiple monoclonal lines (five to ten) were seeded to a single well, and the diversity in editing patterns of the DNA Ticker Tape used to identify clonal identity of each TAPE arrays within oligoclonal pool that resulted. We cultured the resulting oligoclonal cell line for 20 days until we had about 1,000,000 cells, and then used a sc-RNA-seq to recover the TargetBC-associated and InsertBC-decorated 5xTAPE-1 array in each cell. We assessed the read-count of each TargetBCs to select the top 42 TargetBCs (**Revision Figure 1a**); We purposefully set the threshold to capture five TargetBCs that are one hamming-distance away from the most frequent TargetBC (CCATTATA), which are likely to have been a result of PCR/sequencing error on TargetBCs. We then used hierarchical clustering to put these 42 TargetBCs into 6 groups based on their shared appearance within the same cell (**Revision Figure 1c**). If our clustering is indeed capturing the true grouping of integrants in multiple clonally-derived subsets of cells, we would expect that a similar set of InsertBCs would appear in the 5xTAPE-1 associated with the same group of TargetBCs. We counted the appearance of the top 30 InsertBCs to 5xTAPE-1 associated with the particular TargetBCs (**Revision Figure 1b**), and used it to cluster TargetBCs. As predicted, we observed a similar grouping of TargetBCs (**Revision Figure 1d**), suggesting that TargetBCs and InsertBCs consistently reveal the sets of integration events that define the half-dozen monoclonal lines seeded at the start of this experiment.

Revision Figure 1. **a.** Frequencies of TargetBCs observed in all reads. The top 42 most frequent TargetBCs (threshold denoted with the grey dotted line) were used in the downstream analysis. **b.** Read counts of InsertBC observed in TAPE-1 arrays. The top 30 most frequent insertBCs (threshold denoted with the grey dotted line) were used in the downstream analysis. **c.** The 42 TargetBCs were clustered into 6 groups based on their co-recovery within the same cell. Complete linkage hierarchical clustering algorithm was used. **d.** The 42 TargetBCs were clustered based on InsertBC read counts from the associated 5xTAPE-1 arrays. Complete linkage hierarchical clustering algorithm was used. The colouring of TargetBC follows the grouping on the panel (c).

This experiment was conducted in parallel to the new monophyletic DNA Ticker Tape / sc-RNA-seq experiment described above and included in the revised manuscript. While we observed a tight link between the repeated InsertBC barcode patterns in recovering the clonal information (related to Comment #20 below; **Revision Figure 1c,d**), we decided to leave the polyphyletic experiment out of the main manuscript, because we believe that the new monophyletic experiment ended up providing a much more extensive demonstration of the potential of DNA Ticker Tape for high-resolution, continuous lineage recording. That being said, we would be happy to add this data into the paper if the reviewer feels that it adds substantial value above and beyond the new monophyletic experiment.

5. The example of lineage recording is interesting and shows potential, but it is ultimately incomplete. The authors validate their ability to partition each original TargetBC by InsertBCs, presenting a strong foundation for lineage analyses. However, they do not present a feasible means to unify this data into a collective lineage tree, which is presumably the final goal? Without being able to identify individual cells within the monoclonal population, it is unclear how the authors propose consolidating the information across 23 TargetBCs from a single clone into a

composite tree (or how they would leverage the lineage information after constructing such a tree?). Scaling this up to more than one clone would seem to present multiple challenges as well. Some clarification of this from the authors would be helpful.

These are good points. As described above, in response to this and related comments, we now show how we can use single cell RNA-seq to unify data from over a dozen independently integrated DNA Ticker Tapes and construct a single, monophyletic lineage tree (see response to comment #3 above). We also perform and report on a polyclonal version of this experiment, also relying on single cell RNA-seq to integrate information from multiple DNA Ticker Tapes per cell (see response to comment #4 above).

Specific Comments:

6. Page 2, last pp: This is an excellent description of the system that is implemented. However, the associated Figure 1a could be improved. The inclusion of a more detailed drawing that could include the function of the PE enzyme, how it integrates the barcode+key sequence and moves the write head would be more effective for the reader.

Following the reviewer’s suggestion here as well as comments #9 and #22, we revised **Figure 1a-c**. Briefly, we now draw the first two rounds of editing on DNA Ticker Tape, annotating the movement of the write head following the insertion of the barcode+key sequences. In the new **Figure 1b**, we have drawn the function of the PE enzyme and steps involved in a single round of prime editing more explicitly. We welcome any additional feedback on how this figure could be further improved.

Figure 1a-c. Sequential genome editing with DNA Ticker Tape. **a.** Schematic of editing event at the “write head” of the DNA Ticker Tape. The DNA Ticker Tape consists of a tandem array of CRISPR-Cas9 target sites (grey boxes), all but the first of which are truncated at their 5’ ends, and therefore inactive. The 5-bp insertion includes a 2-bp pegRNA-specific barcode as well as a 3-bp key that activates the next monomer. Because genome editing is sequential in this scheme, the temporal order of recorded events can simply be read out by their physical order along the array. **b.** The schematic of prime editing during one round of recording. Prime editing recognizes a CRISPR-Cas9 target and modifies it with the edit specified on the pegRNA molecule. In DNA Ticker Tape, an

insertional editing event generates a new prime editing target for the sequential editing. **c.** Schematic of recording the event order on DNA Ticker Tape. Individual pegRNAs are either event-driven²² or constitutively expressed, together with the PE2 enzyme.

7. Page 3, pp3: “Overall editing rates were modest, as only 5.3%, 7.2%, and 6.6% of all reads for 2xTAPE-1, 3xTAPE-1, and xTAPE-1, respectively, exhibited any editing.” These editing rates are low – can the efficiency be increased? If the utility of this becomes an information encoding process, it would be better to make it more efficient to demarcate past events in cells to increase the likelihood of capture in final cells of interest – especially with rare cell types. Further, with lineage tree construction, is there a reason that some cells may be biased to recording while others are not, as this would skew how effectively the labeling may occur in specific populations?

We fully agree that boosting editing efficiency is an important goal for realising the potential of the method. One avenue will be to identify TAPE monomers that are more efficient than TAPE-1 or TAPE-27 while maintaining high specificity. As illustrated by our preliminary efforts in this direction, this may be challenging.

A second avenue is to increase the efficiency of prime editing. Indeed, we note that this is an active area of research for the field, with significant progress since even the time that we submitted this manuscript. For example, as very recently shown, inhibiting pegRNA degradation by RNA engineering (referred to as ‘epegRNAs’ (Nelson et al. 2021)) can enhance prime editing efficiency. In addition, efficiency improvements may be possible through inhibition of mismatch repair, as recently shown independently by several groups. Similar to epegRNAs, because this approach impacts the efficiency of the prime editing enzyme, we anticipated that it would translate seamlessly to improving DNA Ticker Tape efficiency as well.

Indeed, when we applied combinations of epegRNA, PEmax, and inhibition of mismatch repair pathway via human MLH1 dominant-negative peptide to the DNA Ticker Tape, we also observed a nearly 2-3-fold increase in the editing efficiency, from ~6% to ~20% editing efficiency after a single round of transfection:

Supplementary Figure 2. Improved prime editing on the 5xTAPE-1 DNA Ticker Tape construct. **a.** Editing efficiencies on the first site of 5xTAPE-1 integrated in the HEK293T cells. A pool of plasmids expressing TAPE-1 targeting epegRNAs were transfected with the pCMV-PEmax-P2A-hMLH1dn plasmid. Five pools with different insertion lengths ranging from 5-bp (NNGGA) to 9-bp (NNNNNNGGA or 6N+GGA) were tested separately. The error bars are standard deviations from 3 transfection replicates.

A more general point is that there is a growing field of labs focused on increasing prime editing efficiency for diverse goals, and we will be able to take advantage of all such developments in the field. The last part of this

comment asks whether some cells may be biased to recording while others are not. This may be the case, and we won't have a full picture of this until we actually deploy DNA Ticker Tape in the context of many cell types in a controlled fashion, as would happen in mice or *in vitro* gastruloids (efforts in both directions are in progress). We now allude to the potential for variation in prime editing / DNA Ticker Tape efficiency across cell types, the issues that might cause, and how improvement in prime editing and/or development of bioinformatic tools to analyse DNA Ticker Tape data might address these issues in the updated Discussion:

“A separate risk is that prime editing efficiency might vary substantially across cell types. However, any such variation could potentially be ameliorated by technical improvements to system components^{23,24}, by increasing recording capacity and/or by modelling it during tree reconstruction.”

8. Page 3, last pp: This is an honest discussion of the technology, with specific weaknesses discussed. Would it be possible for the authors to determine a way to reduce the site of integration effects with alternative methods? For example, safe harbor loci, insulator elements, etc?

It is a very reasonable suggestion, but we hope to take this in a different direction, related to the last comment. It is clear that there is variation in the efficiency of DNA Ticker Tape-based recording across independent integrations in the lineage experiment. Given that these transgenic insertions are nearly identical, we assume that this is consequent to site-of-integration effects (e.g. chromatin environment, etc.). We are now performing experiments directed at identifying the causes of this variation. Early data suggests a major role for the local chromatin environment, but we feel that this is a major undertaking in its own right and is out of scope for this paper. However, in the long run, we anticipate that this advancing understanding may allow us to predict *a priori* which sites in the genome facilitate high rates of prime editing, and specifically introducing DNA Ticker Tape at such locations, rather than randomly. We now allude to both these possibilities in the updated discussion (revised portion underlined):

“We further envision that a single, synthetic DNA construct that encodes a prime editing enzyme, multiple recording arrays, and a combination of stochastic and signal-specific pegRNAs, could be used to simultaneously record both lineage and biological signals in any multicellular system, *i.e.* a “recorder locus”. A single locus design would be less affected by the integration-site specific effects, as we have observed with integrating multiple DNA Ticker Tape constructs across the genome. Alternatively, if genomic sites with a high prime editing efficiency can be identified, these sites might be leveraged to boost information capture.”

9. Page 5, Fig 1: We find the ordering of this figure is slightly confusing – the panels for b and c would make more sense as panel a. Adding more detail to this figure that includes the sequence, such as a Cas9 and RT fusion construct drawn, highlighting the cut and editing functions, and ultimately showing the shift in write head would be more helpful. Panels d, e, and f are intuitive, they communicate the ideas well.

We have removed the previous panel b, and re-organized the figure to make the previous panel c to the current **Fig. 1a** (shown above with the comment #6). In the new **Fig. 1b** panel, we have drawn one cycle of genome editing by the prime editing complex (Cas9 nickase fused with RTase complexed with pegRNA). We have annotated the nickase and reverse-transcriptase function on our schematics. We have omitted the actual nucleic sequence from **Fig. 1b**, in order to simplify our schematic and because it is shown in **Fig. 1a**.

We welcome any additional feedback on how these figure panels could be further improved.

10. Page 7 (Fig2c-2e): It is a bit unclear when reading the manuscript which figure panels fall into the episome style versus integration style of experiment. It would be a little easier to read if that was explicitly stated.

To clarify our presentation, we have revised the main text Figure 2 and Supplementary Figure 2 (now combined into the **Supplementary Figure 3** in the revised manuscript) with explicit labelling of episomal target vs. integrated target for sequential editing:

Supplementary Figure 3b-e. Characterising diverse DNA Ticker Tape designs for efficiency and directional accuracy. **b.** Efficiency (fraction of edited reads out of all reads) vs. sequential error rate (fraction of edited reads inconsistent with sequential, directional editing out of all edited reads) for 48 3xTAPE constructs on episomal DNA (left) and piggyBAC transposon integrated DNA (right). Both horizontal and vertical error bars are standard deviations from 3 transfection replicates. **c.** Boxplots of the efficiencies and sequential error rates of 3xTAPE constructs derived from 8 basal sequences for each of 6 design procedures. In general, a longer key sequence was associated with a lower error rate, while a longer insertion did not appreciably impact efficiency (e.g. NNGAC with Design-6 vs. NNGA with Design-5). **d.** Boxplots of sequential error rates (left) and efficiencies (right) of 3xTAPE constructs grouped by their basal CRISPR target sequences. Boxplot elements in **c,d** represent: Thick horizontal lines, median; upper and lower box edges, first and third quartiles, respectively; whiskers, 1.5 times the interquartile range; circles, outliers. **e.** Correlation between the sequential error rate (left) and editing efficiency (right) of each 3xTAPE construct either in the context of episomal DNA vs. integrated DNA.

11. Page 7, last pp: While the experiment increases the prospective number of tags that can be created (moving from 4^2 to 4^3 , and ultimately 4^6) to add complexity to the library of barcodes that can be integrated, the efficiency decreased markedly. One concern is that this could limit the potential of this technology to be a high-resolution, high-capture lineage recorder since the added complexity isn't compatible to labeling more cells. While the resulting lineage relationships detected may be more detailed, this seems like it would benefit from increased labeling efficiency with less signals to encode more meaningful information – such as if a cell had specific gene expression. The transcriptional history of a cell is difficult to capture, though this process could offer a means to that end.

We believe this aligns well with our intended use of the technology. The trade-off between diversity and efficiency of insertion barcodes would need to be considered for each application, and documented further by additional experiments in the revised manuscript (**Supplementary Figure 2a**).

Supplementary Figure 2a. Editing efficiencies on the first site of 5xTAPE-1 integrated in the HEK293T cells. A pool of plasmids expressing TAPE-1 targeting epegRNAs were transfected with the pCMV-PEmax-P2A-hMLH1dn plasmid. Five pools with different insertion lengths ranging from 5-bp (NNGGA) to 9-bp (NNNNNNGGA or 6N+GGA) were tested separately. The error bars are standard deviations from 3 transfection replicates.

For recording a handful of specific events such as signal activity (such as Wnt and NF- κ B signalling, as described in ENGRAM preprint; related to Comment #3 above), insertion barcodes with high “capture rate” (high edit score) could be used to maximise the recording efficiency. This would be ideal for recording events within a single cell. On the other hand, diverse tags can be used to record biological events in a population of cells (whether *in vitro* or via “pseudo-bulking” of the signal histories of subsets of related cells in a developmental model), and used to determine the order of many events. We have clarified this by revising the paragraph 3 of the discussion section (added sentence is underlined):

“At least in principle, such strategies are compatible with DNA Ticker Tape, potentially enabling the temporal dynamics of multiple biological signals or other cellular events to be recorded and resolved. In this context, the use of longer and therefore more diverse insertion barcodes could enable extensive multiplexing, although this might come at the expense of recording efficiency.”

12. Page 9, pp 1: Considering the hamming distance between any two barcodes could be as little as 1, how does sequencing error impact the confidence of calling which barcode was integrated into the system?

We were initially concerned with this issue as well, which was the motivation behind constructing transfection Program-1 and Program-2. To explore it specifically, we intentionally maximised (Program-1) or minimised (Program-2) the edit distance between the adjacent epochs (not only Hamming distance but also transversion versus transition error, where transition errors such as G-to-A/A-to-G and C-to-T/T-to-C are observed more frequently). We were able to accurately reconstruct the correct order of both programs, which demonstrated that the edit distance among barcodes did not interfere with the temporal recording capability, at least in this demonstration. We believe this is because we are collecting multiple sequencing reads from each 5xTAPE recording locus in a population of cells, where the erroneous base-calling is much less frequent than the correct

one (and thus does not affect the overall reconstruction of recorded events). In the lineage tracing demonstration, we used cDNA-UMIs from transcriptome profiling (amplicon-UMIs in previous genomic DNA PCR) to minimise possible biases introduced in PCR amplification further.

That said, we did observe a related error in encoding the short text. When encoding the text #2 (“Mr. Watson, come here!”), we observed a small population of ‘GAG’ barcodes, which is likely to be an erroneous calling of abundant ‘AAG’ barcodes. The difference of magnitude is about 100-fold, which is consistent with our experience with the transition error rate during sequencing and exponential PCR amplification. Therefore, barcodes that have a small edit distance start to affect the confidence of calling when their intensity/abundance differ by a couple orders of magnitude.

13. Page 9, pp 2: This approach of the multiple programs to record information and understand the behaviors of prime editing to encode the information into the TAPE is impressive.

Thank you!

14. Page 10, last sentence: “Taken together, these results show that DNA Ticker Tape can record, recover, and decode complex event histories including the order, overlap, and relative strength of signals.” This is an excellent summary of the potential of the technology. This is also why it may be less attuned for high-resolution lineage tracing than it is for other information encoding. High-resolution lineage tracing approaches generally leverage the random encoding of very complex libraries of barcodes, either generated via scarring or randomized synthesis. This statement outlines why it would be so well suited to not creating RANDOM events, but better at recording SPECIFIC events.

We thank the reviewer for these insights. Our view is that the method has strong potential for either random or specific histories, and more importantly for the superimposition of these two things (indeed, that would be the ideal case, as it might yield a “decorated” lineage tree in a way that would not be easily possible with other methods). However, we certainly get the reviewer’s point, and this limitation of DNA Ticker Tape is now highlighted in **Supplementary Table 3**. To express our interpretation of their point another way, one could argue that scarring or random synthesis based approaches for lineage tracing are more efficient with respect to the number of components that they require -- for example, Cas9-based scarring can produce a multitude of distinct edits from a single, constitutively expressed gRNA, while here we require multiple pegRNAs to encode this diversity. On the other hand, we believe that there are major advantages (e.g. avoiding DSBs altogether; the potential to record many specific signals concurrently with lineage) that strongly justify this inefficiency in encoding random signals. We have revised this section of the discussion to now read as follows:

“In a proof-of-concept experiment, we show how DNA Ticker Tape overcomes the major limitations of earlier editing-based lineage recorders like GESTALT, by reducing ambiguity about the order in which editing events occurred, eschewing double-stranded breaks and thereby minimising the risk of inter-target deletion, predefining the locations to which edits accrue, predefining the “symbol set” from which edits are drawn, and stabilising the rate of editing by ensuring one-and-only-one write-head per active tape. These attributes clearly pay off in the proof-of-concept experiment, as we are able to sustain a seemingly steady accumulation of edits to multi-copy DNA Ticker Tape across 25 days of *in vitro* expansion, from a single cell to 1.2M cells.”

We believe that the newly added large-scale monophyletic lineage tracing experiment is a powerful reflection of what is possible with DNA Ticker Tape. Although many pegRNAs must be introduced (rather than just one gRNA, as would be the case with scarring approaches), the absence of DSBs that delete information as well as the

ability to sustain continuous editing over a long period of time (at least 25 days and 20 generations), make it more than worth it. Concrete evidence of this “added value” include the fact that edits appear to accumulate at a Poisson-like rate (new **Figure 4d**) and the fact that we are able to use “off the shelf” phylogenetic methods (UPGMA, NJ) to reconstruct complex trees (new **Figure 4f,h**, **Supplementary Figure 6f**, and **Supplementary File 1**), with only minor modifications to take advantage of the ordering information provided by DNA Ticker Tape.

15. Page 11, Fig3c-3g: The heatmaps depicted here are intuitive and illustrate the relationships between the PE program and editing events very well.

Thank you!

16. Page 12, “Recording and decoding short text messages” and Fig 4: The concept of recording and decoding the sentences is clever – a fun and whimsical read as a proof-of-principle experiment.

Thank you -- in an otherwise tough year, we were indeed attempting to have some fun with it.

17. Page 15: Based on the frequencies of usage of InsertBC barcodes, we calculated their Shannon entropy to be 3.64 and 3.70 bits for Site-1 and Site-2, respectively, out of a theoretical maximum of 4.46 bits if the probabilities of the 22 InsertBC barcodes were equal.” This sentence is somewhat unclear and difficult to read. Perhaps outlining the relationships between the InsertBCs and the pegRNA and the insertions would help?

In the revised manuscript, this experiment and the corresponding text has been replaced by the sc-RNA-seq-based reconstruction of a much larger monophyletic cell lineage tree.

18. Page 16, first 3 lines: Since the damage to the tape was discovered, this is a type of informational dropout that could occur, though the DNA Ticker Tape system is generally robust to the types of dropout generally observed. We are unsure if the “damaged tape” occurs at a low enough rate since we only see data from one monoclonal experiment. We would like to see this performed in many more cells to determine how often it occurs and if this off-target damage makes information dropout more comparable to other single cell lineage tracing methods.

We agree that this is an important parameter to quantify. The monoclonal experiment may be insufficient to assess this parameter, more so because the 5xTAPE-1 is integrated via lentivirus. The process of generating a lentiviral vector involves an RNA template switching event that may affect the size of the tandem-repeat array of TAPE monomers even before the integration. We reasoned that a better setting to monitor a damage on the TAPE length would be our transfection program experiments, where 5xTAPE-1 constructs are integrated to a pool of polyclonal HEK293T cells using a piggyBAC transposase system. We have sampled this particular pool of 5xTAPE arrays 16 times over the course of 45 days while the array was constantly edited throughout that period (**Supplementary Figure 4a**). When we aligned each sampled 5xTAPE-1 array amplicon sequencing data to different repeats of TAPE sequences (3 to 7 repeats), we observe that the majority of amplicons align to the correct 5xTAPE-1 sequence, and that this proportion is largely stable over time, declining from $74.0 \pm 0.2\%$ at epoch 1 to $67.0 \pm 1.7\%$ at epoch 16 (**Supplementary Figure 4b**). This suggests that a five tandem-repeat array of TAPE-1 sequence is quite stable in the HEK293T cells over the time-scale of months.

b. Program-1: Change in repeat-lengths of TAPE array

Supplementary Figure 4b. Repeat-length change of 5xTAPE-1 array sampled over 16 transfection epochs.

19. Page 16, first full pp: When discussing prime editing error rates, are there more comprehensive statistics that could be shared? We again worry about the power of this experiment since it was inferred from one clonally-related cell population.

We have added an analysis on various error modes during the lineage tracing experiment. In particular, we have quantified the insertion, deletion, and substitution error rate on Site-1 of 5xTAPE-1 across different integrants. We observe that error rates are quite low for insertion and deletion errors (0.05% and 0.02% for 5-bp insertion and 7-bp insertion, respectively). For the substitution error, we quantified the error rate on the ‘GGA’ insertion sequence, shared across all InsertBC. Over 99.5% of the 6-bp insertions had the correct ‘GGA’ sequence on the last 3-bp.

Supplementary Figure 6d,e. **d.** Characterization of indel error rates of prime editing on TargetBC-5xTAPE-1 arrays. The Y-axis is on a log₁₀-scale. Correct length insertions with prime editing are >100-fold more likely than an insertion of a different length product. Furthermore, some of the apparent longer insertions are likely to correspond to a contraction of TAPE-1 monomer within 5xTAPE-1 before the integration, such as contraction of TGATGGTGAGCACG TAPE-1 monomer to the observed TGAGCACG 8-bp sequence appearing between two TAPE-1 monomers. **e.** Characterization of substitution error rates during prime editing-mediated insertion of the GGA key sequence on TargetBC-5xTAPE-1 arrays. The X-axis is on a log₁₀-scale. Correct insertions are >100-fold more likely than insertions with substitution errors. The most frequent class of errors are transition errors, and these may be occurring during PCR amplification or sequencing-by-synthesis of cDNA amplicons, rather than during prime editing.

20. Page 16, pp 3, “Identity by state vs. identity by descent”: Given that this is a single comparison of a monoclonal population, would it be easier to find recurrence if multiple, independent clones are compared? Then if cells with different pools of barcodes are used (though with the same transduced barcode pool) and compared, repeated barcode patterns would determine how effective this method is at creating discernable lineage trees or repeating patterns. This “clone mixing” experiment would be a way to determine the efficacy of the method by measuring repeats in clonal-subclonal signatures.

Following the reviewer’s suggestion, we performed a “clone mixing” experiment, where multiple monoclonal cell lines were cultured in a same well, each having a different set of TargetBCs and InsertBCs introduced by a set of pegRNAs, followed by single cell RNA-seq. This new experiment is described in further detail in our response to previous Comment #4 from this reviewer. We observe that TargetBCs originating from the same monoclonal line share the same set of InsertBCs on 5xTAPE-1, and that this information can be used to infer the correct group of TargetBCs per monoclonal cell line (see **Revision figure 1c,d** above).

As noted above, this experiment was conducted in parallel to the new monophyletic DNA Ticker Tape / sc-RNA-seq experiment. However, we decided to leave the polyphyletic experiment out of the main manuscript, because we believe that the new monophyletic experiment ended up providing a much more extensive demonstration of the potential of DNA Ticker Tape for high-resolution, continuous lineage recording. That being said, we would be happy to add this data into the paper if the reviewer feels that it adds substantial value above and beyond the new monophyletic experiment.

21. Page 19: The synthetic minisatellite experiment is challenging due to the nature of the repeats of the DNA. Even after sequential editing, is it possible that shortening of the minisatellite sequence could occur, leading to information dropout? Could an optimal length of TAPE be determined, to maximize the ordered, recordable information but minimize the possibility of TAPE contraction?

We agree that using longer tandem repeats of TAPE constructs could lead to a contraction or expansion of the DNA Ticker Tape arrays, either during DNA replication or recovery of TAPE arrays using PCR amplifications of genomic DNA. While we observe that the 5xTAPE-1 arrays are quite stable in the HEK293T cells and amenable with PCR amplification (see **Supplementary Figure 4b**), we also observe that extending the TAPE arrays to 12 or 20 repeats can lead to a wide variety of TAPE length when recovered. We now perform more extensive analysis and presentation of this data, as summarised in the below text and accompanying figure panels:

“We grouped CCS reads within each replicate based on a degenerate 8-bp barcode (TargetBC), as these presumably derived from the same integration. On average, each TargetBC group had 3.1 ± 3.4 and 3.8 ± 5.7 reads for ~12xTAPE-1 and ~20xTAPE-1, respectively. Within TargetBC groups, shorter arrays appeared more stable, with a greater proportion matching the maximum length within that group (**Supplementary Figure 7e,f**).”

Supplementary Figure 7e. For TargetBC groups with a given maximum number of TAPE-1 monomers (X-axis), we show the mean proportion with the same number of monomers as the maximum (Y-axis), for both 12xTAPE-1 (red) and 20xTAPE-1 (blue) integrants. We conclude from this that shorter arrays are more stable, and that the length-dependent stability is consistent between the two experiments. **f.** Similar to (e), but showing the full distribution of monomer lengths (Y-axis) for each TargetBC group with a given maximum number of TAPE-1 monomers (X-axis), for both ~12xTAPE-1 (red) and ~20xTAPE-1 (blue) integrants. The size of dots are proportional to these proportions.

Regarding the question about the possibility of information dropout, it is notable that we have not observe any instances of contractions that lead to information loss. Although it's possible that we are missing some class of events in our analyses, it's not obvious that this is occurring, which is interesting in itself. We now comment on this and speculate on a possible reason in the following text:

"Of note, the observed variation in array length tended to occur within the unedited portion of the tape (**Supplementary Figure 7g,h**). We have yet to observe any clear examples of "information erasure", possibly because the edits themselves disrupt the tandem repeats, inhibiting processes that might otherwise lead to erasure from spreading proximal to the write-head."

22. Finally, most of the manuscript is extremely well written, but there are some vague phrases and points that can be clarified about the analysis and system (see above). Overall, the figures are comprehensive, but the reader would greatly benefit from a more technical drawing of the process of the "write head" movement from site 1 to site 2. Fully drawing out the prime editing enzyme and demonstrating the process on the sequence to show the key and barcode integration would be helpful for an audience not intimately familiar with prime editing. Other figures would also do well to be altered to gain more clarity, and place sequences into supplemental tables where they can be easily assessed.

We have revised Figure 1a-c to clarify the underlying process, as described in more detail above. We have also sought to make additional revisions to main and supplementary figures to present material in a clearer manner, as well as to provide tables, primers, raw data, intermediate data used for the new lineage reconstruction

experiment, etc. to maximise the clarity and value of this study to the community (in line with suggestions from this reviewer and the other reviewers). This is a priority, and we welcome any and all additional suggestions or feedback to do better in this regard.

Referee #3 (Remarks to the Author):

This work describes a molecular system for logging temporal events in the genomes of eukaryotic cells. The technology has the potential to make a large impact in the understanding of biological systems. One can imagine immediate uses in improved lineage recording, but the greater potential advances will likely come from the ability to encode temporal sequences of molecular events as they happen (pathway activation, gene expression, cell-cell interactions) and decode them by sequencing later on. The fact that the technology is not yet fully developed enough to enable those applications does not detract from the strong impact of this work.

We thank the reviewer for recognizing the potential impact of our method, as well as the acknowledgement that some of these applications will be realised in future work.

However, because it will require further development, it is critical that this work better describe the fundamental parameters of the system and accurately contextualize the experiments.

We agree with the reviewer that our description of the method and its context within the broader field needs to be improved. We have revised the manuscript to address these points in particular. Examples of such revisions include:

- 1) a more extended consideration and discussion of the strengths and weaknesses of our method in relation to other molecular recorder technologies (summarised in **Supplementary Table 3**)
- 2) markedly improving and better characterising the performance of DNA Ticker Tape as a lineage recorder (summarised in the subsection “Ordered recording and decoding of cell lineage histories”)
- 3) clarifying how we are inferring event orders (related to Comments #10, #11, #12 below)
- 4) additional analysis on the site-specific effect of prime editing within the TAPE-1 array (related to Comments #6 and #7 below).

1. The authors argue in the introduction that one drawback of existing Cas9 molecular recorders is toxicity, “Frequent DSBs are toxic.” The authors should quantify the impact of their recording system on the cell physiology. At minimum they should quantify cell viability and doubling time with all components expressed over extended periods of time.

We thank the reviewer for this suggestion. To assess the impact of the recording system on the cell physiology, we measured the cell doubling time of the monoclonal cell line for the lineage tracing experiment in the revised manuscript. In the presence of the recording systems (Dox-induced PE2 and multiple lentiviral integrated pegRNA-expression cassettes and TargetBC-5xTAPE-1 constructs), the cell doubling time was noticeably longer (20.7 ± 0.3 hours) compared to HEK293T cells lacking them (18.7 ± 0.4 hours). While we could attribute some of these differences to the presence of doxycycline at a high (10 $\mu\text{g}/\text{mL}$) concentration, as the addition of doxycycline increased the doubling time of HEK293T cells to 19.7 ± 0.9 hours, it is possible that the continual

expression of PE2 may be negatively affecting cell physiology. These results are now presented in **Supplementary Figure 6a**:

Supplementary Figure 6a. Cell doubling time measurements for the HEK293T cell line and monoclonal cell line used in the lineage tracing experiment, with or without doxycycline present in the media.

2. The authors describe the system as being used to record signals of interest: “Signals of interest are coupled to the expression of specific prime editing guide RNAs.” However, the signal of interest used in this work is transfection of barcoded pegRNA plasmids, which is of limited interest beyond demonstrating the potential of the technology in future applications. The work would be more accurately described without invoking the vague ‘signals of interest’ that makes it seem like biological events might have been recorded. Unless the authors would like to drive the system with an actual signal (rather than the physical delivery of the material to be encoded), I would recommend revising the language to more literally describe the results. For instance, “Each signal of interest is coupled to the expression or activity of a prime editing guide RNA (pegRNA),” could become something like, “To demonstrate the potential application, we encoded successive transfection events using specific primer editing guide RNAs (pegRNAs).” For another, “At the beginning of each epoch of each transfection program, one or more synthetic signals was introduced to a population...” should just be “...one or more pegRNA plasmids was introduced...” This applies throughout the manuscript to distinguish between what could be done with the system from what has been done with the system.

We agree with the reviewer’s concern with our use of language in the submitted manuscript. We have revised the manuscript to ensure that we are accurate in describing what has been achieved vs. what might be achieved in the future, as well as being more careful around our use of the term ‘signal’. For instance (revised texts are underlined):

Abstract: “Recording is mediated by the activity of specific prime editing guide RNAs at the DNA Ticker Tape².”

Page 2: “Here we describe a DNA-based memory device that is: (1) highly multiplexable, *i.e.* compatible with the concurrent recording of at least thousands of distinct event types.”

Page 13: “At the beginning of each epoch of each transfection program, one or more pegRNA plasmids were introduced to a population of HEK293T cells with integrated 5xTAPE-1 (5xTAPE-1(+) HEK293T) via transient transfection of plasmids expressing the corresponding pegRNA(s) and PE2.”

Page 13: “Programs 1 and 2 each consisted of a distinct, non-repeating sequence of transfection of the 16 pegRNAs, *i.e.* one per epoch.”

Figure 2 legend: “Figure 2. Transfection programs for 16 sequential epochs. a. Schematic of five transfection programs over 8 or 16 epochs. For Programs 1 and 2, pegRNAs with single barcodes were introduced in each epoch for 16 epochs. The specific orders aimed to maximise (Program-1) or minimise (Program-2) the edit distances between temporally adjacent transfections.”

3. As a related point, the authors note while discussing the use of this system that “engineered guide RNAs whose activity is dependent on the binding of specific small molecules or ligands” could be recorded, but that “the further development of such signal conversion systems is a non-trivial challenge.” It is hard to know how much of a challenge this might be without additional experimental insight into the sensitivity and dynamic range. How abundant are these pegRNAs as compared to endogenous transcripts, as assessed by qPCR or some other quantitative metric? What is the dynamic range, i.e. the relative editing efficiencies at different concentrations of pegRNA? Those numbers are critical to enable the necessary further development of the system into an event recorder.

We agree that these are key questions. In parallel to developing DNA Ticker Tape as outlined in this manuscript, we have developed a separate technique referred to as ENGRAM (ENhancer-driven Genomic Recording of transcriptional Activity in Multiplex) (W. Chen et al. 2021). In this manuscript, the dynamic range of pegRNA-based recording of the activity of 300 distinct enhancer elements is shown, and also found to have high correlation with expression-based results of a conventional MPRA assay (for both assays, with activities spanning a roughly 128-fold range).

Adapted from the Fig. 2 of Chen et al. 2021: A set of 300 enhancers known to have a range of activities in a reporter assay were tested using ENGRAM, wherein each enhancer is coupled to the expression of a pegRNA that mediates an enhancer-specific insertion to a shared genomic target (left). The abundances of pegRNAs driven by the 300 enhancers correlate well with the frequency of the corresponding enhancer-specific insertions to genomic DNA.

4. Although the overall concept is communicated well and easy to understand, the molecular implementation took a lot of effort to figure out. It is unclear why 1b and 1c both need to exist as they show largely the same thing. In figure 1c, the schematic could be expanded and better annotated. It would be useful to annotate the write-head onto the figure. It would also be useful to depict the pegRNA with sequence. The text and legend refer to base positions of the PAM, but there are multiple PAMs. I could not figure out what purpose the mismatch bases are serving, which should be better explained.

To clarify our molecular implementation of DNA Ticker Tape, we substantially revised previous **Figure 1a-c**. In particular, we have removed the previous panel 1b, expanded panel 1c with better write-head annotation, and

reorganised the figure such that panel 1c is now **Figure 1a**. In the revised panel, we have annotated a single PAM sequence that is relevant to the current “write-head”. We have also removed the annotation of mismatch bases, which we now explain further in a later section. We have introduced a schematic of prime editing as a new **Figure 1b**, drawing both Prime-Editor (Cas9 nickase tethered to RTase) and pegRNA for editing.

We welcome any additional feedback on how this figure could be further improved.

Figure 1a-c. Sequential genome editing with DNA Ticker Tape. **a.** Schematic of editing event at the “write head” of the DNA Ticker Tape. The DNA Ticker Tape consists of a tandem array of CRISPR-Cas9 target sites (grey boxes), all but the first of which are truncated at their 5’ ends, and therefore inactive. The 5-bp insertion includes a 2-bp pegRNA-specific barcode as well as a 3-bp key that activates the next monomer. Because genome editing is sequential in this scheme, the temporal order of recorded events can simply be read out by their physical order along the array. **b.** The schematic of prime editing during one round of recording. Prime editing recognizes a CRISPR-Cas9 target and modifies it with the edit specified on the pegRNA molecule. In DNA Ticker Tape, an insertional editing event generates a new prime editing target for the sequential editing. **c.** Schematic of recording the event order on DNA Ticker Tape. Individual pegRNAs are either event-driven²² or constitutively expressed, together with the PE2 enzyme.

5. The analysis of the experiment in Figure 1 is lacking in some basic descriptive results. A plot of the relative editing efficiency for each barcoded pegRNA should be included in the main figure, depicting the variability across replicates. The critical value I would suggest plotting is editing score. Given the large range of editing efficiencies with the different barcodes, it would be useful to have the barcode itself on that plot as well, so that one might, at the minimum, select useful barcodes in future iterations. It would be ideal to investigate if there are

any rules to the high editing or low editing barcodes. Is there any chance that the high editing barcodes actually include prime editing errors, inflating their frequency?

Following this suggestion, we have added a new panel (**Figure 1g**) describing the edit score for 16 individual barcodes:

Figure 1g. Edit score of 16 barcodes used in the experiment with 5xTAPE-1. Edit scores for each insertion are calculated as \log_2 of the ratio between insertion frequencies and the abundances of pegRNAs in the plasmid pool, averaged over 3 transfection replicates.

We also now included a brief description of Figure 1g in the main text:

“The maximal edit score difference between the best barcode (CCGGA with an edit score of 0.98 ± 0.02) and the worst barcode (TGGGA with an edit score of -2.38 ± 0.26) is 3.36, which is equivalent to nearly 10-fold difference in the editing efficiency.”

In addition to the edit score for NNGGA using pegRNA, we now experimentally measure and calculate the edit score for NNGGA, NNNGGA, and NNNNNNGGA (6N+GGA) using epegRNAs with enhanced editing efficiency. These results are included in the **Supplementary Figure 2** of the revised manuscript. We note that the use of epegRNA (Nelson et al. 2021) decreased the range of efficiencies for different barcodes, such that 14 of 16 NNGGA barcodes now exhibit efficiencies within a 2-fold range (**Supplementary Figure 2c**).

Supplementary Figure 2c. Edit scores for 16 NNGGA insertions with epegRNA. Edit scores for each insertion are calculated as \log_2 of the ratio between insertion frequencies and the abundances of pegRNAs in the plasmid pool.

We believe it is unlikely that the high editing barcodes actually include prime editing errors at appreciable levels, in part because the error rate on the GGA ‘key’ sequence (shared across all insertions) are less than 1% (discussed in detail below with respect to Comment #8), and where they do occur, their signature is consistent with PCR/sequencing errors (i.e. transition errors are more prevalent than transversion errors). Of course, it is possible that error rates of prime editing are different at the 5’ (barcode) and 3’ (key) ends of the insertion, but data from other experiments (e.g. the sequential transfection experiment) does not support this.

6. The authors point out a pseudo-processivity where the editing frequency is increased from the first to the second position. It appears that it might also decline beyond the second position, apparent in the supplemental data and consistent with results further into the paper. Data for editing efficiency by position for Figure 1 and 3 should be plotted and discussed in terms of the accuracy and temporal limits of the recording.

Following the suggestion from this (#6) and the next (#7) comment, we have calculated and plotted the “site-specific efficiency” (percent of population that has been edited at the monomer given that the preceding monomer has been edited) for integrated 5xTAPE-1 from Figure 1 (left-most) and Figure 3 (Program-1; right-most), as well as new data from episomal 5xTAPE-1 (left-center) and the episomal 3xTAPE-1 (right-center). Despite the difference in experiments (integrated vs. episomal, 5xTAPE vs. 3xTAPE construct, collected after one round of transfection vs. sixteen rounds of transfection), we observed that the editing efficiencies of sites 2-5 tend to be greater than that of site 1.

Supplementary Figure 1a. Conditional, site-specific editing efficiencies across 5 sites within the 5xTAPE-1, calculated as the number of reads that contain an edit in the indicated site over the total number of reads that contain an edit in the immediately preceding site, which activates the indicated site as a target for editing. The number of all 5xTAPE-1 reads were used for calculating the site-specific editing efficiency for the Site-1, which is activated by its own key sequence.

While we are not entirely sure why the editing efficiency declines beyond the second site (consistent across different experiments), one possible explanation is that the later editing occurs after the previous editing event, thus being ‘active’ for less time than predecessors. However, we also note that the temporal resolution and accuracy of recording are not likely to be affected by the site-specific editing rate, as the information across all five sites are combined for the inference, *e.g.* as demonstrated in the experiments represented in Figures 2 and 3.

7. Since the interpretation of the pseudo-processivity is inaccessibility of some editing sites when integrated randomly into the genome, the authors should look for this effect in the episomal plasmid editing (in figure 2). If their interpretation is correct, there should be no difference in editing efficiency by position in the episomal plasmid. If there is, they need to revise their interpretation.

We thank the reviewer for the excellent suggestion. We performed the suggested analysis on the existing 3xTAPE data (**Supplementary Figure 1a**, right-center), while also collecting new data by performing an experiment to observe editing on 5xTAPE without integration (**Supplementary Figure 1a**, left-center). Based on these data, similar ‘pseudo-processivity’ is observed for both integrated and episomal TAPE-array in sequential editing.

While these new data shows that we still do not fully understand the mechanism behind the observed ‘pseudo-processivity’, we would argue that these observations do not exclude our hypothesis that this is partly due to the fact that some targets are better edited than the others. For instance, it has been demonstrated that the chromatin context seems to affect the recognition of targets by CRISPR-Cas9 to a different degree (Schep et al. 2021). Given the fact that transfected plasmids are epigenetically modified to be silenced, it is conceivable that some episomal targets are less amenable to prime editing than others. We have revised our discussion of the result to include these observations and clearly note that we do not fully understand the underlying mechanism behind the observed pattern:

“An interesting phenomenon is that while the observed editing rate of the first TAPE-1 monomer was ~6%, the editing rates of the second or third TAPE-1 monomers, conditional on the preceding monomers already being edited, were ~20% (**Supplementary Figure 1a**). A simple explanation for this ~15% greater “elongation” than “initiation” of editing is that some integrated tapes are more amenable to prime editing than others, resulting in an excess of fully unedited tapes. However, we also observe a similar pattern with episomal tapes, as well as upon multiple sequential transfections of pegRNA/PE2-expressing plasmids to edit integrated tapes (7-15% increase in

the conditional editing efficiency of the second site). Factors that might contribute to the observed “pseudo-processivity” include heterogeneous susceptibility of cells to transfection, chromatin context^{20,21}, and cell cycle phase, but the primary explanation remains unclear. We also observe modest reductions in the conditional editing efficacy after the second site (1-10% decreases), which might be explained by each site being “active” for less time than its predecessor.”

8. In the lineage section it is stated that “Prime editing errors are rare, but do occur.” What is rare? We need a quantification of these errors and error types, with particular focus on errors that could be mistaken as recording data.

In response to this comment, we have performed and added an analysis on various error modes during the lineage tracing experiment. In particular, we have quantified the insertion, deletion, and substitution error rate on Site-1 of 5xTAPE-1 across different integrants. We observe that error rates are quite low for insertion and deletion errors (0.05% and 0.02% for 5-bp insertion and 7-bp insertion, respectively). For substitution errors, we quantified the error rate on the ‘GGA’ insertion sequence (*i.e.* the key), shared across all InsertBC. Over 99.5% of the 6-bp insertions had the correct ‘GGA’ sequence on the last 3-bp.

Supplementary Figure 6d,e. **d.** Characterization of indel error rates of prime editing on TargetBC-5xTAPE-1 arrays. The Y-axis is on a log₁₀-scale. Correct length insertions with prime editing are >100-fold more likely than an insertion of a different length product. Furthermore, some of the apparent longer insertions are likely to correspond to a contraction of TAPE-1 monomer within 5xTAPE-1 before the integration, such as contraction of TGATGGTGAGCACG TAPE-1 monomer to the observed TGAGCACG 8-bp sequence appearing between two TAPE-1 monomers. **e.** Characterization of substitution error rates during prime editing-mediated insertion of the GGA key sequence on TargetBC-5xTAPE-1 arrays. The X-axis is on a log₁₀-scale. Correct insertions are >100-fold more likely than insertions with substitution errors. The most frequent class of errors are transition errors, and these may be occurring during PCR amplification or sequencing-by-synthesis of cDNA amplicons, rather than during prime editing.

9. In Figure 2, most TAPes have error rates that look to be >50%. What sort of errors are these, prime editing errors or directional errors? The authors should include this categorization for all the designs tested.

Great point. We have modified the figure legend and axis labels to make it clear that we are reporting directional (sequential) errors in this figure, and not prime editing errors.

Supplementary Figure 3b-e. Characterising diverse DNA Ticker Tape designs for efficiency and directional accuracy. **b.** Efficiency (fraction of edited reads out of all reads) vs. sequential error rate (fraction of edited reads inconsistent with sequential, directional editing out of all edited reads) for 48 3xTAPE constructs on episomal DNA (left) and piggyBAC transposon integrated DNA (right). Both horizontal and vertical error bars are standard deviations from 3 transfection replicates. **c.** Boxplots of the efficiencies and sequential error rates of 3xTAPE constructs derived from 8 basal sequences for each of 6 design procedures. In general, a longer key sequence was associated with a lower error rate, while a longer insertion did not appreciably impact efficiency (e.g. NNGGAC with Design-6 vs. NNGA with Design-5). **d.** Boxplots of sequential error rates (left) and efficiencies (right) of 3xTAPE constructs grouped by their basal CRISPR target sequences. Boxplot elements in **c,d** represent: Thick horizontal lines, median; upper and lower box edges, first and third quartiles, respectively; whiskers, 1.5 times the interquartile range; circles, outliers. **e.** Correlation between the sequential error rate (left) and editing efficiency (right) of each 3xTAPE construct either in the context of episomal DNA vs. integrated DNA.

10. For the experiments presented in Figure 3c,d, TAPes with an early barcode in position 1 (e.g. CA) should have every later barcode at nearly the same frequency in subsequent positions since they were all present for the same amount of time. Yet, the frequency falls off the further in time a barcode gets from the first barcode.

On the contrary, we believe that we would expect the observed pattern. TAPE monomers are edited without replacement, similar to how chemical substrates are consumed during reaction. This trend can be observed in Figure 3b, where the earliest barcode ('CA' from Epoch 1) mostly appears on the Position-1 of 5xTAPE compared to later barcodes (such as 'CT' from Epoch 6). Similarly, TAPE dimers that have 'CA' on its first position (CA-unedited) are being edited over time without replacement. For example, assuming 10% editing efficiency of barcodes in the first Epochs (barcodes with CA, GC, and TA), we would expect 90% [CA-unedited] population and 10% [CA-GC] population after the second Epoch, and 81% [CA-unedited], 10% [CA-GC], and 9% [CA-TA] after the third epoch (where 10% of remaining [CA-unedited] population has been edited with TA). Therefore, one might expect to have less edits from later barcodes next to the earliest CA barcode, as observed.

That decline in frequency would indicate that TAPEs are only recording a few signals before they stop recording, and therefore the recording is constrained in time. It is possible that this graph is not plotting exactly what I think it is. Either the result should be discussed or the analysis should be modified/clarified.

We would expect that the editing rate would be more or less constant because consumption of one editing substrate (*i.e.*, an activated TAPE monomer within 5xTAPE) would create another editing substrate until the whole DNA Ticker Tape array has been edited. That said, the editing rate would eventually decline with time in a population of cells, as individual TAPE-arrays are run out to their last monomer and thus consumed over time.

Relevant to this point, we note that in the context of the new single cell lineage tracing experiment (new **Figure 4**), we note that the accumulation of edits to the 59 potentially editable sites exhibited a Poisson distribution, with, on average, 39.4 of 59 editable sites edited per cell (**Figure 4d**), over 25 days of expansion of a monoclonal cell line from one to several million cells. During this time, the majority of TAPE-1 arrays were fully edited (*i.e.* all five sites) (**Figure 4c**), such that the number of active “write-heads” across the full set of tapes remained reasonably stable, declining from 13 in the founding cell to an average of 8.6 per cell after 25 days.

Generally, the graphs in Figure 3c-g are very difficult to understand. The authors should consider a different type of plot or some sort of schematic to explain. The schematic in 3a is excellent.

To clarify this, we have added **Supplementary Figure 4**, plotting the inferred event orders and intensities. We hope the addition of these panels clarifies the data depicted in Figure 3c-g (**Figure 2c-g** in the revised manuscript).

11. To the point above, it is unclear if only the first and second position of the 5XTAPE are being analyzed for the order, or if all positions are being used. A useful analysis would be to quantify how much order information is contained in the TAPE by position in the TAPE, to help inform the design of future experiments.

We have analysed the order by considering four bigrams on 5xTAPE-1 editing sites (*i.e.*, Site-1 & Site-2, Site-2 & Site-3, etc.). We have revised the manuscript to clarify this point (revised portion underlined):

“In **Figure 2c,d**, we show heatmaps of bigram frequencies, measured from all 4 pairs of adjacent editing sites on 5xTAPE-1, arranged by the true order in which the signals were introduced for Programs 1 and 2.”

Our assumption is that the quantity of the order information is closely related to the editing efficiency of each TAPE position; editing of two adjacent sites with different insertion barcodes contribute directly to the inference of the overall order, and more sampling leads to a more precise reconstruction without presuming accuracy. In the revised manuscript, we have included an editing efficiency of five sites over time:

Supplementary Figure 4a. Sequential editing efficiency and sum of sequential errors from five sites in 5xTAPE across 16 transfection epochs of Program-1. The error bars are standard deviations from 3 transfection replicates.

12. The temporal recovery analysis used in Fig 3 is nearly identical in principle to the temporal analysis performed in Shipman et al., Science 2016 and Shipman et al., Nature 2017 to recover timing information from data encoded in CRISPR arrays. The data generated in those studies of an earlier temporal DNA molecular recorder is identical in structure to the data generated here. In those papers, CRISPR arrays were analyzed by taking pairwise comparisons of the order of individual spacers within the array (e.g. Shipman et al., Science 2016 Fig 3), just as the bigram frequencies are used in this work. This text should point out the similarities in data and analysis to help contextualize the work and establish standard analyses in this field.

We thank the reviewer for pointing this out. We have revised the text (shown below with revised portions underlined) to indicate the similarities and differences of our data and analysis to those from Shipman et al. (Shipman et al. 2016, 2017). Similar to these papers, we have utilised the pairwise ordering information to reconstruct the entire event order, which is necessary and sufficient ordering information used in a typical sorting algorithm. Shipman et al. incorporated such information by enumerating all pairwise ‘ordering rules’ among events ($(n^2+n)/2$ for n events) and permutation ($n!$ for n events). This was sufficient for ordering a set of 5 events, resulting in a 15-by-120 binary matrix to consider for these papers. However, the size of computation increases sharply with the number of events, where we would need to consider a 136-by-20,922,789,888,000 (~21 trillion) binary matrix for inferring the order of 16 events in our demonstration. Instead, we have taken an approach similar to a sorting algorithm such as a “bubble sort”, which is easy to implement:

“However, the inference can be improved by leveraging the sequential aspect of DNA Ticker Tape, for instance by analysing bigram frequencies or pair-wise appearance of events as used in inferring orders from CRISPR-Cas spacer acquisition process (Cas1-Cas2 system used in bacteria)^{11,30}. For example, if signal B preceded signal A, then we expect many more B-A bigrams than A-B bigrams at adjacent, edited sites within 5xTAPE-1. In **Figure 2c,d**, we show heatmaps of bigram frequencies measured from all 4 pairs of adjacent editing sites on 5xTAPE-1, arranged by the true order in which the signals were introduced for Programs 1 and 2. Indeed, the bigram frequencies appear to capture event order information, evidenced by the gross excess of observations immediately above vs. immediately below the diagonal (e.g. in Program-1, CA-GC >> GC-CA). One way to leverage this information is by enumerating the “ordering rules” among all events for possible permutations and then checking whether the observed data matches the best^{11,30}. However, the number of ordering rules for n events increases to the order of n^2 (for ordering 16 events, there are 136 ordering rules, or $(n^2+n)/2$ in general), while the number of possible permutations increases to n factorial. As a more computationally efficient approach, we implemented the following algorithm: (1) initialise with the event order inferred from the Site-1 unigram frequencies; (2) iterate through adjacent epochs from beginning to end, and swap signals A and B if the bigram frequency of B-A is greater

than A-B; (3) repeat step 2 until no additional swaps are necessary. For both Programs 1 and 2, this algorithm resulted in the correct ordering of the 16 signals, out of 16 factorial or 21 trillion possibilities (**Supplementary Figure 4c**). This inference was robust to the sequencing depth, as the correct order could be reconstructed from as few as 2500 reads of the 5xTAPE-1 amplicon (**Supplementary Figure 4d**)."

13. This sentence doesn't make sense: "Correspondingly, although there are 4³ or 64 potential InsertBC barcodes, only 22 were observed at appreciable frequencies, suggesting that one recurred by chance..." Why one?

In our previous manuscript, we observed 23 TargetBC and 22 InsertBC, which used to infer that one InsertBC is likely to have recurred by chance, given the possible complexity of InsertBC (64 potential combinations of 3-bp) than that of TargetBC (4⁸ = 65,536 potential combinations of 8-bp). We note that the section containing this sentence (**Ordered recording and decoding of cell lineage histories**) has been replaced in the revised manuscript with a new experiment, such that this problematic sentence is no longer in the paper.

14. The first citation to DNA based recordings is to the "theoretically promising but methodologically underdeveloped...DNA memory device," with a citation to a twenty year old patent. These devices have been physically demonstrated in the meantime, so this historical context is perhaps not the most relevant. In particular, the cited Roquet et al. 2016, Sheth et al. 2017, and Shipman et al. 2016 all demonstrate the physical recording of events over time in DNA. Sheth et al. and Shipman et al. use systems that generate incredibly similar data to the one presented here, with the major difference being eukaryotic vs prokaryotic. Building from that literature directly would better contextualize this work.

We agree and thank the reviewer for their suggestion. We have modified the discussion to explicitly point out the strong parallels between our present manuscript and earlier works demonstrating DNA-based molecular recorders using different CRISPR-based genome editing methods, and also attempt to summarise the current repertoire of systems in **Supplementary Table 3**.

"DNA Ticker Tape improves on existing CRISPR-based molecular recorders in important ways (**Supplementary Table 3**). The sequential editing of DNA Ticker Tape resembles Cas1-Cas2-based recording^{10,11,16}, which at present are limited to bacterial systems. In DOMINO¹⁹ and CAMERA¹⁴, base editors are used to record biological signals to "pre-programmed logic circuits" composed of multiple targets for base editing. Although these methods are conceptual predecessors to DNA Ticker Tape, there are critical differences. In particular, with all three methods, a recording event creates a new target for further editing (*i.e.* the write-head). However, with DOMINO and CAMERA, each logic circuit is designed to record a specific order. In contrast, a single DNA Ticker Tape construct can potentially record any order. For example, to distinguish pairwise orderings within a set of *n* events, DOMINO or CAMERA would require *n*-choose-2 recording logic circuits or a system that contains the order of *n*² number of unique gRNA and their targets. In contrast and as demonstrated here (**Figure 2**), DNA Ticker Tape requires only a single target array such as 5xTAPE-1, along with *n* unique pegRNAs that encode different insertions but share the same target."

Molecular recording methods	TRACE; Record-seq; Shipman et al.	GESTALT	HOMING; stgRNA	CHYRON	CAMERA; DOMINO	DNA Ticker Tape
DNA editing modality	Cas1-Cas2 integrase	Cas9 nuclease	Cas9 nuclease	Cas9 nuclease and Terminal deoxynucleotidyl Transferase (TdT)	Base editors (e.g., Cas9 nickase with cytidine deaminase)	Prime editors (e.g., Cas9 nickase with reverse-transcriptase)

Information content per recording	Up to 28-bp preset spacer sequence (<56 bits)	Stochastic NHEJ/MMEJ DNA repair outcome	Stochastic NHEJ/MMEJ DNA repair outcome	8.4-bp insertion on average (<17 bits)	Up to 2-bp within the target editing window (2 bits)	Up to 6-bp preset insertion sequence (<12 bits)
Possible symbols per target	>1 trillion	1	1	1	1	10-1000
Recording operation	Expansion of array during spacer acquisition	Unordered indels at adjacent targets (del > ins)	Accumulated indels at self-targeting site (del > ins)	Accumulated indels at self-targeting site (ins > del)	5+ sequential single base-pair edits	14+ sequential insertional edits w/ pegRNAs
Limiting factor in multiplexing	Number of spacer precursors	Array size	Number of stgRNAs	Number of stgRNAs	Number of sgRNAs and targets	Number of pegRNAs
Temporal inference	Position within the target array	Shared edits among target arrays	Position within the stgRNA	Position within the stgRNA	Programmed to each target-sgRNA circuit	Position within the target array
Necessary cellular components	Cas1-Cas2, Spacer DNA oligos	Cas9, sgRNAs, target arrays	Cas9, stgRNAs	Cas9, TdT, stgRNAs	Base editor, sgRNAs	Prime editor, pegRNAs, DNA Ticker Tape target
Demonstrated host range	Escherichia coli	Eukaryotic	Eukaryotic	Eukaryotic	Prokaryotic and eukaryotic	Eukaryotic
Key limitations	Portability to eukaryotic cells	Inability to capture the event order; Large deletion on target arrays	Inability to form target arrays; Reduced activity for longer stgRNA	Inability to form target arrays; Reduced activity over time	Limited diversity in editing per target	High multiplicity of integration necessary for highly diverse edits
References	(Shipman et al. 2016; Shipman et al. 2017; Schmidt et al. 2018; Sheth et al. 2017)	(McKenna et al. 2016; Chan et al. 2019; Bowling et al. 2020)	(Peri et al. 2016; Kalhor et al. 2017; Kalhor et al. 2018; Park et al. 2021)	(Loveless et al. 2021)	(Tang and Liu 2018; Farzadfar et al. 2019)	This paper

Supplementary Table 3. Comparison of example CRISPR-based molecular recording methods to DNA Ticker Tape. The overall table structure was adapted from Table 1 of Sheth and Wang (2017).

Additionally, while this system enables recordings in eukaryotic cells, it presumably would not work in prokaryotic cells so distinguishing the parallel context for each approach would be useful.

While we acknowledge that the existing technologies such as Cas1-Cas2-based editing methods would be sufficient for recording in a prokaryotic system, we fail to see why our current system would not work in prokaryotic cells in principle. There has been a report that prime editing works in prokaryotic cells with appreciable efficiencies (Tong et al. 2021), and DNA Ticker Tape constructs are likely to be stable in a bacterial context as well.

15. Regarding the name DNA Ticker Tape, the use of the term Ticker Tape is so common in the field of molecular recording and has already been associated with other DNA based recorders that it seems likely to cause confusion. Would previous similar technologies that use DNA for recording, and are self-described as molecular ticker tapes, or biological ticker tapes not fall into this general umbrella? It would be better to include something to distinguish the pegRNA/prime approach within the name for clarity.

In our previous literature search, we found the use of name “molecular ticker tape” (Zamft et al. 2012 PLOS One), “polymerase ticker tape” (Sheth and Wang 2017 Nature Reviews Genetics referring to Zamft et al.), and more recently “protein ticker tape” (Lin et al. 2021 bioRxiv), but we have not observed the use of “DNA Ticker Tape”. Although we see the reviewer’s point, we feel that “DNA Ticker Tape” aptly describes our method while differentiating from the existing ones, and have had difficulty coming up with a better alternative. Our preference would be to keep this name, although we would be happy to further consider changing it if the reviewer feels that it is particularly confusing.

16. The pegRNA plasmid is not described, including no description of the promoter used.

We have revised our Materials and Methods section of the manuscript to include the promoter used for pegRNA plasmids:

“The pegRNA plasmids used in transient transfection experiments were cloned using plasmid backbone pU6-pegRNA-GG-acceptor (Addgene #132777), following the protocol outlined in Anzalone et al.². The resulting pegRNA expression cassette would have a U6 promoter and poly-T terminator.”

The pegRNA plasmid (pU6-CApegTAPE1), along with other representative plasmid essential to this study (piggyBAC-5xTAPE-1-BlastR) has been deposited to Addgene (#175808, #175809) and made available to the public. We have deposited the lineage tracing plasmid compatible with single-cell RNA-seq to Addgene (#183790) in parallel to revision as well.

REFERENCES

- Chen, Peter J., Jeffrey A. Hussmann, Jun Yan, Friederike Knipping, Purnima Ravisankar, Pin-Fang Chen, Cidi Chen, et al. 2021. “Enhanced Prime Editing Systems by Manipulating Cellular Determinants of Editing Outcomes.” *Cell* 184 (22): 5635–52.e29.
- Chen, Wei, Junhong Choi, Jenny F. Nathans, Vikram Agarwal, Beth Martin, Eva Nichols, Anh Leith, Choli Lee, and Jay Shendure. 2021. “Multiplex Genomic Recording of Enhancer and Signal Transduction Activity in Mammalian Cells.” *bioRxiv*. <https://doi.org/10.1101/2021.11.05.467434>.
- Cho, Seung Woo, Sojung Kim, Yongsub Kim, Jiyeon Kweon, Heon Seok Kim, Sangsu Bae, and Jin-Soo Kim. 2014. “Analysis of off-Target Effects of CRISPR/Cas-Derived RNA-Guided Endonucleases and Nickases.” *Genome Research* 24 (1): 132–41.
- Doench, John G., Nicolo Fusi, Meagan Sullender, Mudra Hegde, Emma W. Vaimberg, Katherine F. Donovan, Ian Smith, et al. 2016. “Optimized sgRNA Design to Maximize Activity and Minimize off-Target Effects of CRISPR-Cas9.” *Nature Biotechnology* 34 (2): 184–91.
- Farzadfard, Fahim, Nava Gharaei, Yasutomi Higashikuni, Giyoung Jung, Jicong Cao, and Timothy K. Lu. 2019. “Single-Nucleotide-Resolution Computing and Memory in Living Cells.” *Molecular Cell* 75 (4): 769–80.e4.
- Hsu, Patrick D., David A. Scott, Joshua A. Weinstein, F. Ann Ran, Silvana Konermann, Vineeta Agarwala, Yinqing Li, et al. 2013. “DNA Targeting Specificity of RNA-Guided Cas9 Nucleases.” *Nature Biotechnology* 31 (9): 827–32.
- Kalhor, Reza, Kian Kalhor, Leo Mejia, Kathleen Leeper, Amanda Graveline, Prashant Mali, and George M. Church. 2018. “Developmental Barcoding of Whole Mouse via Homing CRISPR.” *Science* 361 (6405). <https://doi.org/10.1126/science.aat9804>.
- Kim, Do Yon, Su Bin Moon, Jeong-Heon Ko, Yong-Sam Kim, and Daesik Kim. 2020. “Unbiased Investigation of Specificities of Prime Editing Systems in Human Cells.” *Nucleic Acids Research* 48 (18): 10576–89.
- Koblan, Luke W., Jordan L. Doman, Christopher Wilson, Jonathan M. Levy, Tristan Tay, Gregory A. Newby, Juan Pablo Maianti, Aditya Raguram, and David R. Liu. 2018. “Improving Cytidine and Adenine Base Editors by Expression Optimization and Ancestral Reconstruction.” *Nature Biotechnology* 36 (9): 843–46.
- Liu, Yao, Xiangyang Li, Siting He, Shuhong Huang, Chao Li, Yulin Chen, Zhen Liu, Xingxu Huang, and Xiaolong Wang. 2020. “Efficient Generation of Mouse Models with the Prime Editing System.” *Cell Discovery* 6 (1): 27.
- Nelson, James W., Peyton B. Randolph, Simon P. Shen, Kelcee A. Everette, Peter J. Chen, Andrew V. Anzalone, Meirui An, et al. 2021. “Engineered pegRNAs Improve Prime Editing Efficiency.” *Nature Biotechnology*, October. <https://doi.org/10.1038/s41587-021-01039-7>.
- Richter, Michelle F., Kevin T. Zhao, Elliot Eton, Audrone Lapinaite, Gregory A. Newby, Benjamin W. Thuronyi, Christopher Wilson, et al. 2020. “Phage-Assisted Evolution of an Adenine Base Editor with Improved Cas Domain Compatibility and Activity.” *Nature Biotechnology* 38 (7): 883–91.
- Schene, Imre F., Indi P. Joore, Rurika Oka, Michal Mokry, Anke H. M. van Vugt, Ruben van Boxtel, Hubert P.

- J. van der Doef, et al. 2020. "Prime Editing for Functional Repair in Patient-Derived Disease Models." *Nature Communications* 11 (1): 5352.
- Schep, Ruben, Eva K. Brinkman, Christ Leemans, Xabier Vergara, Robin H. van der Weide, Ben Morris, Tom van Schaik, et al. 2021. "Impact of Chromatin Context on Cas9-Induced DNA Double-Strand Break Repair Pathway Balance." *Molecular Cell* 81 (10): 2216–30.e10.
- Shipman, Seth L., Jeff Nivala, Jeffrey D. Macklis, and George M. Church. 2016. "Molecular Recordings by Directed CRISPR Spacer Acquisition." *Science* 353 (6298): aaf1175.
- . 2017. "CRISPR–Cas Encoding of a Digital Movie into the Genomes of a Population of Living Bacteria." *Nature* 547 (7663): 345–49.
- Tang, Weixin, and David R. Liu. 2018. "Rewritable Multi-Event Analog Recording in Bacterial and Mammalian Cells." *Science*. <https://doi.org/10.1126/science.aap8992>.
- Tong, Yaojun, Tue S. Jørgensen, Christopher M. Whitford, Tilmann Weber, and Sang Yup Lee. 2021. "A Versatile Genetic Engineering Toolkit for E. Coli Based on CRISPR-Prime Editing." *Nature Communications* 12 (1): 5206.
- Vergnaud, G., and F. Denoeud. 2000. "Minisatellites: Mutability and Genome Architecture." *Genome Research* 10 (7): 899–907.

Reviewer Reports on the First Revision:

Referees' comments:

Referee #1 (Remarks to the Author):

The reviewer appreciates the authors for their thorough responses and revised manuscript that includes new data and improved performance of the DNA Ticker Tape system. There are no further concerns. This work is fantastic and should be published.

Referee #2 (Remarks to the Author):

The authors have addressed our previous concerns. Specifically, the inclusion of the new monophyletic DNA Ticker Tape/sc-RNAseq experiment addresses many of the previous concerns we had raised.

Referee #3 (Remarks to the Author):

This is a much improved paper that addresses all major issues raised in the initial review. A few minor outstanding points:

-I appreciate the analysis of doubling time with the recording components expressed. It would be nice to include a statistical analysis of this data.

-I remain concerned that the name DNA Ticker Tape is overly general. pegRNA Ticker Tape, PE Ticker Tape, Ticker Tape Prime or something similar would help distinguish the approach from other current approaches and inevitable future approaches.

-Figure 1 schematic is much improved. However, Figure 1b could still be annotated/colored in a way that is closer to 1a (e.g. write head bases going from red to gray with the movement of the write head), or 1b could be removed.

Author Rebuttals to First Revision:

We thank for the provisional acceptance of our manuscript. Below please find our point-by-point response to reviewers' comments. Reviewers' comments are in blue text, and our responses in black text.

Referees' comments:

Referee #1 (Remarks to the Author):

The reviewer appreciates the authors for their thorough responses and revised manuscript that includes new data and improved performance of the DNA Ticker Tape system. There are no further concerns. This work is fantastic and should be published.

We thank the reviewer's enthusiasm for our work, and for their previous comments which substantially improved the manuscript.

Referee #2 (Remarks to the Author):

The authors have addressed our previous concerns. Specifically, the inclusion of the new monophyletic DNA Ticker Tape/sc-RNAseq experiment addresses many of the previous concerns we had raised.

We thank the reviewer for their previous suggestion of the sc-RNA-seq experiment, which we agree greatly improved the manuscript.

Referee #3 (Remarks to the Author):

This is a much improved paper that addresses all major issues raised in the initial review.

We thank the reviewer's enthusiasm for our work, and for their previous comments which substantially improved the manuscript.

A few minor outstanding points:

-I appreciate the analysis of doubling time with the recording components expressed. It would be nice to include a statistical analysis of this data.

Following the reviewer's suggestion, we have performed a statistical analysis of the doubling time data (added sentence in EDF6a legend is underlined):

Extended Data Figure 6a. Cell doubling times measured for HEK293T and the monoclonal lineage tracing cell line (iPE2(+) LT(+)), with or without Doxycycline (Dox). The presence of Dox lengthened the cell doubling time, possibly negatively affecting the cell physiology. P values were obtained using the two-tailed Student's t-test with Bonferroni correction: only *P < 0.05 are shown.

-I remain concerned that the name DNA Ticker Tape is overly general. pegRNA Ticker Tape, PE Ticker Tape, Ticker Tape Prime or something similar would help distinguish the approach from other current approaches and inevitable future approaches.

On further consideration, we understand and accept the reviewer's concern. However, although we appreciate the reviewer's specific renaming suggestions, we find them to be unwieldy, and moreover prime editing is a means to an end, rather than an essential aspect of the concept.

To address the concern, we have changed the name of the method to “DNA Typewriter”, which reflects the following characteristics of the system:

1. The thousands of discrete symbols which can be encoded, analogous to the typeset of a typewriter.
2. The linear, ordered, unidirectional manner in which these symbols are written to a defined medium, analogous to how typewriters work.
3. The fact that only a single site with the monomeric array is competent to be edited at any given moment, analogous to the type-guide of a typewriter.

Taken together, these typewriter-like characteristics distinguish our approach from predecessor systems for molecular recording, and consistent with that, the term has not previously been used to describe a molecular recorder.

We have also replaced our use of the term “write-head” with “type-guide” (as although conceptually similar, the former term is used for disk drives, and the latter for typewriters).

-Figure 1 schematic is much improved. However, Figure 1b could still be annotated/colored in a way that is closer to 1a (e.g. write head bases going from red to gray with the movement of the write head), or 1b could be removed.

We have revised Figure 1b as suggested, by coloring the write head bases going from red to gray with the movement of the write-head (type-guide in the revised manuscript):